# Prolonged persistence of mutagenic DNA lesions in somatic cells

Michael Spencer Chapman[1,2], Emily Mitchell[1,3,4], Kenichi Yoshida[1], Nicholas Williams[1], Margarete A. Fabre[1,3,4,5], Anna Maria Ranzoni[1], Philip S. Robinson[1], Lori D. Kregar[1], Matthias Wilk[6], Steffen Boettcher[6], Krishnaa Mahbubani[7,8], Kourosh Saeb Parsy[7,8], Kate H. C. Gowers[9], Sam M. Janes[9], Stanley W. K. Ng[1], Matt Hoare[10], Anthony R. Green[3,4], George S. Vassiliou[1,3,4], Ana Cvejic[1,3,4,11], Markus G. Manz[6], Elisa Laurenti[3,4], Iñigo Martincorena[1], Michael R. Stratton[1], Jyoti Nangalia[1,3,4], Tim H. H. Coorens[1,12] & Peter J. Campbell[1,3,4 ✉]

DNA is subject to continual damage, leaving each cell with thousands of individual DNA lesions at any given moment[1–3]. The efficiency of DNA repair means that most known classes of lesion have a half-life of minutes to hours[3,4], but the extent to which DNA damage can persist for longer durations remains unknown. Here, using high-resolution phylogenetic trees from 89 donors, we identified mutations arising from 818 DNA lesions that persisted across multiple cell cycles in normal human stem cells from blood, liver and bronchial epithelium[5–12]. Persistent DNA lesions occurred at increased rates, with distinctive mutational signatures, in donors exposed to tobacco or chemotherapy, suggesting that they can arise from exogenous mutagens. In haematopoietic stem cells, persistent DNA lesions, probably from endogenous sources, generated the characteristic mutational signature SBS19[13]; occurred steadily throughout life, including in utero; and endured for 2.2 years on average, with 15–25% of lesions lasting at least 3 years. We estimate that on average, a haematopoietic stem cell has approximately eight such lesions at any moment in time, half of which will generate a mutation with each cell cycle. Overall, 16% of mutations in blood cells are attributable to SBS19, and similar proportions of driver mutations in blood cancers exhibit this signature. These data indicate the existence of a family of DNA lesions that arise from endogenous and exogenous mutagens, are present in low numbers per genome, persist for months to years, and can generate a substantial fraction of the mutation burden of somatic cells.

A diverse set of mechanisms has evolved to repair DNA lesions such as adducted, methylated or oxidized bases[14]. Mutations arise when there are errors in DNA repair or when there is misincorporation opposite an unrepaired lesion during DNA replication. The high rate at which many DNA lesions occur in the genome demands that DNA repair must be equally efficient, meaning that the half-life of an individual lesion is typically much shorter than the time between cell divisions[3,4]. However, a recent study in mice exposed to a single, high dose of the alkylating agent diethylnitrosamine (DEN) showed that some DNA lesions can persist unrepaired through several cell cycles, generating different mutations at each round of replication[15]. Whether this phenomenon extends to other types of DNA damage, especially endogenously derived lesions in humans, remains unknown.

We hypothesized that high-resolution phylogenetic trees of somatic cells would enable us to infer the persistence of endogenous or exogenous DNA lesions across multiple cell cycles (Fig. 1). In a phylogenetic tree of somatic cells, each branch point, formally known as a coalescence, records a historic cell division[16]—successive branch points that trace a 'line of descent' from root to tip record different cell divisions through the ancestry of that clone. A given DNA lesion that persisted across several cell divisions would have the potential to generate a mutation each time the strand with the lesion was replicated, and these separate mutations should be detectable in the phylogeny. If different bases were misincorporated opposite the lesion during sequential rounds of DNA replication, closely related clones would carry two alternative mutations at the same position in the genome ('multi-allelic' variants; Fig. 1a,b), as described in the mouse model of DEN exposure[15]. Furthermore, if the persistent lesion had the same base misincorporated on the opposite strand during different rounds of replication, those mutations could, under some

[1]Wellcome Sanger Institute, Hinxton, UK. [2]Department of Haemato-oncology, Barts Cancer Institute, London, UK. [3]Cambridge Stem Cell Institute, Cambridge, UK. [4]Department of Haematology, University of Cambridge, Cambridge, UK. [5]Centre for Genomics Research, Discovery Sciences, BioPharmaceuticals R&D, AstraZeneca, Cambridge, UK. [6]Department of Medical Oncology and Hematology, University of Zurich and University Hospital Zurich, Zurich, Switzerland. [7]Department of Surgery, University of Cambridge, Cambridge, UK. [8]Cambridge Biorepository for Translational Medicine, NIHR Cambridge Biomedical Research Centre, University of Cambridge, Cambridge, UK. [9]Lungs For Living Research Centre, UCL Respiratory, University College London, London, UK. [10]Early Cancer Institute, University of Cambridge, Cambridge, UK. [11]Biotech Research and Innovation Centre (BRIC), University of Copenhagen, Copenhagen, Denmark. [12]Broad Institute of MIT and Harvard, Cambridge, MA, USA. ✉e-mail: pc8@sanger.ac.uk

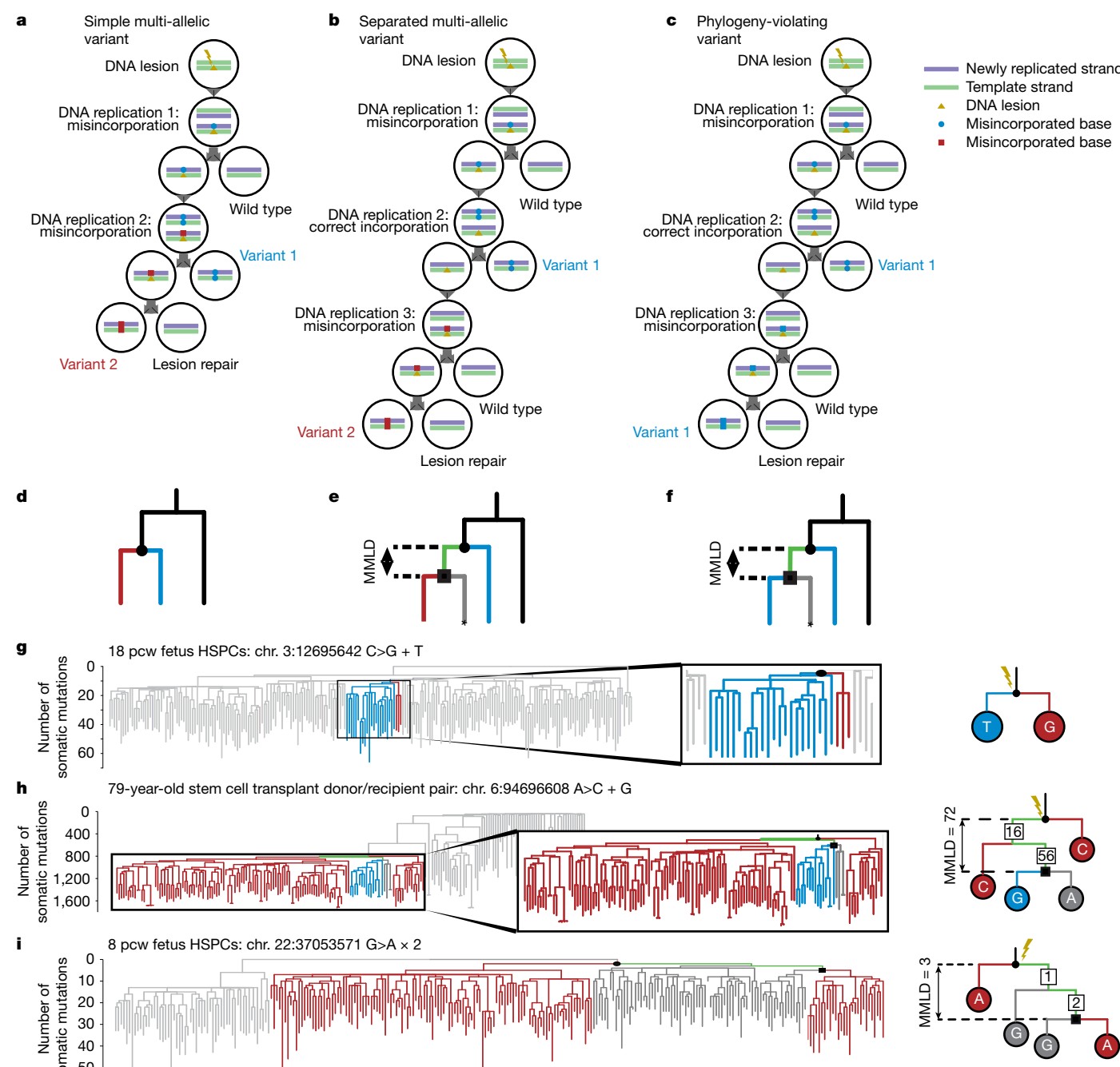

**Fig. 1 | Types of variants that result from persistent DNA lesions. a**, Mechanism of generation of a simple MAV. **b**, Mechanism of generation of a separated MAV. **c**, Mechanism of generation of a PVV. **d**, Appearance of phylogeny resulting from events in **a**. **e**, Appearance of phylogeny resulting from events in **b**. The green line represents the lesion path and the asterisk highlights the subclade that is negative for the mutation resulting from non-mutagenic replication. MMLD, minimum molecular lesion duration. **f**, Appearance of phylogeny resulting from events in **c**. **g**, Example of a simple MAV from the 18 post-conception week (pcw) fetal phylogeny. Branches coloured red have progeny with the C>G variant, and blue branches have progeny with the C>T variant, both on chromosome (chr.) 3, position 12695642. Right, zoomed-in view of the relevant portion of the tree. **h**, Example of a separated MAV. Red branches have progeny with the A>C variant, and blue branches have progeny with the A>G variant, both on chromosome 6, position 94696608. Grey branches have progeny with the reference allele. Right, zoomed-in view of the relevant portion of the tree, with numbers in squares representing the number of mutations assigned to that branch. **i**, Example of a PVV from the 8 pcw fetal phylogeny. Branches coloured red have progeny with the G>A variant on chromosome 22, position 37053571.

circumstances, be recognized through their contravention of the consensus phylogeny (phylogeny-violating variants (PVVs); Fig. 1c and Extended Data Fig. 1). Depending on the pattern of base-incorporation probabilities during translesion synthesis, a given lesion may give rise predominantly to PVVs, multi-allelic variants (MAVs) or a mixture of both.

## Phylogenetic trees of somatic cells

We collated seven published sets of somatic phylogenies from whole-genome sequencing of single-cell-derived colonies[5–9], organoids[10] or laser-capture microdissections (LCMs)[11,12]. The dataset comprises 103 phylogenies from 89 individuals, encompassing a total of 11,429

whole genomes, with a median of 48 samples per individual (range, 11–451; Supplementary Table 1). Each phylogeny was generated from a single tissue type: haematopoietic stem and progenitor cells (HSPCs, $n = 39$), bronchial epithelial cells ($n = 16$) or liver parenchyma ($n = 48$ from 34 individuals, owing to separate phylogenies for 8 anatomical segments of the liver in 2 participants). The HSPC phylogenies were from individuals that fell into five categories: fetal and cord blood ($n = 4$), healthy adults ($n = 13$), stem cell transplant donor/recipient pairs ($n = 10$), patients with myeloproliferative neoplasms ($n = 10$) and patients who had been exposed to chemotherapy ($n = 2$). Variant calling, filtering and reconstruction of phylogenetic trees were undertaken using established and extensively validated pipelines, as described previously[5–12].

## Mutations from persistent DNA lesions

To identify MAVs, we examined all phylogenies for genomic positions where we recorded two or more mutant alleles, revealing 1,079 such sites. Such events may occur by chance if the same position happens to mutate in two lineages independently—indeed, for many of these events, the two clades reporting the MAVs were far apart from one another on the tree and did not share a single line of descent, suggesting that they were not generated from the same persistent DNA lesion ($n = 727$; Extended Data Fig. 2). However, 352 MAVs were close together on the phylogenetic tree and within a single line of descent, a pattern that would be consistent with a persistent DNA lesion. For MAVs found in phylogenies built from clonal samples ($n = 293$), the precise organization of mutant clades could be established. In 80% of these cases (233 out of 293), the two mutant clades had the same ancestral node (Fig. 1a,d,g), whereas in the others (approximately 20% (60 out of 293)), the mutant clades could be linked to a single 'lesion path' through the phylogeny (Fig. 1b,e,h). We refer to these orientations as 'simple' and 'separated', respectively, whereas we term MAVs that are sufficiently distant on the phylogeny to be inconsistent with a single DNA lesion 'unrelated'.

We used three approaches to assess whether these simple and separated MAVs could plausibly have arisen through two independent mutations at the same locus. First, we simulated the null model of MAVs that occur as unrelated mutations to estimate the proportion anticipated to occur in simple and separated orientations by chance. The observed data had a 28-fold higher proportion of simple MAVs and a 3.8-fold higher proportion of separated MAVs than would be predicted by the simulations (Fig. 2a,b and Extended Data Fig. 3a). Second, we assessed MAVs with nearby heterozygous germline polymorphisms for phasing (Extended Data Fig. 3b). A prerequisite for MAVs being caused by a single lesion is that the phasing is to the same parental copy of the chromosome. As expected, MAVs in an orientation inconsistent with generation by a single persistent lesion (unrelated MAVs) had approximately equal proportions of matching and conflicting phasing (128 matching out of 230 total, $P = 0.10$, binomial test). By contrast, the phasing was almost universally matched for both simple and separated MAVs (78 out of 81, $P = 7 \times 10^{-20}$ for simple MAVs; 21 out of 24, $P = 0.0009$ for separated MAVs; Fig. 2c). However, when the mutant clades were separated by two or more nodes with non-mutant clades, they tended towards a more equal distribution of matching and conflicting phasing (Extended Data Fig. 3c), implying that a subset of these MAVs were not caused by a persistent DNA lesion. Therefore, these MAVs ($n = 21$), as well as those with non-matching phasing, were excluded from downstream analysis. Third, we compared the distribution of base changes and local sequence context (the mutation spectrum), for MAVs against the one that is expected to arise from two independently occurring mutations at the same base. This demonstrated that the unrelated MAVs had a very similar spectrum to the one expected for independent mutations (Extended Data Fig. 4), whereas the spectrum of simple and separated MAVs was distinct.

To identify PVVs, we developed a statistical approach to detect mutations for which there was excessive variability (overdispersion) in read counts reporting the variant, either within or outside its assigned branch on the tree (Extended Data Fig. 1e and Methods). As accurate phylogenies are essential for such inference, we included only phylogenies that were built from single-cell-derived samples, excluding the liver samples. This identified 841 mutations that violated the phylogeny. For 239 of these mutations, the locations of the different subclones reporting the PVV on the phylogenetic tree were not consistent with generation by a single persistent DNA lesion—as for MAVs, these probably arose through two separate mutations in independent lineages.

The remaining 602 mutations were in an orientation that would be consistent with a persistent DNA lesion. However, these may also occur owing to independent acquisition by chance; furthermore, incorrect reconstruction of the phylogenetic tree, loss of heterozygosity (LOH) or spontaneous reversion of a somatic mutation within a subclade may also result in a false-positive PVV call. We systematically evaluated each of these possible mechanisms of PVV generation using simulation, phasing, copy number and signatures (Fig. 2d–h and Extended Data Fig. 5; a detailed discussion and analysis of each possible artefactual source of PVVs is reported in Methods). Overall, alternative mechanisms accounted for only a small proportion of identified PVVs, which we excluded from further analysis.

In summary, a substantial majority of MAVs and PVVs occurring in close proximity on the phylogenetic tree cannot be accounted for by two independent mutational events or other trivial explanations. After excluding those variants that could be explained by alternative mechanisms, we took forward a final dataset of 328 MAVs and 490 PVVs (467 single nucleotide variants (SNVs) and 23 insertion–deletion mutations (indels)) for downstream analysis (Extended Data Figs. 6 and 7 and Supplementary Table 2).

## Numbers and signatures of MAVs and PVVs

The ability to detect MAVs and PVVs depends on having several nodes in the phylogenetic tree during the timespan of lesion persistence. Phylogenies that have a rich clonal structure therefore provide most statistical power for their detection—as expected, there was a correlation between the number of post-development nodes in the phylogeny and the number of detected MAVs and PVVs (Extended Data Fig. 8a). However, the number of MAVs per node varied substantially across tissues, being more than ten times higher in bronchial and liver samples than in the HSPC samples (bronchial, 0.36 MAVs per node; liver, 0.22 MAVs per node; HSPCs, 0.02 MAVs per node; $P = 0.04$ for between-group differences, Kruskal–Wallis test; Fig. 3a). Some bronchial epithelium phylogenies had particularly high numbers of MAVs, typically from current or ex-smokers ($P = 0.04$ for MAVs per node for never smokers versus smokers with at least 30 pack years, Kruskal–Wallis test), suggesting that lesions resulting from mutagens in tobacco smoke can persist over multiple cell cycles and lead to variable base incorporation during replication. The MAVs in bronchus and liver phylogenies had a similar spectrum, dominated by T>C/T>A mutation pairs, with some enrichment at ApT dinucleotides (Fig. 3b). This has most resemblance to the predicted MAV signature that would arise from SBS16 (Extended Data Fig. 8b–d), a signature of unknown aetiology that is increased in liver[11,17,18] and tobacco-exposed lung[10].

Most of the PVVs that we identified were in the HSPC phylogenies, although this was explicable by the greater statistical power for detection in these trees (Extended Data Fig. 9). The PVVs in the adult HSPC phylogenies had a distinctive mutational signature, characterized by C>T transitions particularly at CpT dinucleotides (Fig. 3c). It most closely matches COSMIC signature SBS19 (cosine similarity, 0.96) and has the same transcriptional strand bias for G on the untranscribed strand ($P = 0.04$, two-sided Poisson test; Fig. 3d), suggesting that it is

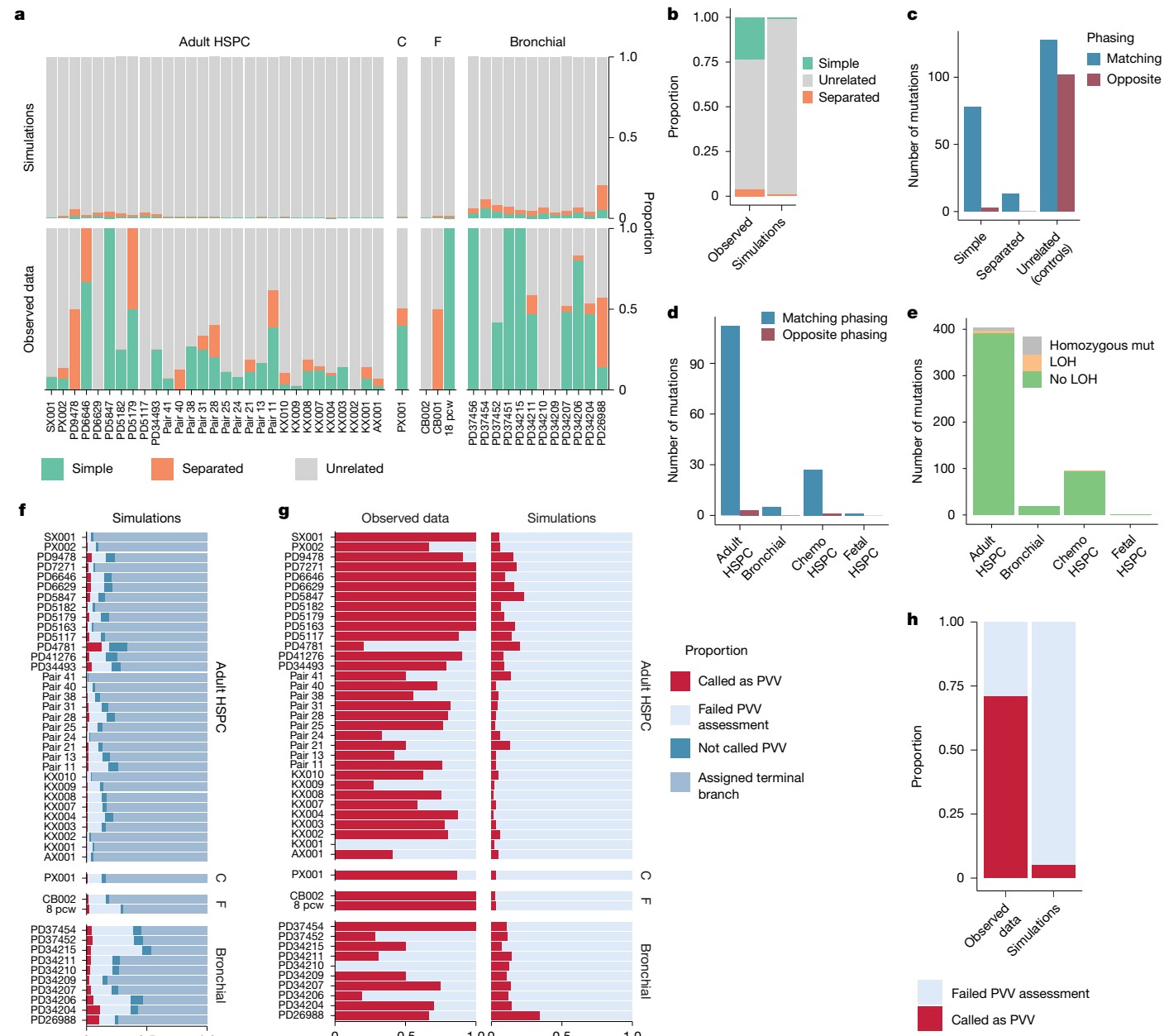

**Fig. 2 | Validation of MAVs and PVVs. a**, Bar plot showing a comparison of the proportion of MAVs occurring in simple or separated orientations for each individual, and in the simulated null model of occurrence by independent mutation acquisition. C, chemotherapy-exposed HSPCs; F, fetal HSPCs. **b**, Overall proportions of simulated independent MAVs that would be classified as simple, separated or unrelated compared with those seen in the data. The simulation proportions are weighted by the total number of MAVs called in the actual dataset for each participant to reflect their contribution to the MAV dataset. **c**, Phasing of the two mutant alleles of MAVs, including unrelated MAVs (those in an orientation inconsistent with generation by a persistent DNA lesion). **d**, The results of a phasing comparison of the positive subclades of PVV, including only those with two or more positive subclades for which phasing could be

confirmed. Chemo HSPCs, chemotherapy-exposed HSPCs. **e**, Stacked bar plot showing the results of a copy number analysis of the PVV negative subclade(s) to identify LOH. mut, mutation. **f**, Results of simulation of two independent mutations at the same site, showing that very small proportions would be detected and classified as PVVs using the described approach, compared with those either not detected by the analysis algorithm or those excluded from downstream analysis. **g**, As in **f**, results of simulation of apparent PVVs caused by two independent mutations, showing those that are detected by the analysis algorithm. **h**, Overall proportions of simulated independent PVVs that would be classified as a PVV, compared with the observed data. Proportions are weighted by the total number of PVVs called in the actual data for each participant to reflect their contribution to the PVV dataset.

the guanine that carries the lesion. The high number of PVVs and lack of MAVs at a similar sequence context, suggest that translesion synthesis in this case is highly specific for incorporation of a pyrimidine opposite the damaged guanine but there is a lack of specificity for which pyrimidine is selected (Extended Data Fig. 9d).

Several previous studies have noted that the characteristic mutational spectrum of HSPCs is different to that seen in other tissues, with

more pronounced peaks of C>T at CpT dinucleotides[5,16,19]. This normal HSPC mutational spectrum could be accurately reconstructed from a combination of SBS1, SBS5 and SBS19 (Extended Data Fig. 10), with SBS19 contributing 16% of mutations overall. By contrast, the spectrum of mutations in the normal cells of other tissues[20–23] was effectively reconstructed from SBS1 and SBS5 alone, with SBS19 contributing only up to 4% (Extended Data Fig. 10b). SBS19 was first identified in a subset

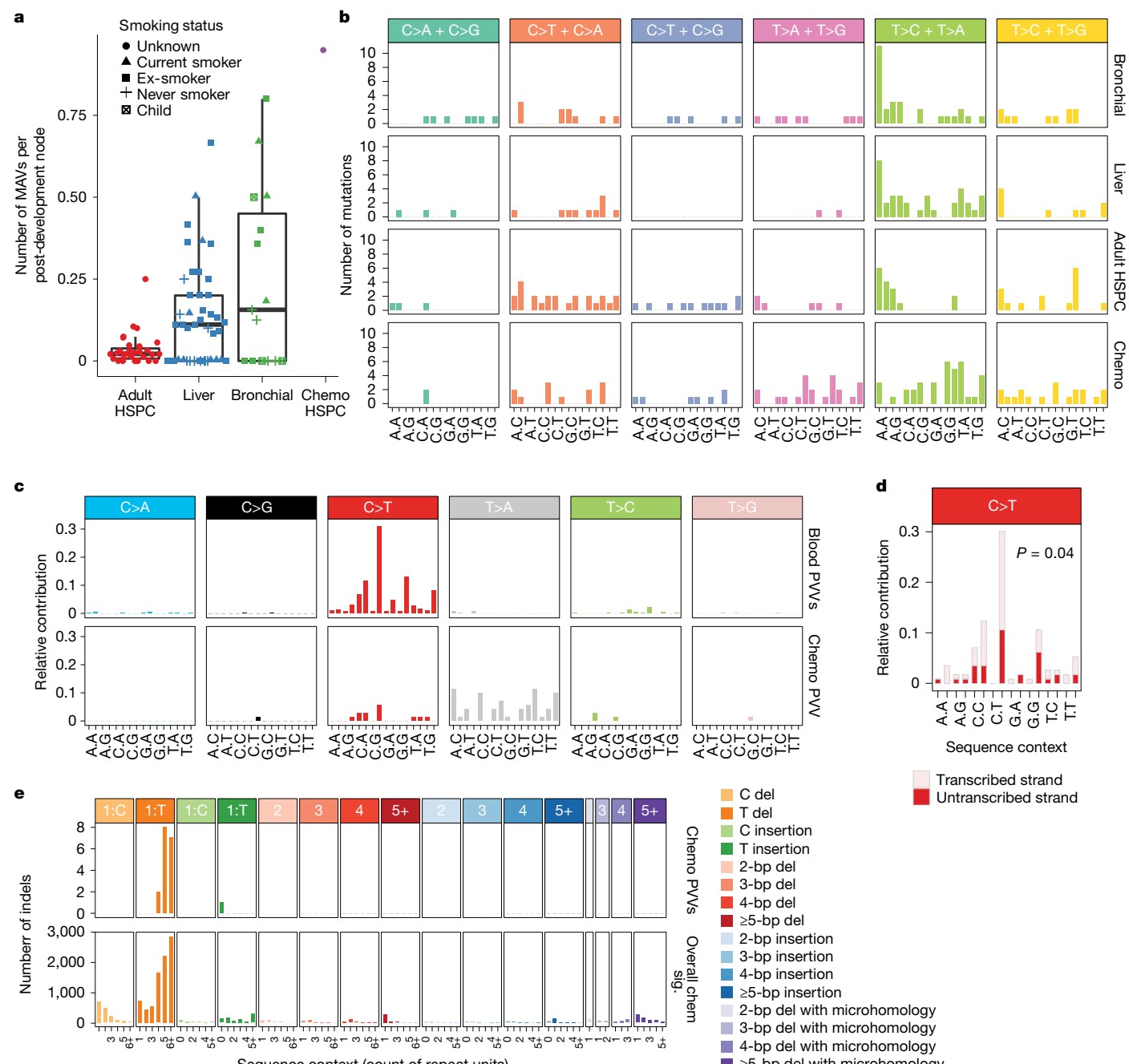

**Fig. 3 | Signatures of MAVs and PVVs. a**, Box plot showing the number of MAVs per post-development node for each sample, divided by tissue type, with raw data superimposed. Smoking status, where known, is indicated by shape. Data derived from *n* = 37 independent individuals for adult HSPCs; *n* = 48 for liver; *n* = 16 for bronchial epithelium; and *n* = 2 chemotherapy-exposed patients. **b**, Custom 96-profile MAV signatures, displaying the 6 possible SNV combinations that make up an MAV and, for each, the 16 possible combinations of flanking bases around the reference base. These are shown for MAVs in bronchial, liver, adult HSPC and chemotherapy-exposed HSPC phylogenies. MAVs incorporating an indel or multi-nucleotide variant are excluded. **c**, A 96-channel spectrum of the observed PVVs in adult HSPC phylogenies, and the chemotherapy-exposed HSPC phylogeny. The trinucleotide context is shown as four sets of four bars, grouped by whether an A, C, G or T is 5′ to the mutated base, and within each group of four by whether A, C, G or T is 3′ to the mutated base. **d**, Transcription strand bias of C>T mutations in the adult HSPC PVVs. The *P* value refers to a two-sided Poisson test for strand bias, not corrected for multiple testing. **e**, Signature of indel PVVs in the chemotherapy-exposed HSPCs, and the overall indel signature in the same samples. The coloured bars across the top denote the size of the insertion or deletion (del) as indicated; within each facet, the *x* axis denotes the count of repeated units in the reference genome matching the inserted or deleted sequence. In box plots in all figures, the centre line denotes the median, the box delineates the interquartile range and whiskers extend to the largest and smallest values that are no more than 1.5 times the interquartile range.

of blood cancers, liver carcinomas and pilocytic astrocytomas[13,24], but its aetiology is unknown.

Chemotherapy can induce a range of distinct lesions and mutational signatures[25]. The chemotherapy-exposed HSPC phylogeny showed increased numbers of MAVs and PVVs, each with a distinctive spectrum. MAVs showed marked dominance of mutations at T:A pairs, although with minimal context-specificity. A notable feature was mixed SNV and indel MAVs, with 10 out of 12 representing single nucleotide T deletions

at CpT sites combined with T>A or T>G transversions. Out of the 90 detected PVVs, 19 (21%) were indels, a far higher proportion than in the rest of the dataset (2%). These were predominantly single nucleotide T deletions at homopolymer tracts of four or more T bases, mirroring the overall indel signature in this individual (Fig. 3e). The remaining 67 SNV PVVs were predominantly T>A transversions (78%; Fig. 3c). These data suggest that the chemotherapy that this patient received, which included alkylating agents, generated many DNA lesions that persisted through multiple cell divisions. It is notable that the same lesion could generate an indel in one round of replication and a substitution in another (indel/SNV MAVs); two indels interspersed with a correct base incorporation (indel PVVs); or two identical substitutions interspersed with a correct base incorporation (substitution PVVs). This provides in vivo evidence for biochemical studies of translesion synthesis showing that a single lesion can result in indels, SNVs or correct base incorporations, depending on whether slippage or extension occurs during lesion bypass[26,27].

## Onset and duration of persistent lesions

Mutations in HSPCs accumulate at a constant rate throughout postnatal life, with this rate showing minimal cell-to-cell variation either within or between healthy individuals[5,19,21,28]. At birth, blood cells have around 50 mutations each[5,19], and these are acquired at relatively constant rates through the 38 weeks of gestation[7]. Since we know the nodes on the tree at which a persistent DNA lesion must have existed and, for PVVs and separated MAVs, the earliest node at which it was either repaired or lost from the observed phylogeny, we can estimate the chronological age at which it occurred and a lower bound on the length of time it persisted unrepaired.

Of note, 24 MAVs and 16 PVVs were likely to have been acquired in utero, as they could be timed to nodes at less than 50 mutations of molecular time (the average mutation burden in cord blood cells[5,19]; furthermore, some were identified in the phylogenetic trees of fetal HSPCs (Fig. 1g,i). In three cases, the causative lesion could be traced to the fertilized egg[7,29] or, alternatively, one of its daughter cells if the mutation arose during translesion resynthesis[30] or there was extreme lineage bias after the first cell division[31] (Figs. 1i and 4a). In a further seven cases, the lesion could be traced to a cell in the first five cell divisions of life. A previously published somatic phylogeny also found an MAV present in multiple germ layers[32], consistent with a pre-gastrulation lesion. The rates of mutations arising in utero from persistent DNA lesions were lower than seen for post-development nodes—for example, the rate of MAVs in nodes timed to less than 50 mutations was approximately 0.004 MAVs per node, a fifth of the rate for adult nodes. Thus, persistent DNA lesions can occur in utero, albeit at lower rates than postnatally. Given the shielded environment of the fetus, it seems likely that this DNA damage arises through endogenous processes, although an exogenous mutagen that crosses the placenta cannot be excluded.

In adult blood, MAVs and PVVs occurred steadily throughout the lifespan in numbers commensurate with our power to detect them (Fig. 4b), consistent with their generation by a clock-like mutational process. As expected, the timing of PVVs and MAVs in tissues exposed to exogenous mutagens, namely the smoking-exposed bronchial epithelium and chemotherapy-exposed HSPCs, varied among individuals and through time, dependent on individual mutagen exposure (Fig. 4c).

We also estimated lower bounds on the duration of each molecular lesion for all PVVs and separated MAVs, corresponding to the number of mutations acquired elsewhere in the genome while the lesion persisted unrepaired. If the mutations elsewhere in the genome originate from clock-like processes, this minimum molecular duration can be converted to a minimum chronological duration. For PVVs in the adult HSPC phylogenies, most minimum lesion durations ranged between 10 and 100 mutations (median, 21; interquartile range, 12–37; Fig. 4d). This corresponds to a median for the minimum chronological duration of 1.3–1.5 years, and suggests that durations of more than 3 years are common (85th centile at 55 mutations of molecular time). The distribution of lesion durations varied depending on the mutation type, with C>T PVVs having shorter minimum durations than T>C PVVs (median, 21 versus 37.5 mutations respectively; $P = 0.02$, Mann–Whitney test; Fig. 4e), but did not vary significantly across the lifespan ($P = 0.11$, linear regression; Extended Data Fig. 10c). Durations of the separated MAVs were similar to the PVVs ($P = 0.29$, Mann–Whitney test; Fig. 4f).

The minimum durations of lesions generating PVVs in the bronchial and chemotherapy-exposed HSPC phylogenies were more variable and in many cases much longer when calculated in molecular time (Fig. 4d). However, since background mutation rates in these settings do not show the clock-like properties seen in unexposed HSPCs, we cannot convert these to chronological time—indeed, it is likely that the longer apparent molecular durations in these settings derives from shorter periods of accelerated mutation rates rather than long real-time durations.

## Frequency and properties of PVV lesions

Our framework for identifying PVVs requires that a lesion must persist across at least two nodes in the phylogeny and that the subclones with the PVV are separated on the tree by at least one wild-type subclone. This provides considerable constraint on our power to detect such events—despite this, we called 490 of them across the cohort, suggesting that the underlying lesions must be relatively frequent in the stem cell population. We used approximate Bayesian computation (ABC) to generate estimates of the distribution of lesion durations and their frequency in stem cells (Extended Data Fig. 11). We simulated complete aged haematopoietic stem cell (HSC) phylogenies with persistent DNA lesions using an uninformative prior on mean lesion duration, recording (1) if they would result in a detectable PVV and (2) the measured minimal lesion durations if detected. We then compared the simulations against the observed numbers and durations of C>T PVVs to obtain posterior estimates of their distribution (Methods).

The posterior distribution of the mean lesion duration had maximum density at 2.2 years of molecular time (95% credible interval, 1.6–3.0 years; Fig. 5a), broadly in keeping with the direct estimates calculated above (Fig. 4d). From the posterior distribution, 25% of lesions would be expected to last 3 or more years. The proportion of simulated lesions that would have generated detectable PVVs was, as expected, low with less than 1 in $10^5$ lesions introduced into complete phylogenies resulting in a detectable PVV. This implies that there would be, on average, eight such lesions present in any given cell at any given moment in time (95% credible interval, 2–16; Extended Data Fig. 11h).

We also calculated the base-incorporation probabilities opposite these lesions during genome replication. For a basic PVV structure crossing two nodes (Fig. 1f), detection of the PVV requires a fixed tree structure in which the two mutated subclones are separated by a wild-type subclone—such PVVs therefore offer no information on the base-pairing probability. However, 72 PVVs had a lesion path crossing more than 2 nodes, meaning that detection of the PVV does not depend on which base is incorporated for at least 1 subclone. The frequency of C versus T incorporations in these unbiased subclones was almost equal, with 41 C and 49 T base incorporations, giving a T pairing probability of 0.54 (95% confidence interval, 0.43–0.65). Whether these two alternative outcomes reflect stochastic base incorporation by a single DNA polymerase or alternative polymerases with different incorporation preferences is unclear. Both mechanisms have been observed in experimental models of translesion synthesis[33–35].

These estimates of the prevalence and misincorporation rates of PVV lesions accord well with the observed rates of SBS19 in HSPCs. With

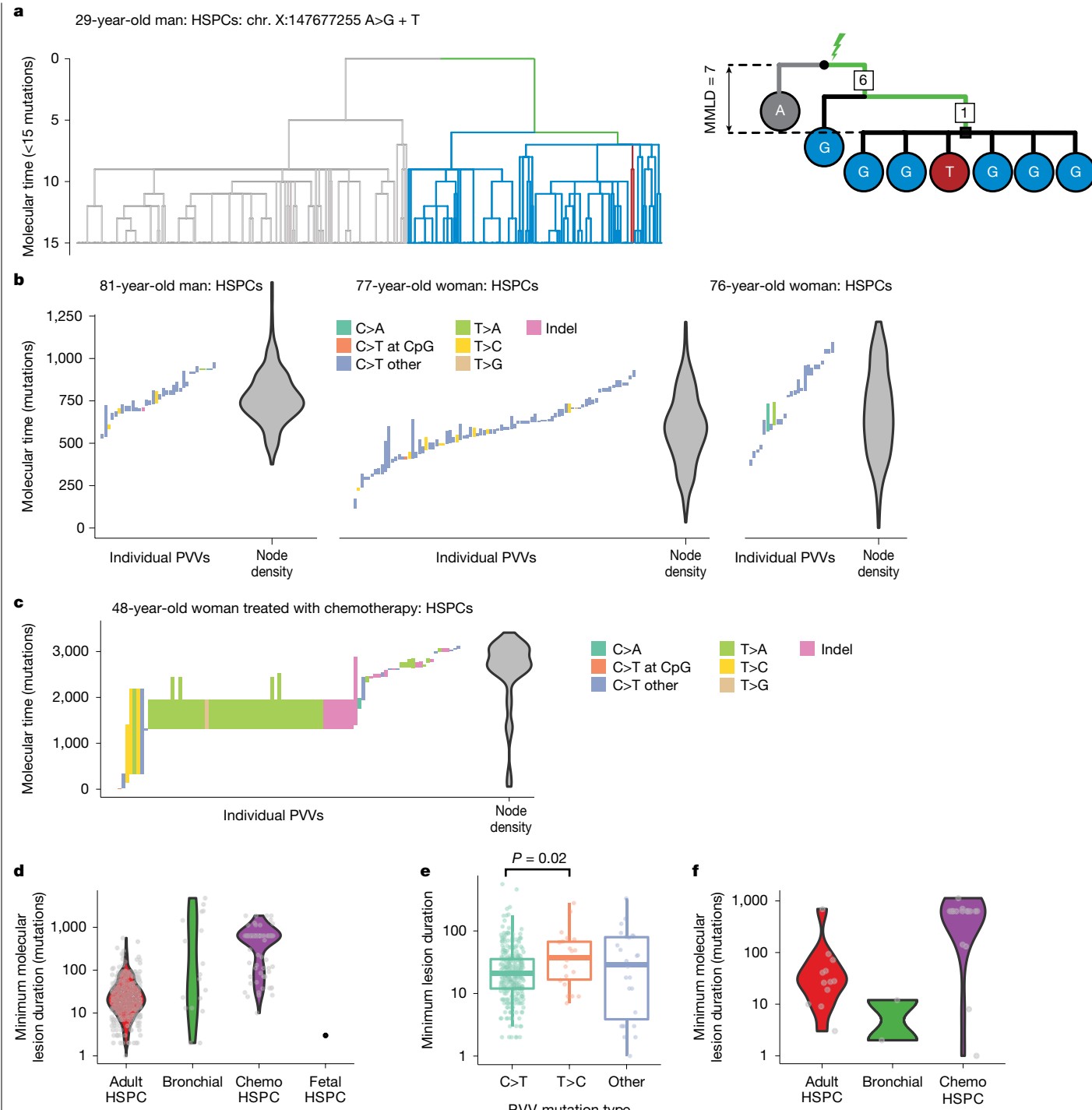

**Fig. 4 | Timing and duration of PVVs. a**, Example of a lesion causing an MAV present in the zygote or early blastomere. The phylogenetic tree has been truncated to the first 15 mutations of molecular time. Branches are coloured by whether descendants carry the A>G mutation (blue), A>T mutation (red) or reference allele (grey). Right, schematic showing the clade structure and lesion path, with numbers in squares representing the number of mutations assigned to that branch. **b**, Left, plot showing the latest time of lesion acquisition and the earliest time of lesion repair for the three adult HSPC phylogenies with the most PVVs observed. Each column represents an individual PVV-causing lesion, ordered by the time of lesion acquisition: the lowest value is the latest time of lesion acquisition; the highest is the earliest time of lesion repair; the height

therefore represents MMLD. Columns are coloured by mutation type. Right, the density of post-development internal nodes is shown as a violin plot (kernel-smoothed density). **c**, as in **b**, but for PVVs for PX001, the chemotherapy-exposed HSPC phylogeny. PVVs are clustered around 500–2,500 mutations of molecular time, which corresponds to the timing of chemotherapy in this patient. **d**, Violin plot showing the density of MMLDs of the detected PVVs by phylogeny category. Individual data points are superimposed. **e**, Box-and-whisker plot showing the MMLD of PVVs. Individual data points are superimposed. Data are derived from $n = 490$ PVVs. The $P$ value refers to a Mann–Whitney test for duration differences, uncorrected for multiple hypothesis tests. **f**, Violin plot showing density of MMLDs of separated MAVs.

16% of clock-like mutations in HSPCs deriving from SBS19 (Extended Data Fig. 10a,b), the rate of SBS19 would be around two to three mutations per HSPC per year[5,16,19]. The cell division rate of haematopoietic

stem cells in humans[36,37] is estimated to be 1–2 per year, suggesting a prevalence of around 8 persistent DNA lesions at any moment in time, half of which would generate a mutation in a given cell division, would

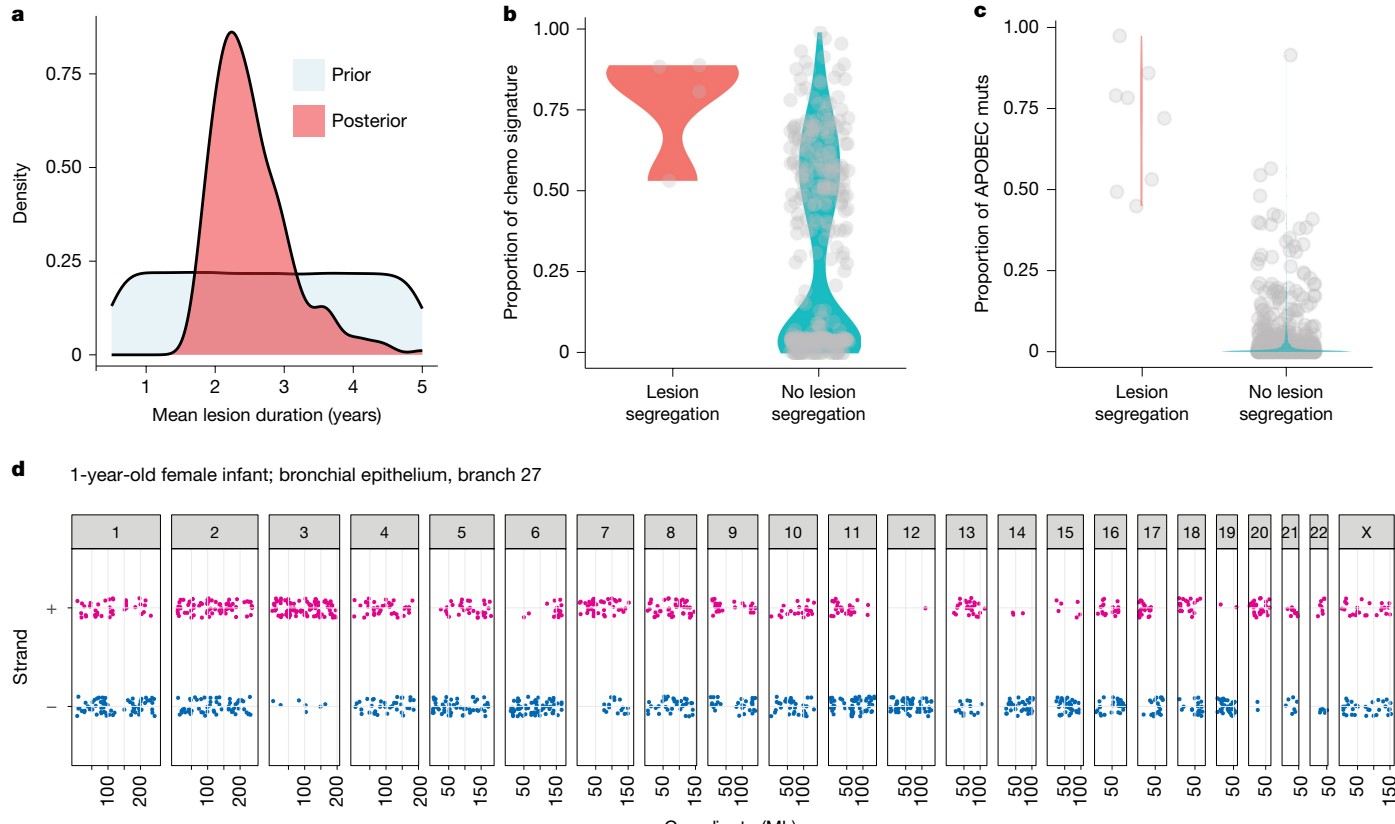

**Fig. 5 | Lesion segregation. a**, Density plots showing the prior and posterior distribution for mean lesion duration in molecular time, measured in numbers of mutations. **b**, Violin plot with overlying individual data points showing the proportion of chemotherapy signature 1 in branches of the chemotherapy-exposed HSPC phylogeny PX001, divided by those with significant lesion segregation ($n = 3$) and those without. **c**, Violin plot with overlying individual data points showing the proportion of APOBEC mutations from all bronchial epithelial phylogeny branches, divided by those with significant lesion segregation ($n = 8$) and those without. **d**, The chromosomal strand and position of mutations from a branch affected by APOBEC mutagenesis from bronchial epithelial phylogeny PD37456, demonstrating significant lesion segregation (chromosomes 3, 12 and 14, for example).

therefore generate around 4 mutations per cell per year. These calculations suggest that the frequency and persistence of PVV-causing DNA lesions is entirely sufficient to explain the observed rate of SBS19 mutations in haematopoietic stem cells. Following from this, we analysed whether SBS19 causes driver mutations in HSPCs. From genome and exome sequencing data of myeloid cancers, we found that SBS19 was responsible for 10% of coding mutations in myeloid cancer genes, including *DNMT3A*, *TET2*, *ASXL1* and *TP53*, and up to 16% in some genes (Extended Data Fig. 10d). PVVs showed no correlation with genomic features other than DNA methylation density, which showed a bimodal pattern of enrichment (Extended Data Fig. 10e and Supplementary Table 3)—with larger numbers of variants, it may be informative to compare the genomic distribution of DNA lesions acquired in development versus in adulthood.

## Strand asymmetry and lesion segregation

A key discovery in the paper that reported persistent DNA lesions in mice exposed to DEN was strand asymmetry of the mutations[15]. With the one-off dose of DEN, adducts were generated on thymines on both strands of a given chromosome in a given cell—at the next cell division, the two daughter cells inherited one each of those strands, but only one daughter cell seeded the eventual liver cancer. This created marked asymmetry of T>N versus A>N mutations on a chromosome-by-chromosome basis in the tumour—this asymmetry depends on numerous lesions per chromosome generated in a single cell cycle with limited dilution from mutations in other cell cycles.

We deployed methods for detecting strand asymmetry of mutations[15] to analyse individual branches from our phylogenies for such lesion segregation. As expected, most phylogenies did not have any branches with evidence of strand asymmetry because of the clock-like properties of most mutational processes in these cells. We did detect lesion segregation in four branches of the chemotherapy-exposed HSPC phylogeny, with positive branches all having large contributions of the chemotherapy mutational signature (Fig. 5b), consistent with the patterns seen in the DEN mouse model[15].

Notably, 6 out of 16 bronchial epithelial phylogenies had at least 1 branch with significant strand asymmetry. These were not from smokers with many MAVs, but instead derived from branches carrying large proportions of mutations from mutational signatures SBS2 and SBS13 (Fig. 5c). These signatures are caused by the base-editing activity of APOBEC enzymes that act on cytosines[38–42]. The distribution of strands affected was such that about half of the chromosomes showed APOBEC mutations that were equally balanced between forward and reverse strands, with a quarter each showing marked asymmetry to the forward or to the reverse (Fig. 5d)—these samples did not carry structural variants or copy number changes that could explain the observations. Such proportions could only occur if APOBEC lesions were generated within a single cell cycle, with skewed patterns arising when the daughter cell inherited, for example, the forward strand of both parental copies of the chromosome. Studies of cell lines have reported that APOBEC mutagenesis can happen in episodic short bursts[43]—our data demonstrate that APOBECs can generate many hundreds to thousands of lesions across the whole genome within

a single cell cycle. Strand coordination of clustered APOBEC mutations near structural variants has been repeatedly observed in cancer genomes[38,41]—these were previously attributed to APOBEC acting on single-stranded DNA[38,42]. Our data suggest that strand coordination need not result from APOBEC modification of single-stranded DNA, but could arise from lesion segregation even with APOBEC acting on double-stranded DNA.

## Discussion

A diverse register of DNA lesions emerges from the quotidian chemistry of life coupled with the rather more elective chemistry of our lifestyles. For example, genotoxic aldehydes can arise from endogenous sources, through innate folic acid metabolism, and from exogenous sources, such as hepatic metabolism of dietary ethanol—a sophisticated repertoire of pathways has evolved to either detoxify them or repair the DNA lesions that they cause[44–46]. Other pathways replace or repair DNA carrying cross-links, bulky adducts, oxidative damage, ultraviolet light photoproducts or abasic sites[14]. Presumably, the most frequent and most damaging DNA lesions exert the strongest pressure for the evolution of rapid repair mechanisms; the corollary being that there may be a class of lesions that are less prevalent and/or less detrimental for which repair is slower. Such lesions may be invisible to the usual techniques for direct discovery through chemistry and mass spectrometry[47] because of their low prevalence, and their corresponding DNA repair pathways may be difficult to uncover with standard experimental and knockout approaches.

Here we used high-resolution phylogenetic trees based on normal human stem cells as an approach to deduce the presence of persistent DNA lesions that generate somatic mutations across successive cycles of genome replication. Although indirect, this approach provides a relatively comprehensive view of the lesion-to-mutation life cycle. For example, for the PVVs in blood, we can infer: that lesions occur steadily throughout life, including in utero, and are therefore a probable consequence of endogenous cellular processes; that lesions persist in the DNA for months to several years; that lesions preferentially affect guanines in an ApG context; that lesions are subject to transcription-coupled nucleotide excision repair; that lesions are present at a density of around 1 per billion bases in a given haematopoietic stem cell; and that DNA replication across the lesion has a 50:50 chance of a misincorporation or correct insertion opposite the lesion. These estimates of lesion prevalence and duration are orders of magnitude away from the hundreds to thousands of 8-oxogunanines and methylated bases that are present in a cell, with their associated half-lives of minutes to hours[3,4], data that have informed the high-frequency, rapid repair model of the lesion-to-mutation life cycle. The signature that emerges from persistent DNA lesions accounts for 16% of all mutations in blood cells, with similar proportions among the mutations that drive blood cancers, a fraction similar to that seen for mutations arising from, for instance, spontaneous deamination of cytosine (SBS1, 14%), a much more frequent lesion in the genome[2]. Thus, DNA damage that occurs at low frequency with slow repair also carries considerable threat to genomic integrity.

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

## Methods

### Curation of high-resolution phylogenetic trees

We combined data from seven previously published sets of somatic phylogenies. Each phylogeny was made up of cells from a single tissue type—haematopoietic stem and progenitor cells (HSPCs; 39 participants)[5–9], bronchial epithelial cells (16 participants)[10] or liver parenchyma (34 participants)[11,12]. The phylogenies were built from somatic mutations discovered in whole-genome sequencing of single-cell-derived colonies (HSPCs), single-cell-derived organoids (bronchial epithelium) or LCM from liver[48]. Details of the research participants, sample acquisition, sequencing and variant calling are provided in the original papers, and the clinical and demographic data are summarized in Supplementary Table 1. Written informed consent was obtained from all participants in the study under approvals from the relevant ethics committees—specific details are available in the original publications[5–12].

In total, we collected 103 phylogenies from 89 individuals, with 2 participants from the liver study providing 8 phylogenies each owing to independent sampling from all 8 anatomical segments of the liver in these individuals[12]. There was a median of 48 samples or clones per individual (range: 11–451). For the studies of the haematopoietic system, we defined 3 categories: (1) fetal and cord blood HSPCs ($n = 4$; 2 from fetal haematopoietic organs, 2 from cord blood); (2) adult HSPCs ($n = 28$; 10 from deceased donor bone marrow with no known blood disorder, 3 from individuals with known clonal haematopoiesis, 10 from patients with myeloproliferative neoplasms, 10 from donor/recipient pairs of allogeneic haematopoietic stem cell transplant); and (3) chemotherapy-exposed HSPCs ($n = 2$, 1 treated twice for Hodgkin's lymphoma with alkylating-agent-containing regimens; 1 treated with R-CVP, a regimen containing cyclophosphamide and vincristine)—the R-CVP chemotherapy-exposed blood phylogeny was analysed with the 'adult HSPCs' group, rather than the 'chemotherapy-exposed HSPCs' group as R-CVP did not have significant mutagenic consequences for the HSPC population.

For individuals with single-cell-derived samples, mutations were filtered using similar approaches, with combinations of filters designed to remove germline variants, sequencing artefacts and in vitro-acquired mutations. For individuals from the liver study, collected using laser-capture microdissection, a matched normal was used for mutation calling, with some different downstream filtering steps[11,12]. The numbers of mutations per sample varied considerably across individuals depending primarily on tissue type, donor age, mutagen exposure (such as smoking, alcohol or chemotherapy), and disease status.

For the single-cell-derived colonies, phylogenetic trees were inferred using a maximum parsimony algorithm. For samples from the liver LCM studies, phylogenetic trees were inferred in a two-step procedure: (1) the set of $n$-dimensional vectors of variant allele fractions for each mutation were clustered using a hierarchical Dirichlet process[11]; and (2) the phylogenetic tree describing these clusters was inferred using serial application of the pigeonhole principle[49]. The robustness of variant calling and phylogenetic tree reconstruction using these methods has been extensively tested, with further details available in the original manuscripts.

### Identification of MAVs

A somatic MAV occurs if a reference base at the same genomic position is mutated to two different mutant alleles in the same individual. For example, there may be evidence of both a C>A and C>T mutation at exactly the same chromosome and position. In the context of the phylogeny data analysed here, these different mutant alleles will be evident in different clones from the same individual.

Within the phylogeny of each participant, we identified mutation pairs with overlap of the mutated locus. For SNV pairs, this is simply identifying SNVs at the same chromosome and position. For deletions and multi-nucleotide variants (MNVs, affecting two or more nucleotides), any degree of overlap was classified as a MAV. Each MAV was classified as simple, separated or fail in a hierarchical manner, on the basis of the orientation of their allocated branches in the phylogeny. If the two mutations had the same parent node (as in Fig. 1d), the MAV was classed as simple, and the parent node classified as the lesion node. If this was not true, but the allocated branch of one mutation fell within the clade defined by the parent node of the other (as in Fig. 1e), the MAV was classed as separated and the parent node that encompassed both mutations was classified as the lesion node. If neither of these criteria were met, the MAV was classed as unrelated. Separated MAVs where the two mutant alleles were separated by two or more subclades with the reference allele had a higher probability of occurring via independent mutation events, as evidenced by the more equal proportion of matching to non-matching phasing comparisons (Extended Data Fig. 3c) and simulation. Therefore, these were reclassified as unrelated, even though a proportion were likely to be caused by a persistent DNA lesion. Unrelated MAVs were likely to have been caused by independent events occurring at the same genomic locus by chance, a hypothesis supported by their mutational signatures which closely resembled the expected signature for this mechanism (Extended Data Fig. 4).

For separated MAVs, the lesion path was defined by starting from the lesion node and working stepwise down the phylogeny along the path containing a mixture of alleles at the genomic position of interest. The last node encompassing two different alleles was classified as the lesion repair node. In most cases this node was only one branch from the lesion node (as in Fig. 1e), but there were occasional examples of longer lesion paths (Fig. 1h). MMLD was calculated as the sum of branch lengths between the lesion and lesion repair nodes.

### Identification of PVVs

A PVV is defined here as a mutation that is discordant with the consensus phylogeny. That is to say, the distribution across the phylogeny of clones carrying the mutant allele is not consistent with a single mutation-acquisition event and consistent inheritance in descendants thereafter.

The mutation assignment algorithm assumes that a single variant results from a single, fixed mutational event, after which all daughter cells carry the mutation. For the vast majority of mutations, these assumptions hold true: a branch exists that forms a clade containing only samples with the variant allele (Extended Data Fig. 1a). The point here is that the tree-building algorithm (treemut) assumes that each mutation has occurred exactly once, and finds the single branch with the maximum likelihood for explaining the observed data. For PVVs, these assumptions no longer hold true—with such variants, the assigned mutant clade either: (1) contains a subset of samples without the mutation (Extended Data Fig. 1b); or (2) does not contain all the samples with the mutation (Extended Data Fig. 1c).

The statistical challenge is how to pick out the minority of mutations that do not neatly fit a single acquisition event, amongst up to 100,000 or more somatic mutations per phylogeny. The approach we developed relies on the formal detection of overdispersion of read counts, described in detail below. Before settling on this approach, we experimented with two alternatives:

(1) Identifying mutations that had a low $P$ value calculated by the treemut algorithm. Each branch assignment by treemut comes with an associated $P$ value: the chance of observing the read counts in the data, given a single acquisition event on the assigned branch. However, whatever $P$ value cut-offs we used, there was low specificity and sensitivity for the variants of interest. For example, some indels present in large clades had consistently biased mutant versus wild-type read counts leading to low $P$ values, even when a variant was consistently present only within the assigned clade.
(2) Identifying mutations for which there were either: (1) negative samples within the assigned clade; or (2) positive samples outside the assigned clade. Similarly, this binary approach had poor sensitivity

and specificity. It frequently yielded occurrences where there were a few variant reads in a single sample outside the assigned clade (probably sequencing errors or other artefacts), or negative samples within the positive clade where depth was relatively low and there were no reads reporting the variant allele by chance.

Instead, a beta-binomial test for overdispersion proved a powerful approach to detect occurrences where there were in fact two distinct positive and negative populations either within or outside the assigned clade. This approach is analogous to that used to detect true somatic mutations from artefacts in the mutation-calling algorithms[50]. The reason this approach works better than the other two methods we tried is that the observed data represents a mixture of two binomial distributions—a given sample will report the variant allele at either the base call error rate (if the sample is wild-type) or with probability 0.5 (for a sample with the mutation heterozygously on an autosome). Under a simple mutation assigned to the correct branch of the phylogeny, the two binomial distributions will assort perfectly such that the descendants in the mutant clade all report the variant allele at 0.5 and those outside the clade report it at the base call error rate—there would be no overdispersion beyond this simple partition. For a PVV, however, there will always be a set of samples for which this partition of expected variant allele fraction is violated—either a wild-type subclone within the assigned mutant clade or a mutant subclone outside the assigned mutant clade. A test for overdispersion beyond that expected by chance would report on either scenario and formalizes the statistical inference.

In more detail, this can be detected by testing the read counts of all clades that should theoretically be uniformly positive or negative for the variant, and testing for overdispersion of the observed counts. We quantified the overdispersion by assuming the counts to come from a beta-binomial distribution and finding the maximum-likelihood $\rho$ parameter using the optim() and dbetabinom() functions from R packages stats and VGAM. As DNA lesions causing PVVs must occur on internal branches to allow detection, and will then at least partially follow the phylogeny, we reasoned that such mutations would invariably be allocated to internal branches. We therefore assessed only such internal branch mutations for overdispersion (1) within and (2) outside the clade formed by the assigned node, quantifying the maximum-likelihood $\rho$ parameter for each. PVVs will have a high $\rho$ (empirically set as ≥0.1) for one of these parameters. Additionally, we required strong evidence for either a 'negative' subclade within the assigned clade (no variant reads detected with a minimum depth of 13), or a 'positive' subclade outside the assigned clade (variant allele frequency (VAF) ≥ 0.25 and ≥3 variant reads). Mutations meeting either pair of corresponding criteria (typically ~1 in 500 internal branch mutations) were considered phylogeny-violating and taken forward for further assessment.

Next, we assigned the putative 'lesion node' for each PVV, namely the node containing all samples with the variant. For variants with evidence of overdispersion within the assigned clade (Extended Data Fig. 1b), this was the same as the assigned branch. For variants with evidence of overdispersion outside the assigned clade (Extended Data Fig. 1c), we iteratively travelled node-by-node up through the phylogeny, until there were no positive clades outside the clade defined by that node.

Finally, we iteratively worked down from the lesion node, attempting to define the lesion path through the phylogeny. If caused by a DNA lesion, PVVs should have a specific orientation: at each cell division, one daughter cell will contain the DNA strand resulting from replication opposite the lesion and will therefore have a fixed genotype at the locus of interest (either the reference or mutant allele), which will be consistently inherited by its progeny. This is therefore a uniform subclade with a consistent genotype throughout. The other daughter cell will inherit the lesion itself, and therefore still has the potential for generation of the two alternate alleles, and therefore seeds a 'mixed' subclade. The lesion path is defined by the path containing mixtures of genotypes, and the outcome of replication at each cell division defined by the genotype of each uniform subclade arising from this lesion path. Once both daughters of the assessed node are uniform (that is, one contains only mutant samples, the other contains only wild-type samples), the lesion repair node has been reached (Extended Data Fig. 1b). If both daughter nodes of the lesion node contain a mixture of positive and negative clades, this is inconsistent with generation by a persistent DNA lesion (Extended Data Fig. 1c). Such mutations were deemed to have been generated by an alternative mechanism—indeed, mutational signature analysis showed that they were predominantly C>T at CpG sites, consistent with their generation by independent mutations at the same site (Extended Data Fig. 5d,e).

### Phasing of MAVs and PVVs to validate their derivation from a single DNA lesion

To assess if the variants of a MAV pair were on the same allele (on the same chromosome copy of a homologous pair), we attempted to phase each variant with proximate heterozygous SNPs. Approximately a third of MAVs had a suitable heterozygous SNP sufficiently nearby for assessment. We extracted heterozygous SNPs within 1 kb of the mutation locus using the VCF files of mutations. Each read-pair crossing both the variant and a heterozygous SNP locus was categorized by the base supported at each (Extended Data Fig. 3b). For phylogenies built with single-cell-derived samples, matching phasing was confirmed if samples carrying each variant of an MAV pair had read-pairs with either (1) the same SNP base and their respective variant base or (2) the same SNP base and the reference base. For phylogenies built with inferred clones (the participants from the liver LCM studies), matching phasing was confirmed only if reads containing each variant base of an MAV pair had the same heterozygous SNP base—that is, the same SNP base phasing with the reference base was insufficient because of the potential inclusion of normal cells in the microdissection. The heterozygosity of the SNP was confirmed in each case. Phasing of the positive subclades of PVVs was similarly assessed. In cases with >2 positive subclades, we considered phasing as confirmed if one or more subclade pair was successfully phased.

### Assessment of two independent mutations as an artefactual cause of MAVs

MAVs are rare events, and a potential alternative mechanism for their generation is two independent mutations occurring at the same locus by chance. As independent events, they would be expected to be equally likely in any lineage at any time in their history. This differs from the expectation of MAVs occurring as a result of persistent DNA lesions, which are necessarily in closely related lineages. We therefore aimed to estimate the degree to which there was an excess of closely related MAVs in the data, compared to a null hypothesis that the MAVs result from independent events.

The branch lengths, scaled to numbers of mutations, are estimates for the amount of time passed in that lineage. We could therefore formally assess the proportion of MAVs that would be expected to fall in simple, separated or unrelated orientations by modelling MAV pairs occurring as random independent events within the phylogeny. To simulate this, we randomly selected pairs of phylogeny branches with probabilities proportional to their branch length, repeating this 50,000 times for each phylogeny. Each pair was categorized by the orientation of the two selected branches and compared with the proportions observed in the data. As for the observed data, any pair classified as separated but with two or more intervening negative subclades were reclassified as unrelated. To assess the overall degree to which the set of MAVs may be contaminated by those occurring by chance, we calculated a weighted mean of the simulated proportions in each category, using the total number of MAVs detected in each phylogeny as weights.

## Assessment of two independent mutations as an artefactual cause of PVVs

As with MAVs, PVVs may result from two independent mutations at the same locus by chance. In addition, PVVs may theoretically result from spontaneous reversion of a somatic mutation in a subclade of the original mutant clade, a phenomenon that has previously been observed in cases where wild-type cells have a selective advantage. However, the orientation of PVVs did not appear to be that expected from either of these mechanisms. To formally test this, we designed simulations of each, testing all phylogenies with at least one PVV.

(1) Independent mutations at the same site. Similar to the MAV simulations, we randomly selected pairs of branches with probabilities proportional to their branch lengths. Each sample was assigned a depth for the simulated PVV locus according to a random draw from a Poisson distribution with the λ parameter being the overall mean depth in that individual. Samples within clades formed by the selected branches, 'positive' samples, were assigned variant counts according to random binomial draws with $P = 0.5$ and $n = $ depth. Other 'negative' samples were assigned variant counts by similar random draws but with $P = 1 \times 10^{-6}$ (the error distribution). Analysis then proceeded as with the data: a single branch was assigned with treemut; if this branch was a terminal branch, there was no further analysis; if it was a shared branch, the counts within and outside the assigned branch clade were assessed for overdispersion using the beta-binomial distribution and the same $\rho$ thresholds as for the data; the lesion node was assigned and the mutation classified as 'pass' or 'fail'. For pass mutations, a lesion node, lesion repair node, MMLD and minimum number of cell divisions was calculated and recorded. We repeated this 10,000 times for each phylogeny and recorded the outcome for all. For all individuals, the majority of independent mutations at the same site were assigned to terminal branches and would not have been included among the PVVs in the dataset. A proportion were assigned to shared branches but did not meet the filtering criteria for a PVV. Notably, very low proportions fell in an orientation consistent with generation by a persistent DNA lesion ('pass' PVVs: median, 0.018; range, 0.002–0.12), with the lowest proportions in those phylogenies with the highest numbers of 'pass' PVVs in the data. We then compared the 'pass' PVV numbers as a proportion of the total detectable PVVs in the data and simulations. To assess the overall degree to which the set of PVVs may be contaminated by those occurring by this mechanism, we calculated a weighted mean of the simulated 'pass' or 'fail' proportions, using the total number of PVVs detected in each phylogeny as weights (Fig. 2g).

(2) Spontaneous somatic reversion of somatic mutation. For a somatic reversion event to be evident in a phylogeny, there must first be a somatic mutation, and subsequently a reversion event within the captured lineages of the mutant clade. The probability of a branch giving rise to a captured somatic reversion is therefore proportional to the product of the branch length and the sum of branch lengths within that clade. Intuitively, this can be thought of in these terms: a long branch has lots of mutations with the potential for reversion, and many subsequent long branches within that clade gives much time and many independent lineages for those mutations to revert. Therefore, we selected branches with probabilities weighted by this product. The reversion branch was then chosen from branches within the selected branch clade, again with probabilities weighted by their branch lengths. Simulation then proceeded analogously to the 'independent mutation' simulation, but with positive samples defined as those within the somatic mutation clade, but not in the reversion clade. We repeated this 2,000 times for each phylogeny and recorded the outcome of each (Fig. 2f). Interestingly, only a minority of somatic reversion events were detected as PVVs (median proportion, 0.173), as they often result in large positive clades with a single negative sample. These have little impact on the inferred $\rho$ value, as such occasional negative samples are not unexpected with binomial sampling of the variant and wild-type alleles as occurs with sequencing of heterozygous sites. However, as expected, almost all those detected were classified as 'pass' PVVs.

## Assessment for LOH as an artefactual cause of PVVs

PVVs may result if a cell containing a somatically acquired mutation loses the mutant allele through say whole chromosome deletion or smaller-scale LOH mechanisms such as mitotic recombination and focal deletion. To exclude this as the cause of the observed PVVs, we used two complementary approaches. First, we applied ASCAT, an allele-specific copy number algorithm[51] to each sample in turn, using a phylogenetically unrelated sample from the same individual as the matched normal. For each PVV, we defined the negative subclades and determined the minor allele copy number at the mutant locus for each sample within that clade. The mean value across negative subclade samples was rounded to the nearest integer: a value of 1 was classed as 'no LOH' (heterozygosity is maintained) and 0 was classed as 'LOH'. A small proportion of PVVs (10 out of 426, 2.5%) were shown to be caused by LOH (Fig. 2e), and were excluded from further analysis.

It is also formally possible that short sub-kilobase LOH events were missed by ASCAT. We therefore used a second approach to directly interrogate sequencing reads and confirm that samples in the negative subclade included reads from the chromosome copy with the mutation. To do this, we first phased the mutation with nearby heterozygous SNPs by interrogating the reads from samples with the mutation. This was possible in approximately one third of cases (Extended Data Fig. 5a). We then interrogated colonies from the negative subclade to confirm the presence of reads reporting the SNP allele from the parental chromosome that carried the mutation.

Sequencing of only the wild-type allele by chance is another potential cause of a PVV, particularly if the negative subclade is a single sample with low depth (although a minimum depth of 12× was required in the PVV identification stage). This read-based assessment also excludes this mechanism. Where assessable, the read-based LOH assessment agreed with the result from ASCAT in all but one case (Extended Data Fig. 5b).

## Assessment of incorrect phylogeny structure as an artefactual cause of PVVs

For all PVVs, there is an alternative phylogeny structure with which the PVV would in fact be consistent (Extended Data Fig. 1d). To confirm that PVVs did not simply result from phylogeny inference errors, we counted the mutations in conflict with the structure suggested by the PVV (in most cases this is the same as the MMLD), and confirmed that they were robust. This required ensuring that the mutation calls themselves were not false positives (manual inspection of sequencing alignments) and that the set of colonies reporting the mutations was correctly assigned (Extended Data Fig. 1e). In addition, we manually checked up to five such mutations for a large number of PVVs, confirming that they validated the consensus phylogeny as expected. In all but one case, there was more than one somatic mutation that convincingly confirmed the consensus phylogeny—thus, for these branches, there was considerably more evidence for the original phylogeny than for the alternative phylogeny suggested by the single PVV.

We further looked into all PVVs with <5 mutations confirming the consensus phylogeny, with a total of 19 PVVs meeting this criterion. Eleven out of nineteen were highlighted as potentially low confidence by either the MPBoot bootstrap support, read count bootstrap support or alternative tree-building approaches discussed in the following paragraphs. We visualized the remaining eight mutations, all of which appeared genuine PVVs caused by prolonged DNA lesions rather than any other mechanism. In all eight cases there was more support for the consensus phylogeny than for the alternative phylogeny.

The consensus phylogeny, as relates to each PVV, is defined by the clades within the 'lesion node' and the 'lesion repair or loss node'. If these are robust, then this suggests that the phylogeny is correct and that the PVV is the exception. We therefore assessed the robustness of these nodes using three different approaches: (1) bootstrap support for the nodes using the bootstrap support values from the MPBoot algorithm; (2) bootstrap support for the nodes using bootstrapping of the read count matrices; and (3) the nodes being identical when the phylogenies are reconstructed using alternative algorithms.

**MPBoot bootstrap support for the lesion node and lesion repair node.** The MPBoot algorithm used to construct the phylogenies gives bootstrap support values for each node[52]. Therefore, for 440 of 501 PVVs (those for which these bootstrap support values were available, all within the haematopoietic phylogenies), we examined the bootstrap support values for the lesion node/lesion repair node for each PVV. This showed that for 90% of PVVs both lesion node and lesion repair node had bootstrap support values ≥98%, and for 96% of PVVs, both had support values 80% (Supplementary Fig. 1a). Only 15 out of 440 PVVs (3.4%), across 9 different nodes, had <80% support for either node, these are discussed further below. This demonstrates the high confidence in relevant phylogeny structures.

**Read count bootstrap support for the lesion node and lesion repair node.** An alternative bootstrapping approach is to bootstrap the variant read counts across all samples and mutation sites, as previously described[7,32]. In this approach, we use the partially filtered mutation set and bootstrap the sequencing read counts for each colony at each locus before subjecting this raw read count data to the same filtering and phylogeny-building algorithms as the original data, with 200–250 replicates per individual. This is computationally intensive and therefore was applied to only 9 blood phylogenies, which accounted for 202 out of 501 PVVs. Again, this suggested that 93% of PVVs had 100% bootstrap support for both lesion node and lesion repair node, and only 7 out of 202 assessed PVVs (3.5%) had less than 80% bootstrap support for either node, discussed further below (Supplementary Fig. 1b,c). All but one of these had also been identified as low confidence by the MPboot bootstrap support measure.

**Support for the lesion node and lesion repair node from alternative phylogeny algorithms.** Another way to assess the robustness is to compare the phylogeny generated by MPBoot to other phylogeny-reconstruction approaches to see if phylogenies are generated with identical lesion node and lesion repair node clades. Overall, the three algorithms gave very high measures of concordance—for example, Robinson–Foulds similarities for MPBoot versus IQ-tree and MPBoot versus SCITE were almost all >0.95; quartet similarities were almost all >0.99 for the two pairwise comparisons (Table S4).

For 3 of the blood datasets accounting for 439 out of 501 PVVs we compared the clades defined by the lesion nodes and lesion repair nodes in phylogenies generated by IQ-tree[53], and where feasible, SCITE[54]. The output for the SCITE algorithm frequently generated phylogenies with polytomies at the site of PVVs, causing fairly frequent discrepancies in the nodes despite the presence of the underlying inconsistent genotypes. Therefore, for the remainder of the analysis we focussed on the IQ-tree phylogenies. For 93% (409/439) of PVVs, the lesion node and lesion repair node were identical. For the remainder, we re-ran the PVV detection algorithm using the trees generated by the IQ-tree algorithm, and confirmed that although the clades had subtle differences (sometimes due to removal of a low-coverage sample), 18 out of 30 of the remaining PVVs were still called. The remaining 12 mutations affected 8 different nodes, and again 8 of the 12 overlapped with those identified as low confidence by the MPBoot bootstrap confidence approach.

**PVVs highlighted as possible low confidence.** Altogether, 19 out of 440 (4.3%) blood PVVs were identified as potentially low confidence by one or more of the above approaches. We manually inspected each of these to understand which of these were incorrectly called, and the probable underlying reasons. Several patterns emerged:

Scenario 1. Owing to the PVVs, there is a degree of equipoise as to the true underlying phylogeny; therefore an alternate configuration for the phylogeny may be produced from bootstrapping or alternative phylogeny reconstruction. However, there remains no single phylogeny that fits all mutations, and one or more PVVs is always present. In this scenario, there may be question marks over which mutations follow the true phylogeny, and which represent PVVs generated by persistent lesions. In several cases, the original MPBoot phylogeny was still best supported (Supplementary Fig. 2a,b). However, in one example, highlighted by all three approaches, the original phylogeny was supported by three mutations, but the alternate phylogeny was supported by four mutations (Supplementary Fig. 2c). This made the alternate phylogeny more likely. In addition, the mutational profile of the three mutations that became PVVs according to the alternate phylogeny were more in keeping with the typical PVV profile, as all were C>T at CpT sites. This suggested that the IQ-tree was the correct configuration. We therefore removed the original four PVVs and replaced them with the three from the IQ-tree analysis, and updated the consensus tree structure accordingly.

Scenario 2. An alternative phylogeny is generated which highlights a potential issue with the original phylogeny, and that there is in fact unlikely to be a prolonged DNA lesion. The reasons for the incorrect original phylogeny include:
(1) Low sample coverage in one or more samples leading to incorrect phylogeny building (Supplementary Fig. 2d,e).
(2) Inappropriate inclusion of a mixed colony in the MPBoot phylogeny which therefore cannot be correctly placed on the phylogeny (Supplementary Fig. 2f).
(3) Probable independent acquisition of the same mutation leading to an incorrect original phylogeny (Supplementary Fig. 2g,h). In one case (the chemotherapy phylogeny), this mutation is a driver mutation in *PPM1D*, showing an example of convergent evolution (Supplementary Fig. 2h).

Scenario 3. The reason for the low bootstrap support from MPBoot is unclear and the PVV appears correctly identified on manual inspection (Supplementary Fig. 3a–d).

As a result of this detailed interrogation of 440 out of 501 PVVs, a total of 10 PVVs were removed, and 3 added to the final mutation set. Overall, this demonstrates that, within the limits of the analysis, the vast majority of PVVs (98%) are robust. However, not all alternative mechanisms can be excluded for each individual PVV.

**Coverage, spectrum and distribution of branch-creating mutations.** As final checks, we assessed the coverage, spectrum and genomic distribution of the mutations that created branches carrying PVVs, compared these patterns to other mutations on the phylogenetic tree. The reason for these checks was the possibility that the branch-creating mutations may represent low-coverage or low-confidence mutations or other mutation-calling artefacts that would falsely affect the tree-building algorithms.

We found that the ~8,000 branch-creating mutations had an almost identical mean coverage to the mutation set as a whole. The mutational spectrum of the 8,000 branch-creating mutations was the same as the full mutation set, suggesting that they were not a distinct set of artefacts, which tend to have specific mutational profiles. Finally, we compared various genomic features of these branch-creating mutations to the overall mutation set. Although for 2/37 there was a suggestion of a weak association (distance to ALU repeat region, *q* value = 0.03; distance to z-DNA motif, *q* value = 0.06), there was no obvious relationship visually. Overall, we believe the vast majority of

the branch-creating mutations to be genuine and the trees therefore robust.

## Inference of expected MAV mutational signatures

We aimed to identify the most likely mutagenic processes to cause the MAVs observed in the bronchial epithelium and PX001 (chemotherapy-exposed HSPC) phylogenies. We started with the SNV mutational signatures that were inferred as present in each tissue from the original studies[5–12] (Extended Data Fig. 6b–d, left). For the bronchial epithelium and liver, these signatures were previously extracted using a Bayesian hierarchical Dirichlet process as implemented in R package HDP (https://github.com/nicolaroberts/hdp)[55]. We used the same package to extract mutational signatures from phylogeny PX001, the chemotherapy-exposed case, using mutations on individual branches as single samples. For each signature, an expected MAV signature could then be inferred as proportional to the product of the context-specific relative likelihoods of the two SNVs in each MAV, weighted by the abundance of that context in the human genome. Accordingly, we calculated the expected MAV signatures resulting from each extracted mutational signature in bronchial epithelium, liver and blood (Extended Data Fig. 6b–d, right) and compared this to the observed MAVs.

## Inference of expected PVV mutational signatures

For each potential alternative mechanism that may generate a PVV, a specific, predictable mutational signature is expected. Going through each in turn:

(1) Two independent mutations at the same site. The expected signature reflects the square of the likelihood of a given mutation at a specific trinucleotide context, weighted by the frequency of that context in the human genome. Raw mutational signatures can be converted into likelihood signatures by correcting for the trinucleotide frequency. Owing to the low abundance of CpG sites in the human genome, this predominantly has the effect of demonstrating the high likelihood of C>T mutations at CpG. This likelihood signature is squared to reflect the fact that the same mutation has to occur twice at that site, before being multiplied by the trinucleotide frequencies to get the expected abundance of such events across the genome. For adult HSPC signature, this results in a signature dominated by C>T mutations at CpG sites, particularly A$\underline{C}$G trinucleotides (Extended Data Fig. 5d(i)). This signature is very similar to that observed for the 109 adult HSPC PVVs that are not in an orientation consistent with generation by a persistent DNA lesion, suggesting that these are predominantly caused by this mechanism (cosine similarity, 0.9; Extended Data Fig. 5e). As a further check on the possibility of chance co-occurrence of the same mutation, we assessed the numbers and spectrum of mutations occurring at the same site in different research participants. Across 27 adult haematopoietic phylogenies, with 5,733,980 mutations among them, there were a total of 34,862 that were shared by 2 or more individuals. We found a very consistent rate of sharing of $2.5 \times 10^{-9}$ of all mutation pairs when both individuals had mutational profiles dominated by the blood signature (calculated as number of shared mutations/[total mutations in individual 1 × total mutations in individual 2]). As expected, the mutational spectrum of these shared mutations was dominated by C>T transitions in a CpG context (Extended Data Fig. 5f), remarkably similar to the predicted spectrum of chance co-occurrence of mutations in blood cells (Extended Data Fig. 5d(i)) and different to the spectrum observed for PVVs in the same individual (Fig. 3c).

(2) Spontaneous reversion of a somatic mutation. For a spontaneous reversion to occur, a mutation at a given trinucleotide context must, at a later time point, be followed by the reverse mutation (at the same context)—for example, C>T at A$\underline{C}$G must be followed by a T>C at A$\underline{T}$G. The likelihood of this occurring is proportional to the product of the likelihood of the original mutation and the likelihood of the reversion mutation. The expected signature is therefore

this likelihood, multiplied by the trinucleotide frequencies. For the adult HSPC signature, this reveals a signature with a dominant peak at T>C mutations at A$\underline{T}$G sites, reflecting the high likelihood of the reversion mutation (Extended Data Fig. 5d(ii)).

(3) LOH, biased allele sequencing or incorrect phylogeny. All of these mechanisms of PVV generation are agnostic to the identity of the original mutation. Therefore the expected mutational signature of such PVVs should reflect that of the overall tissue signature (Extended Data Fig. 5d(iii)).

The observed PVV signature (Fig. 3c) is clearly distinct from any of these predicted signatures (cosine similarities: 0.15, 0.3, 0.62 for independent mutations, spontaneous reversion and other mechanisms respectively). This supports the premise that these are caused by a specific mutational process.

## Inference of mean lesion duration

The C>T PVVs in the HSPC phylogenies had a consistent signature and broadly consistent MMLDs. We therefore hypothesized that these PVVs are caused by a single underlying lesion. The MMLD of the C>T PVVs has a median value of 21, corresponding in chronological time to ~1.3–1.5 years. However, this value may not be a good estimate of the true lesion durations for three main reasons:

(1) For each PVV, the MMLD is a minimum of the true lesion duration.
(2) Lesions captured as PVVs may represent the longest durations from the underlying distribution of lesion durations.
(3) The duration of detected PVVs is dependent on the phylogeny structure. For example, if there was only one branching region of the phylogeny suitable for capturing PVVs, any detected PVVs would have the same MMLD, defined by the phylogeny.

We designed an ABC model to account for these factors and estimate the true mean duration of the underlying lesion. This used only the four phylogenies from the older individuals from the normal ageing haematopoiesis dataset[5], as these had some of the largest and richest phylogeny structures (and therefore the least potential bias), and had substantial numbers of C>T PVVs per individual (range: 8–73).

**Simulating complete HSC population phylogenies with lesions.**
The simulation framework started with 40 simulated 'elderly' complete haematopoietic stem cell populations using parameters drawn from the posterior distribution of the ABC in our normal haematopoiesis study[5], an HSC population size of 100,000, and an age of 75. These populations reflected the diversity of oligoclonality found in the four elderly phylogenies (Extended Data Fig. 11a). Within each of these populations, lesions were randomly introduced with lesion durations drawn from a gamma distribution with shape = 1, and mean varying from 0.5 years to 5 years, $\mu = \{0.5, 0.6, 0.7 … 5\}$. For each population and lesion duration, sufficient lesions were introduced to ensure that at least 60,000 PVVs were theoretically detectable within the complete phylogenies. For longer lesion times, ~$2 \times 10^6$ lesions were sufficient, but for short lesion times ($\mu = 0.5$ years), introduction of >$3 \times 10^7$ lesions was sometimes necessary. These 40 complete aged HSC populations, each with 47 different sets of 60,000 potentially detectable PVVs caused by lesions of varying duration, formed the starting point for the subsequent simulations.

**Simulating the data.** We then performed simulations to mirror the data. For each simulation run, 4 of the aged HSC populations were randomly selected to represent the four elderly phylogenies in the data, and a single mean lesion duration was chosen from $\mu = \{0.5, 0.6, 0.7 … 5\}$. Each selected population was then randomly down-sampled to the size of the actual phylogenies (328 tips for KX003, 922 tips for KX004, 315 tips for KX007 and 367 tips for KX008). We then iterated in random order through the potentially detectable lesions present in the complete phylogeny (generated from the simulations of persistent lesions with the chosen value for $\mu$) to see if they remained detectable in the down-sampled phylogeny. Typically, only ~1 in 1000 remained

detectable. Once adequate lesions were detected to represent those found in the data (33 for KX003, 80 for KX004, 9 for KX007 and 22 for KX008), the simulation stopped. In 41% of cases, despite assessing all 60,000 PVVs in each phylogeny, the total number detectable remained somewhat fewer than the data, though >99% had at least 100 of the targeted 129 PVVs. Each down-sampled tree was then rescaled to molecular time using the get_elapsed_time_tree function from the rsimpop package, and the MMLD of each captured PVV recorded. As with the data, any lesion with MMLD > 200 was removed from the set. Finally, a gamma generalized linear regression model was used to estimate mean and dispersion parameters on the combined MMLD set, using the glm function from the R package 'stats'. The dispersion and mean of the MMLD distribution clearly varied according to the mean duration of the underlying distribution (Extended Data Fig. 11b,c), although the parameters plateaued somewhat once the mean lesion duration reached 3–4 years.

**Assessing the performance of the ABC framework.** We next assessed the performance of the new framework in recovering the true lesion durations from simulated data. We therefore simulated 4 additional elderly phylogenies, and again introduced simulated lesions of varying duration drawn from a gamma distribution of varying mean, $\mu = \{0.5, 0.6, 0.7 \ldots 5\}$ and shape = 1. As before, we then down-sampled the phylogenies to the size of those from the data and determined if each PVV from the full phylogeny remained detectable, and its MMLD. For each mean lesion duration, we sampled MMLDs from the complete set such that the total numbers were the same as the data. Finally, for each value of $\mu$ we performed the parameter inference using the the *abc* function from the 'abc' package (https://doi.org/10.32614/CRAN.package.abc), using a tolerance of 0.05 and the 'rejection' method, meaning that no regression step was performed. This showed that the simulations and inferences performed broadly well across the range of mean lesion durations (Extended Data Fig. 11d,e), with the 'ground truth' mean lesion duration falling within the 95% posterior intervals of the inferred values in 45 out of 46 cases.

**Performing the parameter inference.** Having established that the framework performed as expected on simulated data, we ran the ABC inference on the observed data, again using the abc function from the abc package and the same settings. This produced the approximate posterior distribution of the true mean lesion duration (Fig. 5a).

**Posterior predictive checks.** We performed posterior predictive checks, drawing values of the mean lesion duration parameter $\mu$ from the posterior distribution and running additional simulations. In this way, we performed 230 additional simulations. The summary statistics from these produced distributions of summary statistics, which, in contrast to the summary statistics from the prior (Extended Data Fig. 11f), closely matched the data (Extended Data Fig. 11g).

**Inferring the average number of lesions per cell at any time.** In our ABC framework, we recorded how many lesions needed to be introduced in order to generate sufficient PVVs to match the data. We also know the overall amount of 'lineage time' of the complete HSC population into which lesions are simulated, which is the sum of all the edge lengths (scaled to days). Using this data, and the distribution of lesion durations, we can readily infer the average number of lesions expected in any given cell at any given moment in time. We therefore created a posterior distribution for this parameter, using the parameters from the accepted set of simulations from the abc step (Extended Data Fig. 11h). One limitation is that the sensitivity for PVVs varies by phylogeny clonal structure, and this was not accounted for in the model: lesions were simply added to a phylogeny until sufficient PVVs were detected to match the data. Therefore, for a simulated 'insensitive' clonal structure the implied lesion density may be artificially high if it were trying to generate the same number of PVVs as a more sensitive clonal structure from the data. However, it at least gives a broad sense of the range in which the true value may sit.

**Detection of lesion segregation**

We used the 'calculate_lesion_segregation' function from the R package MutationalPatterns v3.01 (https://bioconductor.org/packages/MutationalPatterns)[56] to assess for lesion segregation in each individual branch from each phylogeny. As reported in the DEN-exposed mouse model study[15], we used the binomial, Wald-Waldowitz and $rl_{20}$ tests—branches were considered positive if the $rl_{20}$ was ≥6 and at least one of the binomial or Wald-Waldowitz tests had a Benjamin–Hochberg-adjusted $P < 0.05$.

**Reporting summary**

Further information on research design is available in the Nature Portfolio Reporting Summary linked to this article.

## Data availability

Raw WGS data are available via managed-access through the European Genome-Phenome Archive (https://ega-archive.org/) as follows: (1) healthy haematopoietic colonies (WGS accession numbers EGAD00001007684 and EGAD00001007851); (2) myeloproliferative neoplasm colonies (WGS accession number EGAD00001007714); (3) fetal haematopoiesis colonies (WGS accession number EGAD00001006162); (4) allogeneic stem cell transplant donor and recipient colonies (WGS accession number EGAD00001010872); (5) clonal haematopoiesis colonies (WGS accession number EGAD00001007684); (6) colonies from patients exposed to chemotherapy (WGS accession number EGAD00001015339); (7) bronchial organoids (WGS accession number EGAD00001005193); and (8) liver laser-capture microdissections (WGS accession number EGAD00001006255).

## Code availability

All analyses reported in this study used the statistical software R (v.4.4.0). All code and most derived data necessary for reproducing the analysis are publicly available at https://github.com/mspencerchapman/Prolonged_persistence_of_DNA_lesions. Other larger derived data files are available on Mendeley Data at https://doi.org/10.17632/9tw3kbj2cw.1.

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

**Acknowledgements** Much of the sequencing in blood cells underpinning this work was supported by the WBH Foundation. Investigators at the Sanger Institute are supported by a core grant from the Wellcome Trust. M.S.C. has been supported by a Clinical PhD fellowship from the Wellcome Trust. Work in the A.R.G. laboratory is supported by the Wellcome Trust, Bloodwise, Cancer Research UK, the Kay Kendall Leukaemia Fund, and the Leukemia and

Lymphoma Society of America. The A.R.G. laboratory is supported by a core support grant from the Wellcome Trust and Medical Research Council to the Cambridge Stem Cell Institute. Work in the E.L. laboratory is supported by a Wellcome Trust Sir Henry Dale Fellowship, BBSRC and a European Haematology Association Non-Clinical Advanced Research Fellowship. S.M.J. and P.J.C. are supported by Cancer Research UK for their work in bronchial epithelium. M.H. is supported by Cancer Research UK for the work in liver.

**Author contributions** M.S.C. and P.J.C. conceived the study, with input from I.M., M.R.S. and T.H.H.C. P.J.C. supervised the work. M.S.C. developed the bioinformatic algorithms used to detect the persistent lesions and performed all other downstream analyses. L.D.K. assisted with the analysis of associations of persistent lesion genomic loci with DNA methylation features. M.S.C., E.M., K.Y., N.W., M.A.F., A.M.R., P.S.R., M.W., S.B., K.M., K.S.P., K.H.C.G., S.M.J., S.W.K.N., M.H., A.R.G., G.S.V., A.C., M.G.M., E.L., I.M., M.R.S., J.N., T.H.H.C. and P.J.C. all contributed to sample collection, laboratory manipulation, sequencing and/or analysis of samples used for whole-genome sequencing and generation of the phylogenies.

**Competing interests** P.J.C., M.R.S. and I.M. are co-founders, stock holders and consultants for Quotient Therapeutics Ltd. M.H. is a consultant for Quotient Therapeutics Ltd, AstraZeneca and Boston Scientific and has received unrestricted scientific grants from Pfizer. M.A.F. is a current employee and stockholder of AstraZeneca. A.M.R. is a senior editor at *Nature Medicine*, a Nature Portfolio journal. A.M.R. was not involved in the editorial handling of this *Nature* paper (journals within the Nature Portfolio are editorially independent). All other authors declare no competing interests.

**Additional information**
**Correspondence and requests for materials** should be addressed to Peter J. Campbell.

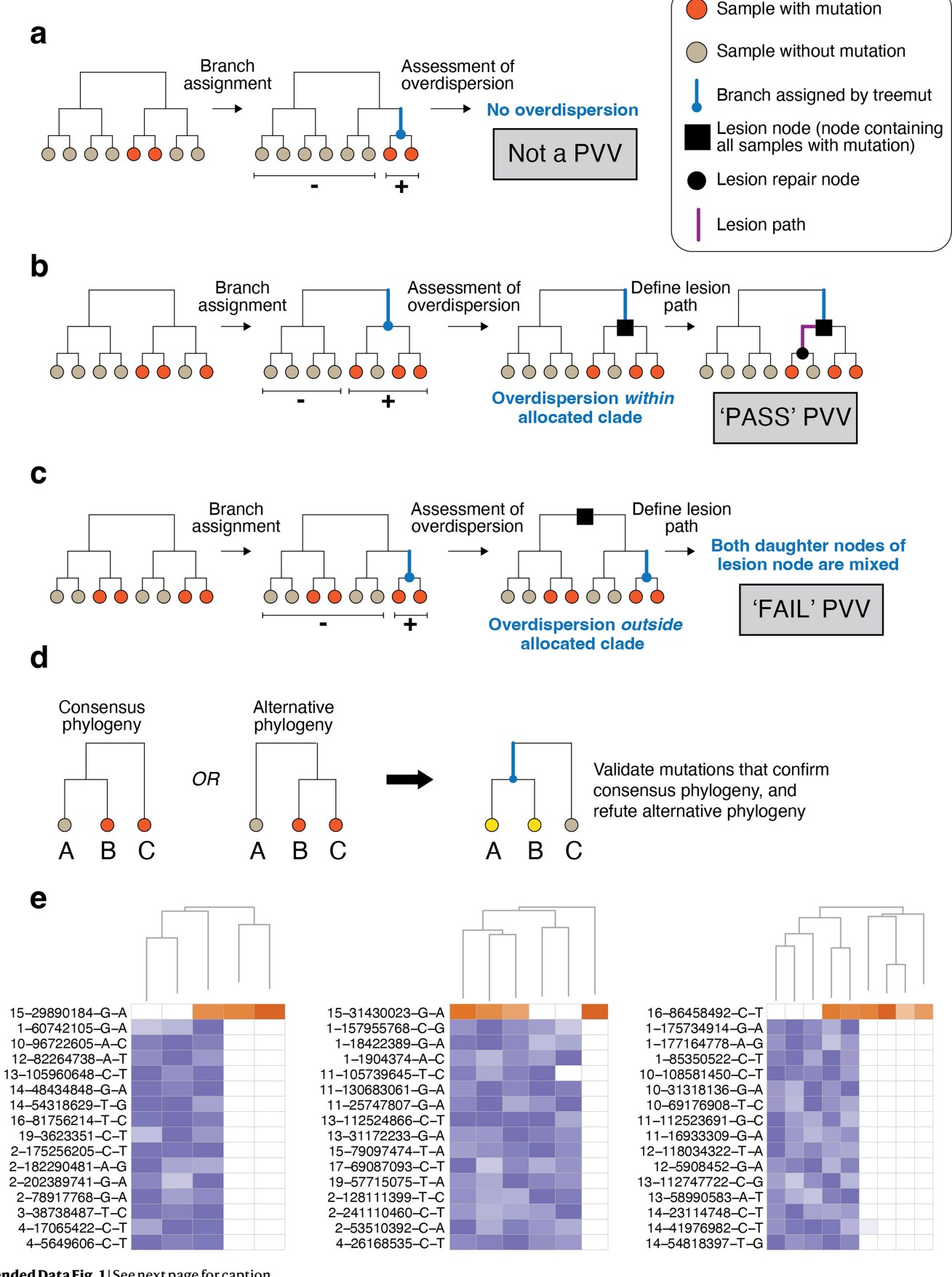

**Extended Data Fig. 1** | See next page for caption.

**Extended Data Fig. 1 | Detection of phylogeny-violating variants (PVVs).**
**a**, A mutation that fits the consensus phylogeny will have no overdispersion within the positive clade, or outside the positive clade. **b**, In this example, treemut() attempts to assign a branch for the mutation, even though there is no single branch assignment that fits the data. There is therefore a negative sample within the clade that should be positive with a single mutation. This orientation is consistent with a persistent DNA lesion, and a single 'lesion path' can be defined with a 'lesion node' and 'lesion repair node' assigned. This example would be classified as a 'PASS PVV'. **c**, An example of a variant where no single branch assignment fits the data, but in this example the significant overdispersion is detected outside the allocated clade. However, this variant cannot be explained by a persistent DNA lesion because both daughter nodes of the lesion node show a mix of mutant and wild-type descendants – this lesion is therefore classed as a 'FAIL' PVV. **d**, For any given PVV, there is a reordering of the branches from the consensus phylogeny such that the PVV no longer contradicts the tree structure (alternative phylogeny). We can then assess the distribution of reads reporting the mutations that distinguish the two phylogenies across the descendant colonies for whether they support the consensus or alternative phylogeny. **e**, Three examples of PVVs showing the validation of the consensus phylogeny inferred for that patient. For each, a zoomed-in subsection of the tree is shown above the heatmap; the PVV is shown in the red colour scale; and the mutations confirming the consensus phylogeny are shown in the blue colour scale. The colour scales of the heatmap denote the variant allele fraction, with white as VAF = 0. The mutations are annotated as 'chromosome-position-reference-variant'.

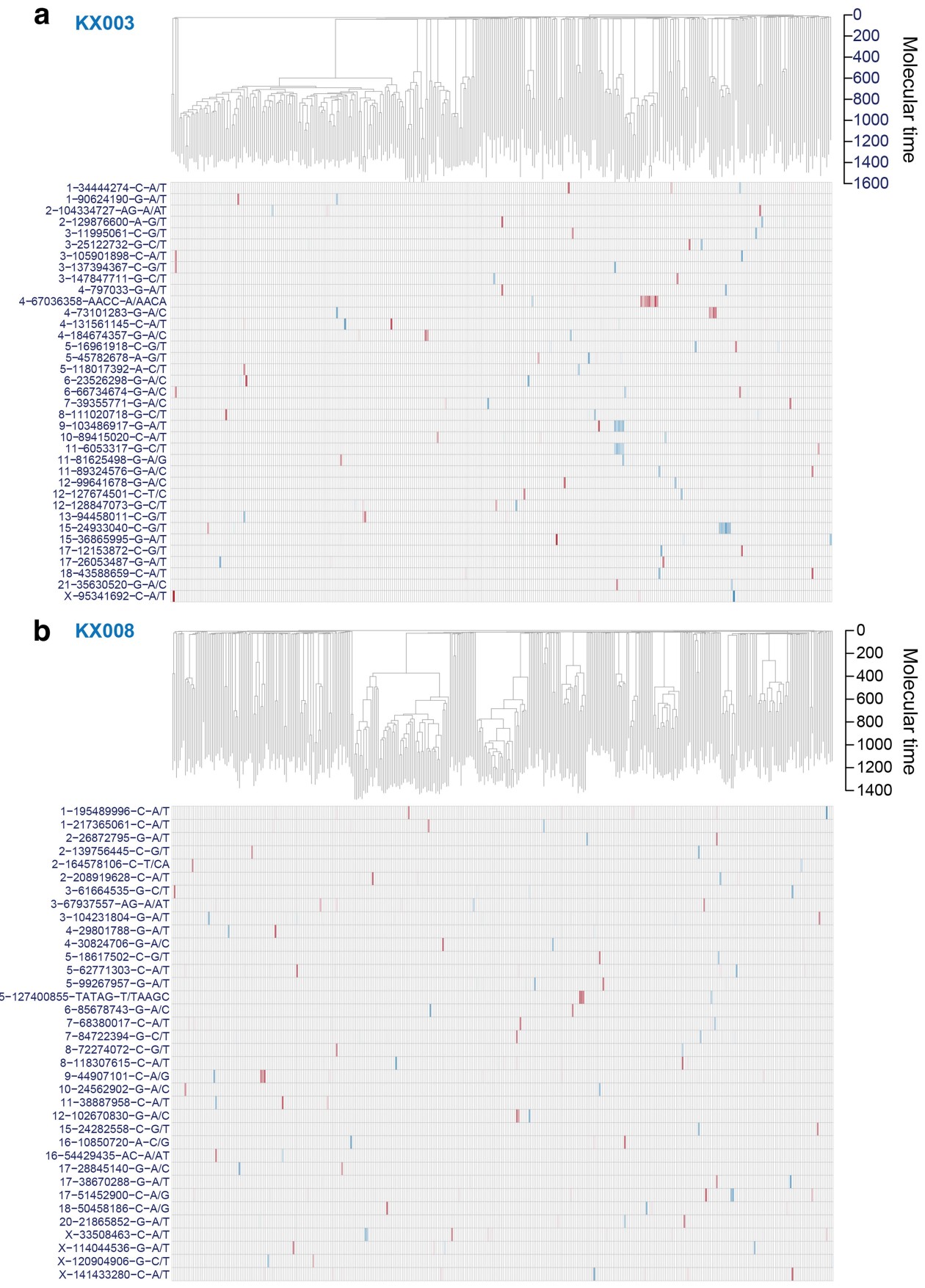

**a** KX003

**b** KX008

**Extended Data Fig. 2** | See next page for caption.

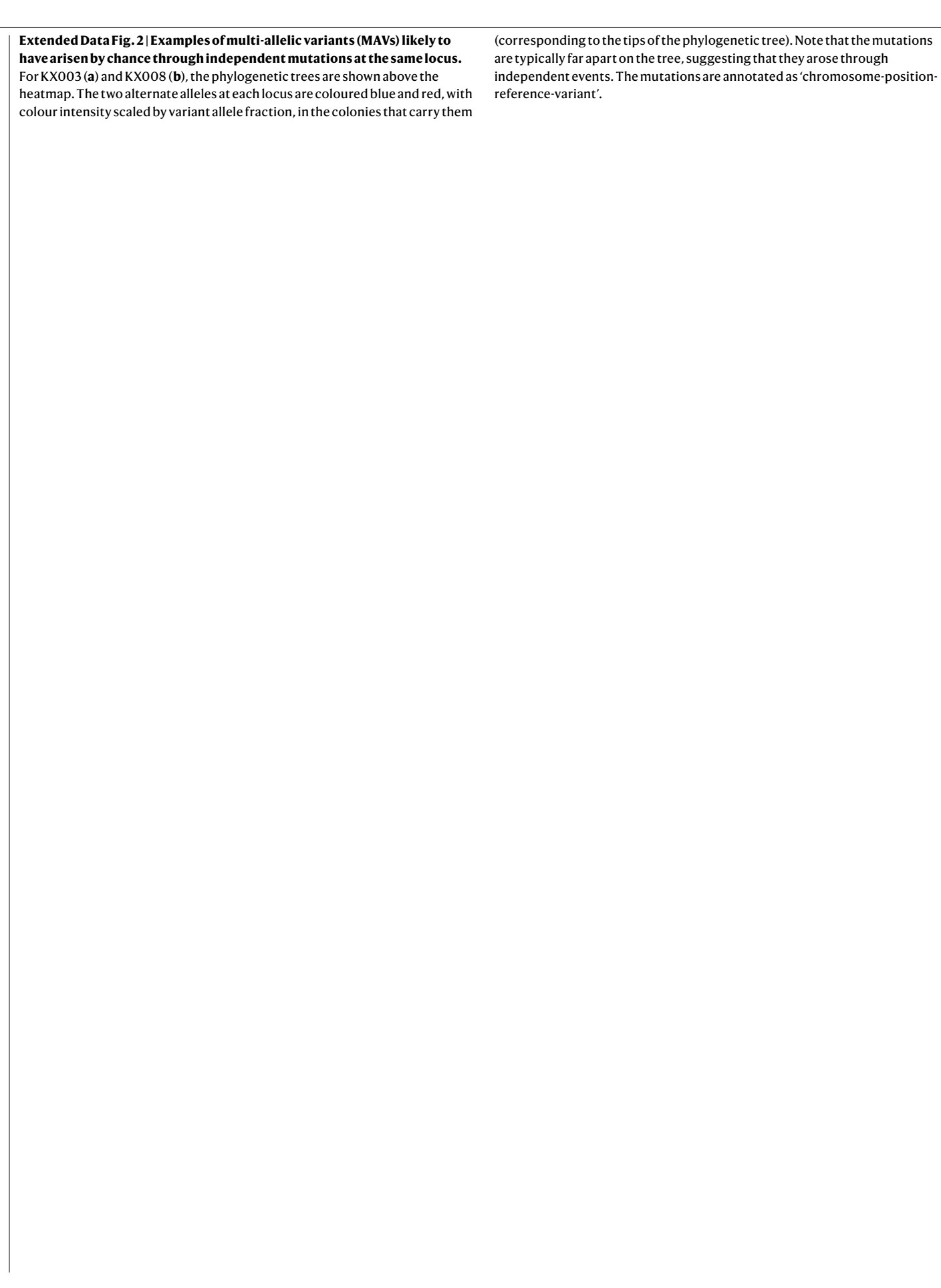

**Extended Data Fig. 2 | Examples of multi-allelic variants (MAVs) likely to have arisen by chance through independent mutations at the same locus.** For KX003 (**a**) and KX008 (**b**), the phylogenetic trees are shown above the heatmap. The two alternate alleles at each locus are coloured blue and red, with colour intensity scaled by variant allele fraction, in the colonies that carry them (corresponding to the tips of the phylogenetic tree). Note that the mutations are typically far apart on the tree, suggesting that they arose through independent events. The mutations are annotated as 'chromosome-position-reference-variant'.

**a**

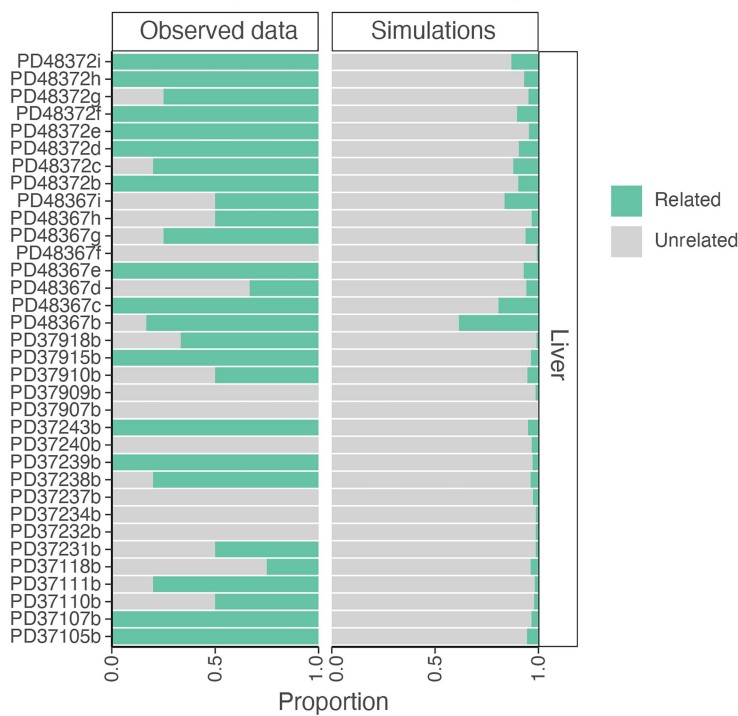

Simulated independent MAVs in liver samples

**b**

Example of phased MAV and germline heterozygous SNP

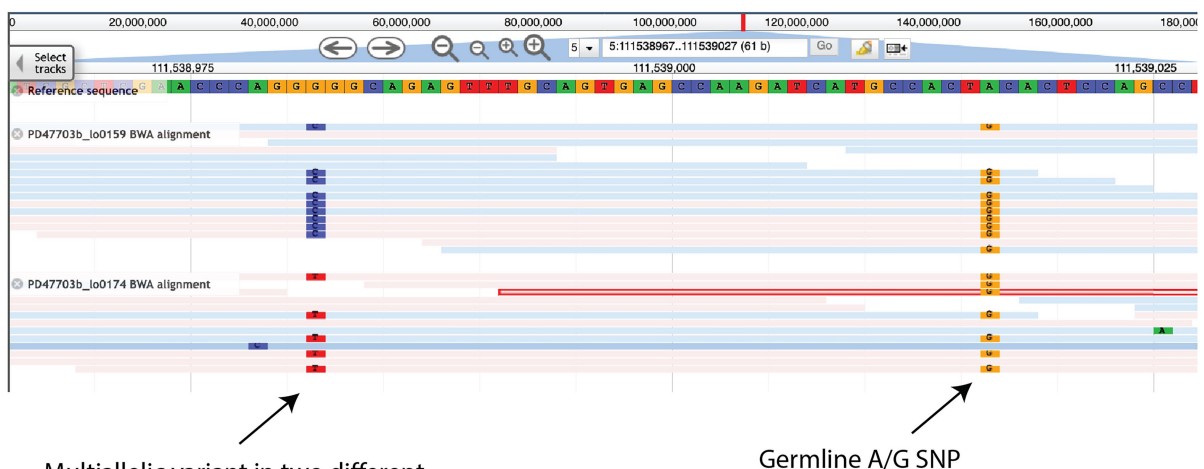

Multiallelic variant in two different,
closely related colonies

Germline A/G SNP

**c**

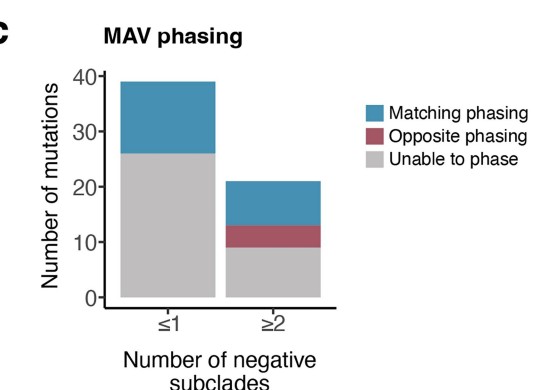

**Extended Data Fig. 3** | See next page for caption.

**Extended Data Fig. 3 | Validation of multi-allelic variants (MAVs). a**, Bar plot showing a comparison of the proportion of MAVs occurring in 'related' or 'unrelated' orientations for each liver phylogeny, and in the simulated null model of occurrence by independent mutation acquisition. **b**, A JBrowse plot showing an example of a correctly phased MAV with heterozygous germline SNP. Alignments of individual reads to the relevant section of the human genome from two different colonies that are closely related on the phylogenetic tree are shown, with pink reads denoting those mapped to the forward strand; blue reads to the reverse strand. Base calls that do not match the reference genome are shown as coloured rectangles. **c**, Phasing comparison results of 'separated' MAVs with 1 or fewer intervening negative subclades, versus those with 2 or more intervening negative subclades. In the latter group the ratio of matching to non-matching phasing is somewhat more balanced, and therefore these MAVs were excluded from downstream analysis. F, Foetal HSPCs; C, Chemotherapy-exposed HSPCs.

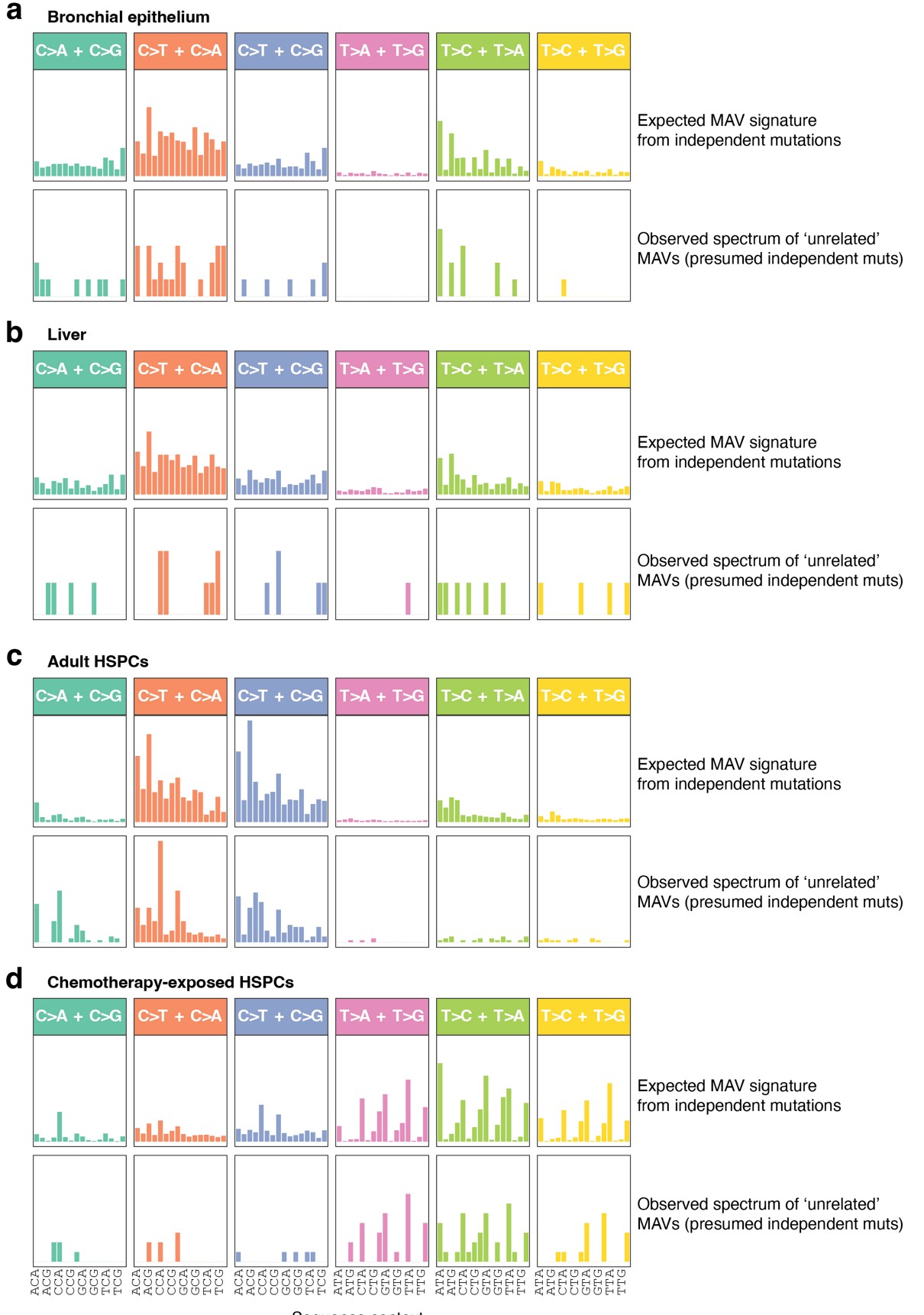

**Extended Data Fig. 4 | Expected versus observed spectrum of independent MAVs. a**, Barplot showing the expected signature of MAVs for bronchial epithelium for two mutations occurring at the same locus independently by chance (top) and the observed spectrum for 'unrelated' MAVs (those that had an orientation on the phylogeny that was incompatible with arising from a single lesion). **b**, As for **a**, but for MAVs in the liver samples. **c**, As for **a**, but for MAVs in the adult HSPCs. **d**, As for **a**, but for MAVs in the chemotherapy-exposed HSPCs.

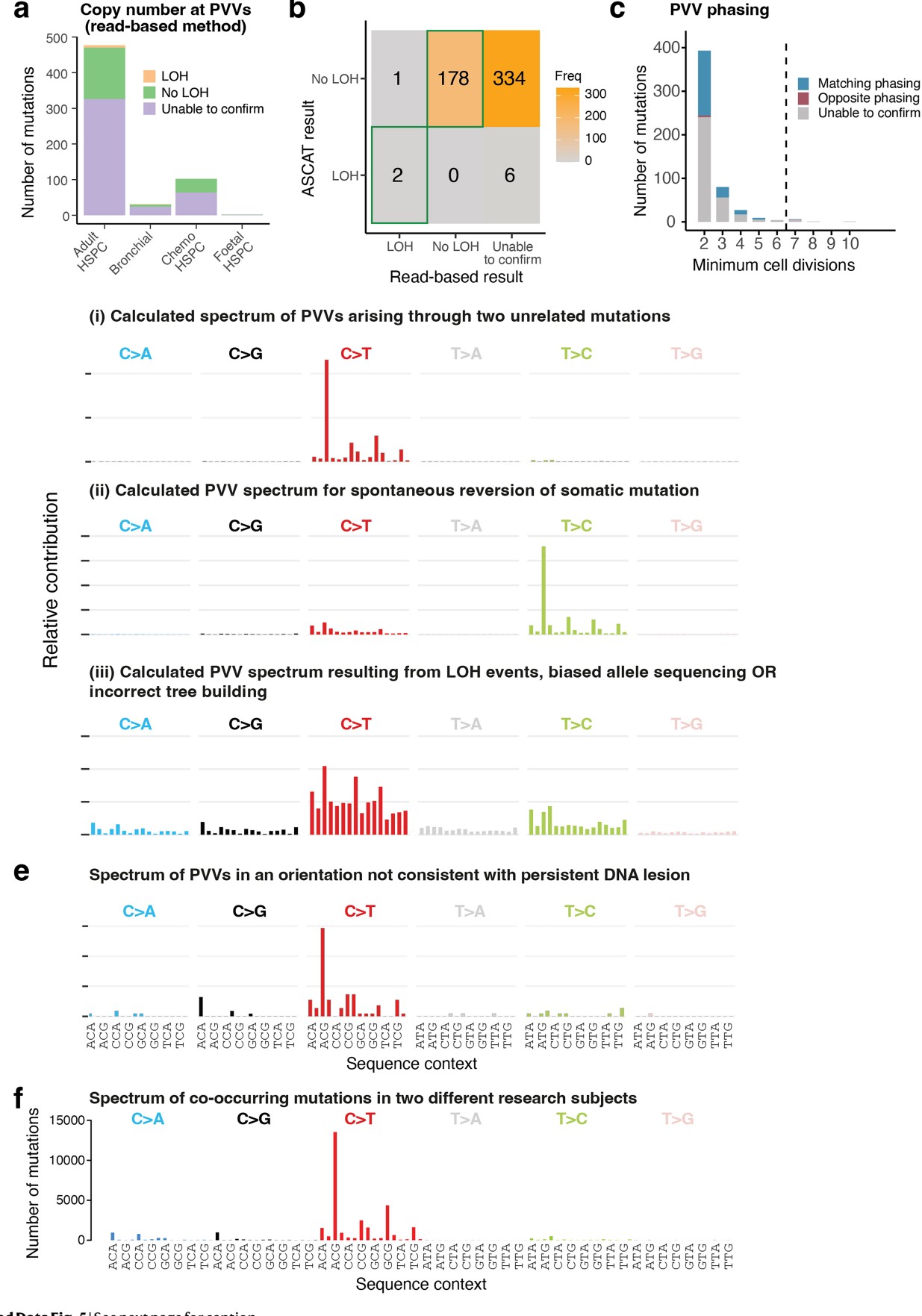

**a** Copy number at PVVs (read-based method)

**b**

**c** PVV phasing

**(i) Calculated spectrum of PVVs arising through two unrelated mutations**

**(ii) Calculated PVV spectrum for spontaneous reversion of somatic mutation**

**(iii) Calculated PVV spectrum resulting from LOH events, biased allele sequencing OR incorrect tree building**

**e** Spectrum of PVVs in an orientation not consistent with persistent DNA lesion

**f** Spectrum of co-occurring mutations in two different research subjects

**Extended Data Fig. 5** | See next page for caption.

**Extended Data Fig. 5 | Validation and spectrum of PVVs. a**, Barplot showing the results of copy number analysis using a reads-based method to ensure small (<1 kb) deletions were not resulting in artefactual PVV calls. **b**, Contingency graph showing comparison of copy number analyses by ASCAT (y axis) versus the reads-based method (x axis). **c**, Stacked barplot showing the minimum number of cell divisions through which the PVV-causing lesion must have persisted unrepaired, coloured by phasing. **d**, Calculated blood PVV mutational signatures expected for those occurring (i) as two independent events, (ii) spontaneous reversion events, or (iii) other alternate mechanisms. **e**, Actual mutational spectrum of 'FAIL' PVVs from the adult HSPC phylogenies, namely variants that were in an orientation inconsistent with a single DNA lesion – note the resemblance to **d**, panel (i). **f**, Mutational spectrum of mutations co-occurring at the same site in two different research subjects.

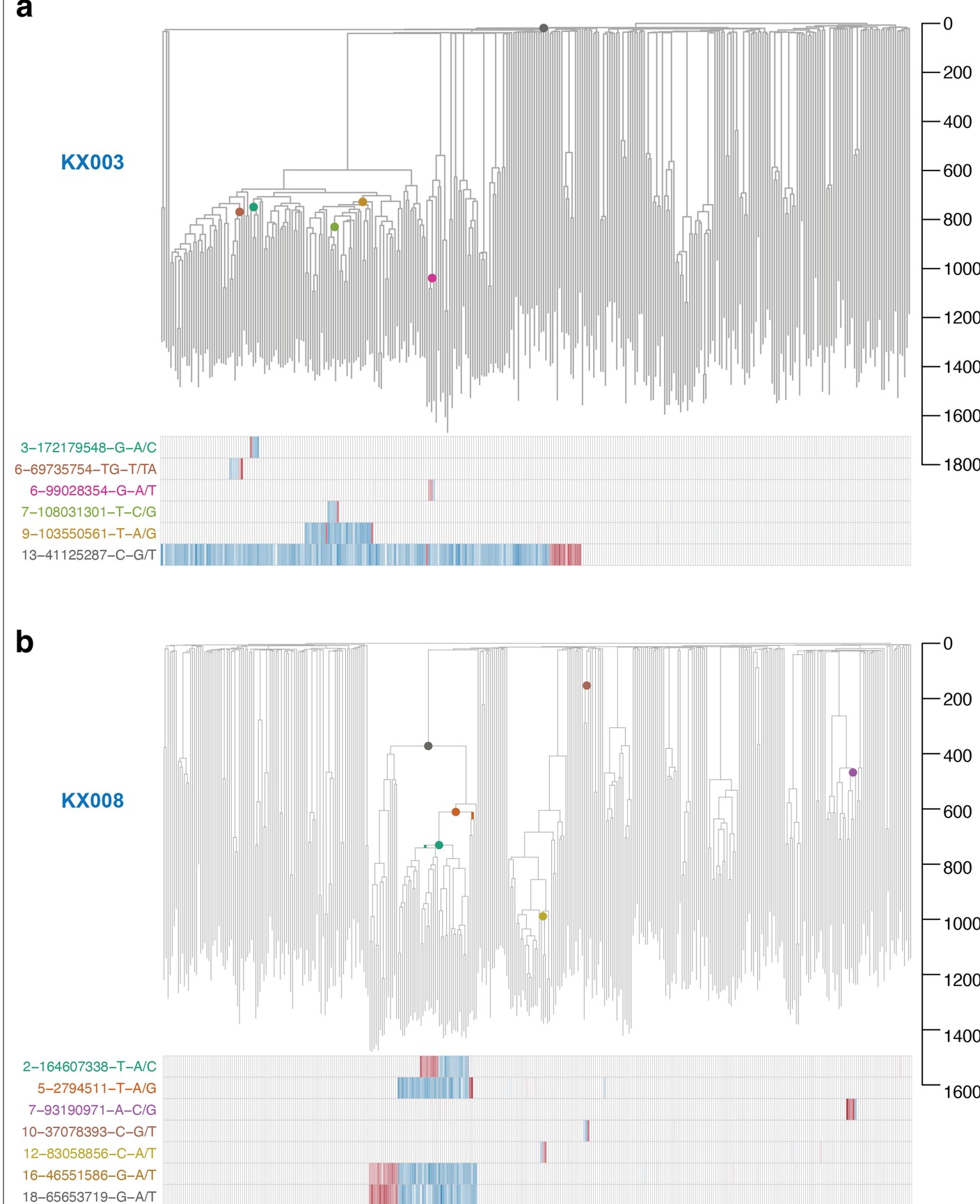

**Extended Data Fig. 6** | See next page for caption.

**Extended Data Fig. 6 | Examples of multi-allelic variants (MAVs) that occurred close together on phylogenetic tree and in an orientation consistent with a persistent DNA lesion.** For KX003 (a) and KX008 (b), the phylogenetic trees are shown above the heatmap. The two alternate alleles at each locus are coloured blue and red in the heatmap, with colour intensity scaled by variant allele fraction, in the colonies that carry them (corresponding to the tips of the phylogenetic tree). The node at which the DNA lesion must have existed is marked on the phylogenetic tree with a coloured circle; colours correspond to the colours of the mutation annotation in the heatmap. For separated MAVs, the minimal lesion persistence path is highlighted in the same colour along the relevant branches. Note that the mutations are typically close together on the tree, suggesting that they arose from the same lesion. The mutations are annotated as 'chromosome-position-reference-variant'.

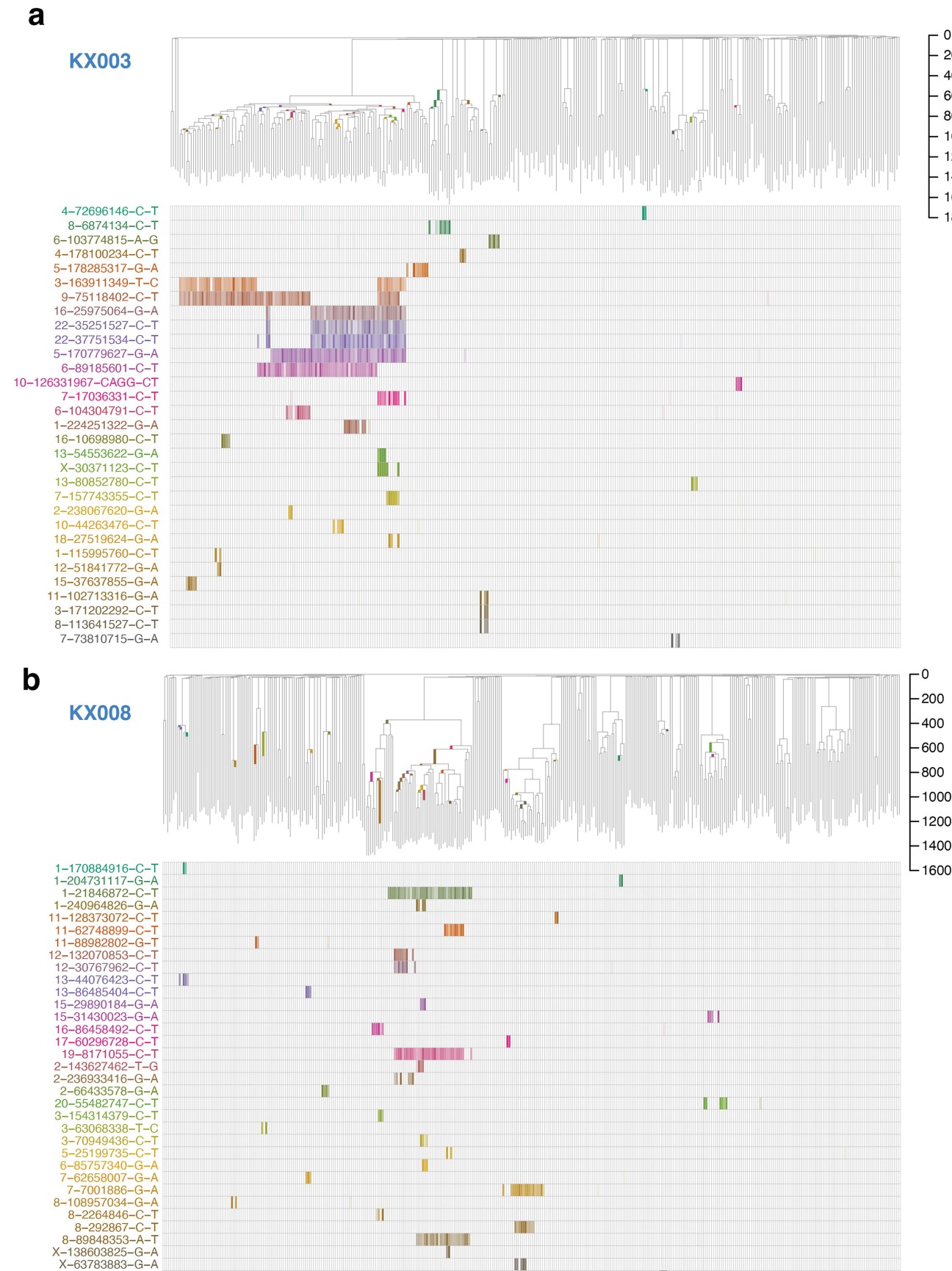

**Extended Data Fig. 7** | See next page for caption.

**Extended Data Fig. 7 | Examples of phylogeny-violating variants (PVVs) that occurred close together on phylogenetic tree and in an orientation consistent with a persistent DNA lesion.** For KX003 (a) and KX008 (b), the phylogenetic trees are shown above the heatmap. The individual PVVs each represent a row of the heatmap, with colour intensity scaled by variant allele fraction in the colonies that carry them (corresponding to the tips of the phylogenetic tree). The minimal lesion persistence path is highlighted in the same colour along the relevant branches of the phylogenetic tree. Note that the mutations are typically close together on the tree, but with an interspersed subclade of wild-type colonies that enables identification as a PVV. The mutations are annotated as 'chromosome-position-reference-variant'.

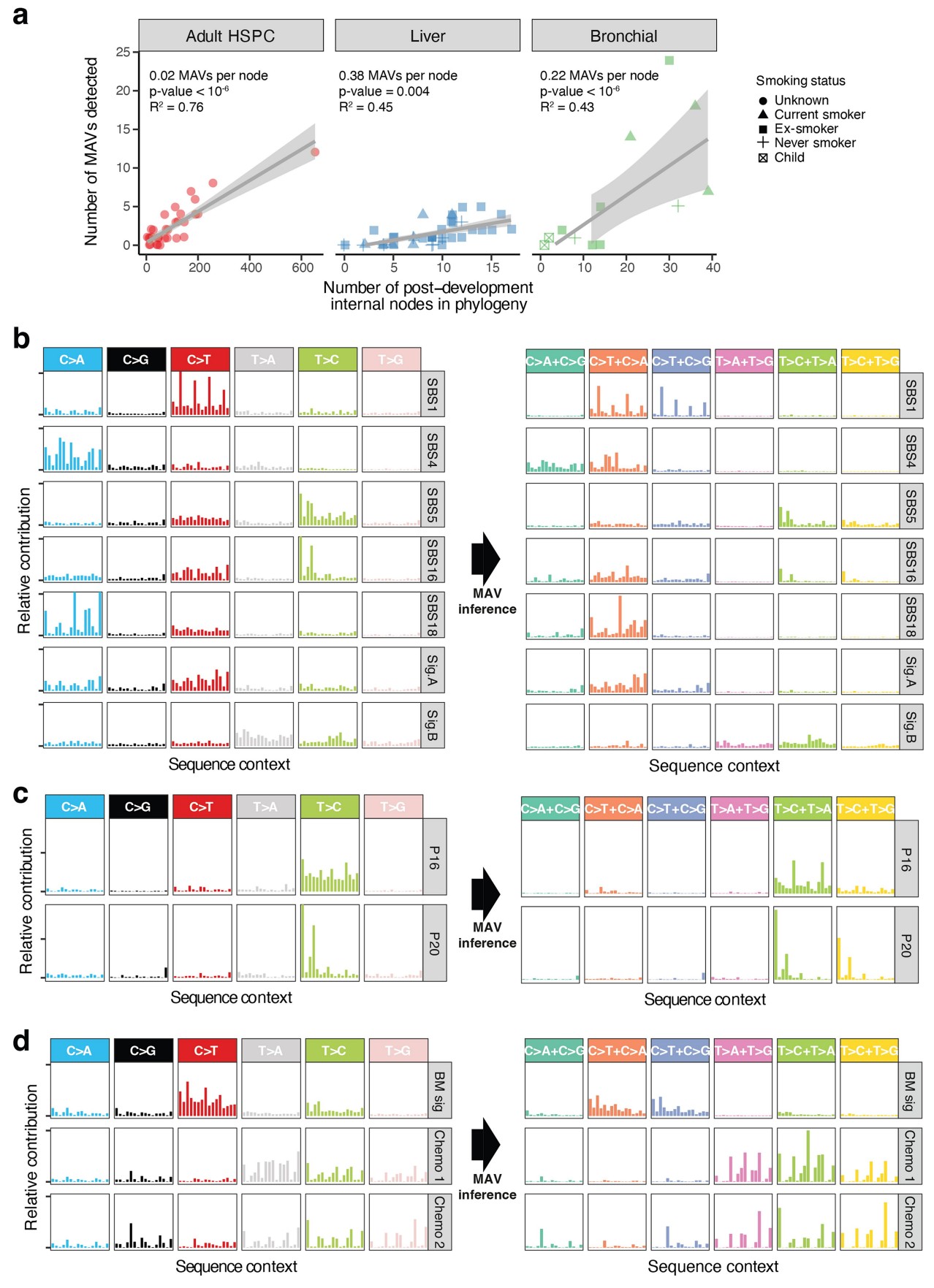

**Extended Data Fig. 8** | See next page for caption.

**Extended Data Fig. 8 | Number and expected spectrum of MAVs. a**, Scatter plot showing the numbers of MAVs in each phylogeny against the total number of post-development nodes within that phylogeny, separated by tissue. Smoking status (where known) is indicated by the shape. Grey lines display the correlation between these values by univariate linear regression, with the shaded area showing the 95% confidence interval for the estimated slope. The p values refer to two-sided likelihood-ratio tests on whether the slope is different to 0, uncorrected for multiple hypothesis testing. **b**, On the lefthand side are the extracted 96-profile SBS signatures from the bronchial epithelium as per Yoshida et al.[10]. On the righthand side is the calculated MAV signature expected for that signature. **c**, As in **b**, but for the liver samples. Only two of the 30 extracted liver signatures[12] are displayed. These are the signatures that most closely match the observed MAV signature. **d**, As in **b**, but for the chemotherapy-exposed blood samples. Note that the 'BM signature' is the same signature extracted for the normal adult blood phylogenies. BM, bone marrow; SBS, single base substitution.

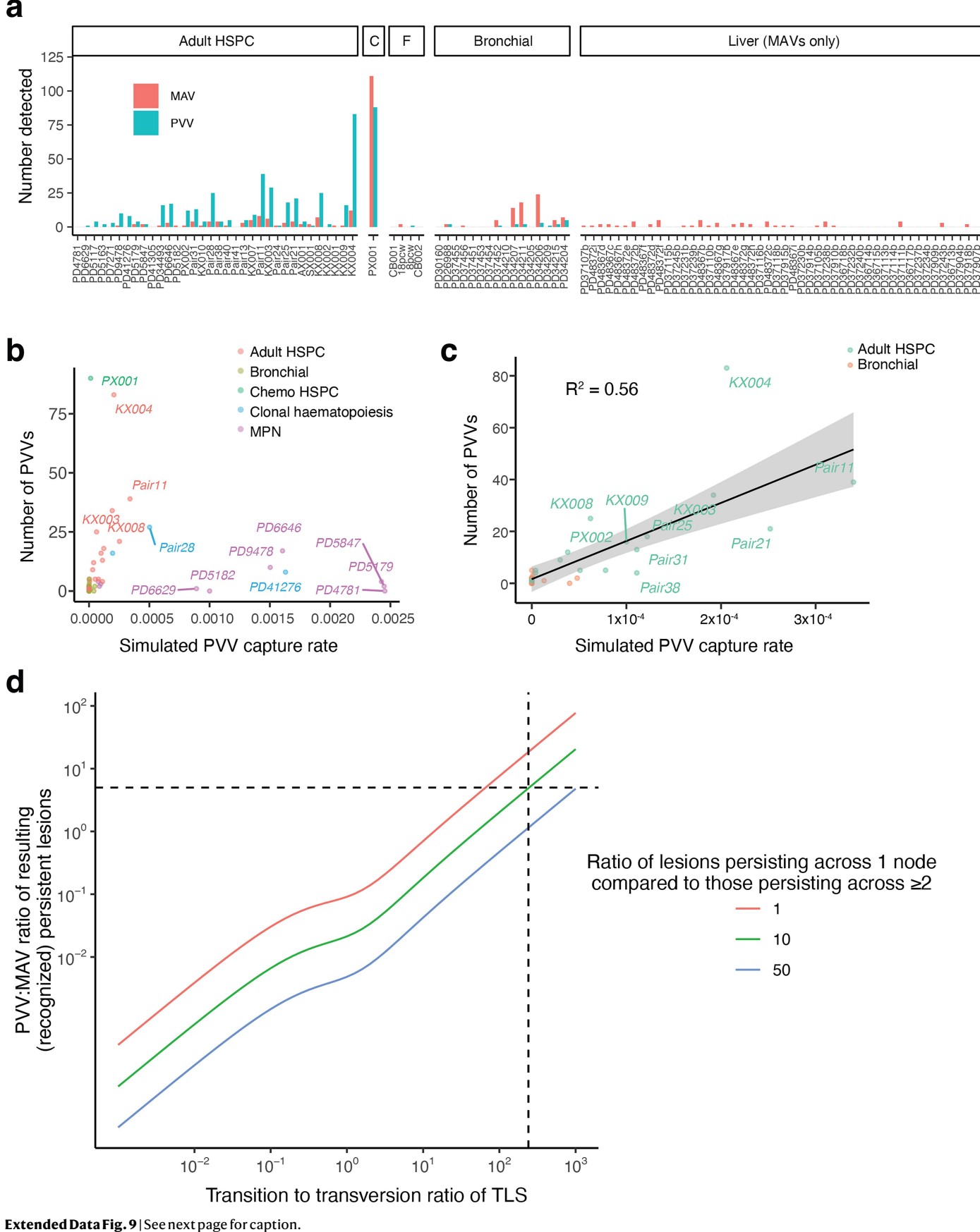

**Extended Data Fig. 9** | See next page for caption.

**Extended Data Fig. 9 | Statistical power to detect PVVs. a**, Barplot showing the number of MAVs and PVVs detected in each phylogeny. Phylogenies are divided by their tissue type, and within each type, phylogenies are ordered by number of samples. The 331 MAVs and 501 PVVs were unevenly distributed across samples. **b**, Relationship between the simulated PVV capture rate and the observed number of PVVs for the whole dataset. Individuals who have a large clonal expansion (MPN, in purple, and those with clonal haematopoiesis, in blue) have lower than expected numbers of PVVs than the simulations would expect. **c**, Scatter plot of the simulated PVV capture rate and observed numbers of PVVs, excluding samples with a single large clonal expansion or chemotherapy exposure. The black line displays the correlation between these values by univariate linear regression, with the shaded area showing the 95% confidence interval of the estimated slope. **d**, Simulated ratio of PVV:MAVs depending on the specificity for transition mutations during translesion synthesis (such as correctly incorporating a pyrimidine opposite a damaged guanine), rather than transversion mutations. TLS = translesion synthesis; PVV = phylogeny-violating variant; MAV = multi-allelic variant.

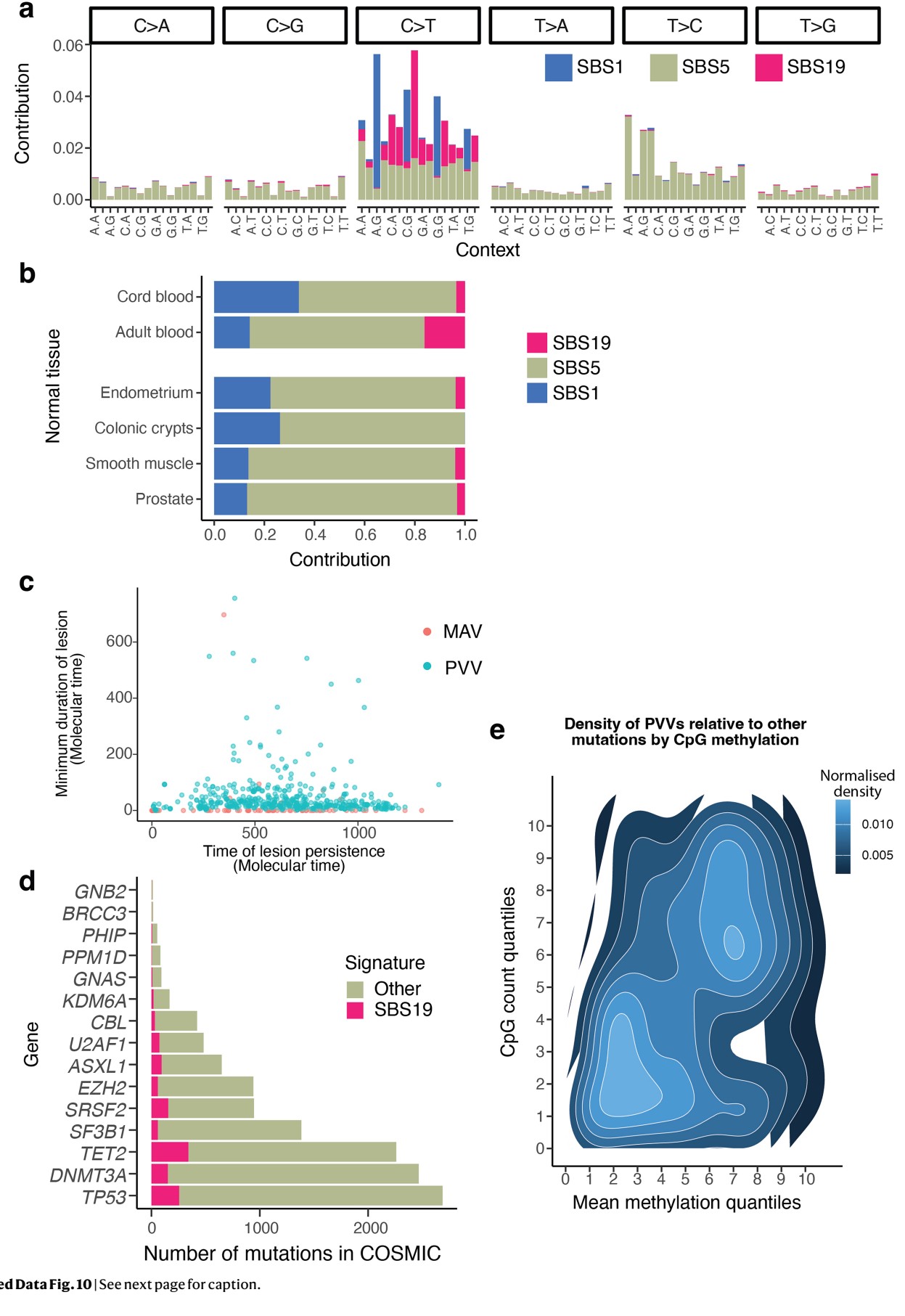

**Extended Data Fig. 10** | See next page for caption.

**Extended Data Fig. 10 | SBS19 causes 19% of mutations in HSPCs, including driver mutations. a**, Stacked barplot showing the signature decomposition of mutations in normal HSPCs. Bars are grouped by the 6 mutation types, and within each by the 16 base contexts comprising the base before and the base after the mutation. Each context has three stacked bars, denoting the fractional contribution of each signature to mutations in that specific context. **b**, Stacked barplot showing the fraction of mutations attributable to SBS1, SBS5 and SBS19 in normal cells from different organ systems. **c**, Scatterplot showing duration of persistent lesions (y axis) against the estimated age of occurrence in molecular time (x axis). **d**, Stacked barplot showing the number of mutations attributable to SBS19 in driver genes for myeloid cancers from the COSMIC database. **e**, 2D density plot showing the distribution of PVVs within the overall methylation landscape of the genome. The CpG count and mean methylation are scaled as by their quantiles within the overall genome, and therefore the null expectation would be even density across the plot.

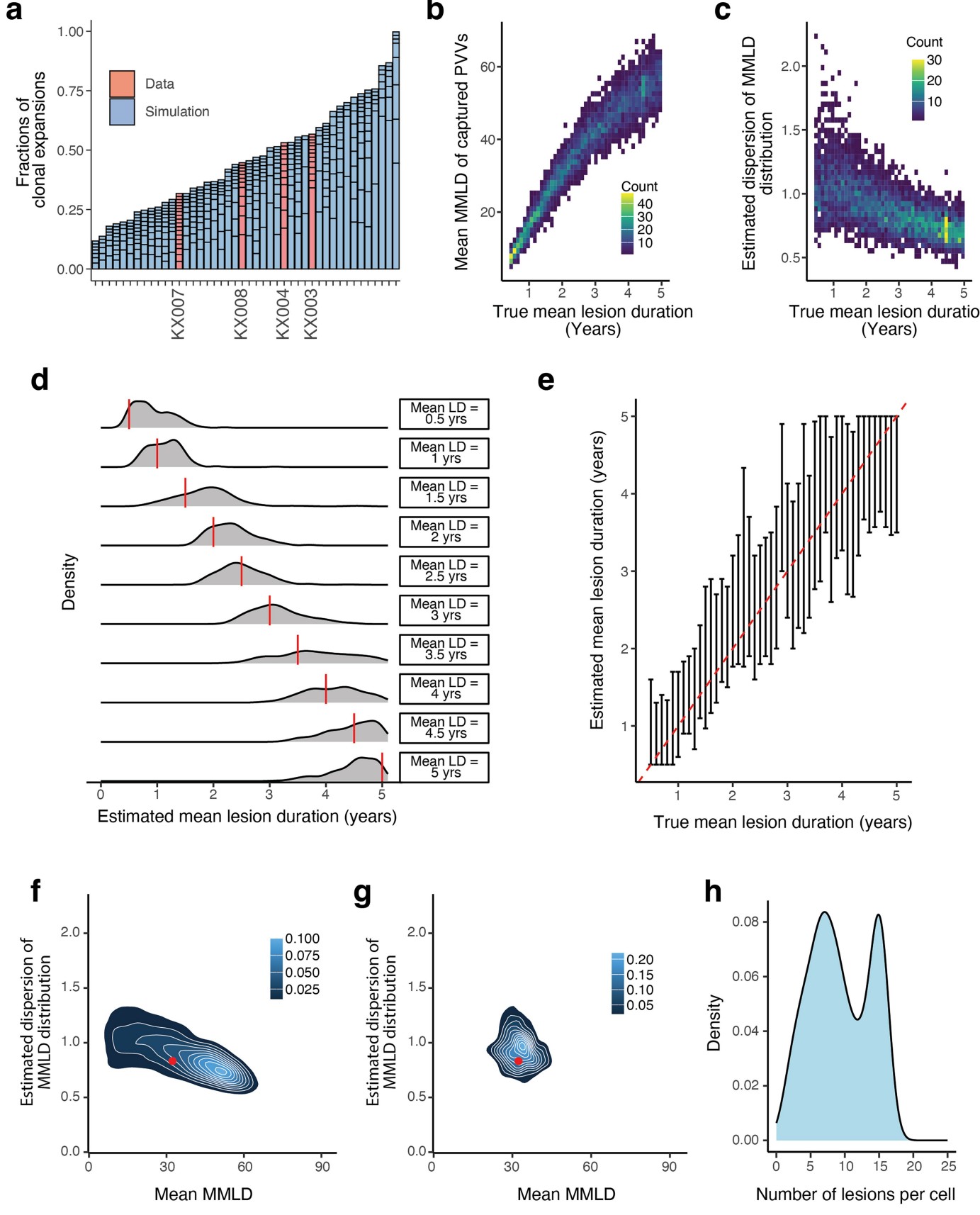

**Extended Data Fig. 11** | See next page for caption.

**Extended Data Fig. 11 | ABC for PVV lesion duration and frequency. a**, Stacked bar plot showing the contribution to the stem cell pool from clonal expansions (clonal fraction >1%) in the four old individuals used for the ABC (red) and the 40 simulated populations used for the ABC approach. Each bar represents a single clone. **b**, 2-dimensional density plot showing the mean MMLD of captured PVVs in simulation, compared to the true underlying mean lesion duration. **c**, as in b, but for the estimated gamma dispersion coefficient of the set of captured MMLDs in simulation. **d**, Density plot showing the posterior distributions from the ABC on selected down-sampled phylogenies with known lesion durations. For each, the 'ground truth' lesion is shown by the red line and is labelled on the right. **e**, error bar plot showing the 95% credibility intervals of the posterior distribution of the estimated mean lesion duration by applying the novel ABC framework to simulations. Red dotted line (y = x) illustrates the 'ground truth' values across the spectrum of simulations. Estimates are derived from n = 60,000 simulated PVVs per phylogeny. **f,g**, 2-D density plots showing the distribution of summary statistics (mean and estimated dispersion of the MMLDs of phylogeny-violating variants). **f** shows the data (red dot) compared to the summary statistics from the complete set of simulations, and **g** shows the data (red dot) compared to the summary statistics from simulations with parameters drawn from the posterior distribution, namely the posterior predictive checks. **h**, Posterior density plot showing the average number of lesions in any lineage at any time, inferred from the accepted simulations from ABC. MMLD, Minimum molecular lesion duration; ABC, Approximate Bayesian Computation; LD, lesion duration.

# Reporting Summary

## Statistics

For all statistical analyses, confirm that the following items are present in the figure legend, table legend, main text, or Methods section.

| n/a | Confirmed | |
|---|---|---|
| ☐ | ☒ | The exact sample size (*n*) for each experimental group/condition, given as a discrete number and unit of measurement |
| ☐ | ☒ | A statement on whether measurements were taken from distinct samples or whether the same sample was measured repeatedly |
| ☐ | ☒ | The statistical test(s) used AND whether they are one- or two-sided<br>*Only common tests should be described solely by name; describe more complex techniques in the Methods section.* |
| ☐ | ☒ | A description of all covariates tested |
| ☐ | ☒ | A description of any assumptions or corrections, such as tests of normality and adjustment for multiple comparisons |
| ☐ | ☒ | A full description of the statistical parameters including central tendency (e.g. means) or other basic estimates (e.g. regression coefficient) AND variation (e.g. standard deviation) or associated estimates of uncertainty (e.g. confidence intervals) |
| ☐ | ☒ | For null hypothesis testing, the test statistic (e.g. *F*, *t*, *r*) with confidence intervals, effect sizes, degrees of freedom and *P* value noted<br>*Give P values as exact values whenever suitable.* |
| ☐ | ☒ | For Bayesian analysis, information on the choice of priors and Markov chain Monte Carlo settings |
| ☐ | ☒ | For hierarchical and complex designs, identification of the appropriate level for tests and full reporting of outcomes |
| ☐ | ☒ | Estimates of effect sizes (e.g. Cohen's *d*, Pearson's *r*), indicating how they were calculated |

*Our web collection on statistics for biologists contains articles on many of the points above.*

## Software and code

Policy information about availability of computer code

| | |
|---|---|
| Data collection | None |
| Data analysis | List of programs and softwares:<br>• R: version 4.1.1<br>• BWA-MEM: version 0.7.17 (https://sourceforge.net/projects/bio-bwa/)<br>• cgpCaVEMan: version 1.13.14/1.14.1/1.15.0/1.15.1 (https://github.com/cancerit/CaVEMan)<br>• cgpPindel: version 3.3.0/3.5.0 (https://github.com/cancerit/cgpPindel)<br>• ASCAT NGS: version 4.2.1/4.3.2/4.3.3/4.5.0 (https://github.com/cancerit/ascatNgs)<br>• cgpVAF: version 2.4.0 (https://github.com/cancerit/vafCorrect)<br>• Julia language: https://julialang.org/<br>• treemut (v1.1, https://github.com/NickWilliamsSanger/treemut)<br>• hdp (v0.1.5, https://github.com/nicolaroberts/hdp)<br>• GenomicRanges (v1.46.1), IRanges (v2.28.0), Rsamtools (v2.10.0), MASS (v7.3-55), stringr (v1.4.1), dplyr (v1.0.10), tidyr (v1.2.1),  ape (v5.6-2), deconstructSigs (v1.8.0), ggplot2 (v3.4.0), MutationalPatterns (v3.4.1, 10.18129/B9.bioc.MutationalPatterns), gridExtra (v2.3), ggrepel (v0.9.2), RColorBrewer (v1.1-3), tibble (v3.1.8), dichromat (v2.0), seqinr (v4.2-16), phytools (v1.2-0), devtools (v2.4.5), lmerTest (v3.1-3), phangorn (v2.10.0), optparse (v1.7.3), parallel (v4.1.3)<br>Custom code made available (also stated in manuscript): https://github.com/mspencerchapman/Prolonged_persistence_of_DNA_lesions<br>No commercial software used. |

For manuscripts utilizing custom algorithms or software that are central to the research but not yet described in published literature, software must be made available to editors and reviewers. We strongly encourage code deposition in a community repository (e.g. GitHub). See the Nature Portfolio guidelines for submitting code & software for further information.

## Data

Policy information about availability of data

All manuscripts must include a data availability statement. This statement should provide the following information, where applicable:
- Accession codes, unique identifiers, or web links for publicly available datasets
- A description of any restrictions on data availability
- For clinical datasets or third party data, please ensure that the statement adheres to our policy

Sequence data that support the findings of this study have been deposited in the European Genome-Phenome Archive (https://www.ebi.ac.uk/ega/home) under the accession numbers relating to the original studies. These are listed below:
The longitudinal dynamics and natural history of clonal haematopoiesis (WGS Accession number EGAD00001007684)
Life histories of myeloproliferative neoplasms inferred from phylogenies (WGS Accession number EGAD00001007714 )
Clonal dynamics of haematopoiesis across the human lifespan (WGS Accession number EGAD00001007851)
Convergent somatic mutations in metabolism genes in chronic liver disease (WGS Accession number EGAD00001006255)
Tobacco smoking and somatic mutations in human bronchial epithelium (WGS Accession number EGAD00001005193)
Lineage tracing of human development through somatic mutations (WGS Accession number EGAD00001006162)
Clonal dynamics after allogeneic haematopoietic cell transplantation using genome-wide somatic mutations (WGS accession number TBC)
The long-term effects of chemotherapy on normal blood cells (WGS accession number TBC)
All scripts and downstream data matrices required to reproduce the figures are available on github (https://github.com/mspencerchapman/
Prolonged_persistence_of_DNA_lesions).
hg37 human reference genome has been used, as per all the original studies.

# Field-specific reporting

Please select the one below that is the best fit for your research. If you are not sure, read the appropriate sections before making your selection.

☒ Life sciences ☐ Behavioural & social sciences ☐ Ecological, evolutionary & environmental sciences

For a reference copy of the document with all sections, see nature.com/documents/nr-reporting-summary-flat.pdf

# Life sciences study design

All studies must disclose on these points even when the disclosure is negative.

| | |
|---|---|
| Sample size | We included all available datasets that had (1) multiple phylogenies built from whole genome sequencing data that included large numbers of samples per phylogeny (>30), (2) for which there was some clonal structure after the initial period of development in at least some of the dataset, to allow detection of prolonged DNA lesions. |
| Data exclusions | Data exclusions were done as per the original studies, with analysis done on the filtered dataset. |
| Replication | To ensure the robustness of the findings, analysis was done in multiple datasets including distinct cohorts analysed at different times. |
| Randomization | This is not relevant to our study. Samples were included as per the original studies. |
| Blinding | Blinding was not relevant to our study. There was no test performed that required blinding. |

# Reporting for specific materials, systems and methods

We require information from authors about some types of materials, experimental systems and methods used in many studies. Here, indicate whether each material, system or method listed is relevant to your study. If you are not sure if a list item applies to your research, read the appropriate section before selecting a response.

## Materials & experimental systems

| n/a | Involved in the study |
|---|---|
| ☒ | ☐ Antibodies |
| ☒ | ☐ Eukaryotic cell lines |
| ☒ | ☐ Palaeontology and archaeology |
| ☒ | ☐ Animals and other organisms |
| ☐ | ☒ Human research participants |
| ☒ | ☐ Clinical data |
| ☒ | ☐ Dual use research of concern |

## Methods

| n/a | Involved in the study |
|---|---|
| ☒ | ☐ ChIP-seq |
| ☒ | ☐ Flow cytometry |
| ☒ | ☐ MRI-based neuroimaging |

# Human research participants

Policy information about studies involving human research participants

| | |
|---|---|
| Population characteristics | The dataset comprised 11,429 whole genomes from 89 individuals. Each phylogeny was generated from a single tissue type: haematopoietic stem and progenitor cells (HSPCs, n=39), bronchial epithelial cells (n=16) or liver parenchyma (n=48, from 34 individuals, due to separate phylogenies for 8 anatomical segments of the liver in 2 subjects). The HSPC phylogenies were from individuals that fell into five categories: foetal and cord blood (n=4), healthy adults (n=13), stem cell transplant donor/ recipient pairs (n=10), patients with myeloproliferative neoplasms (n=10) and chemotherapy-exposed patients (n=2). Detailed metadata is available in Table S1. |
| Recruitment | Recruitment procedures were as outlined in the original studies (see references 9 - 16). |
| Ethics oversight | As per the original studies (see references 9 - 16). |

Note that full information on the approval of the study protocol must also be provided in the manuscript.

