## [Peer Review file · Nature]

Prolonged persistence of mutagenic DNA lesions in somatic cells

Corresponding Author: Dr Peter Campbell

Version 0:

Reviewer comments:

Referee #1

(Remarks to the Author)

The manuscript "Prolonged persistence of mutagenic DNA lesions in stem cells" by Campbell and colleagues builds on recent findings that show mutagenic DNA lesions can persist for multiple cell generations. They demonstrate that the persistence of lesions through multiple cell generations is a common occurrence through human development and tissue maintenance, and is consequently a significant and previously unappreciated source of mutations and somatic genetic variation. Using high-resolution phylogenetic trees to identify the mutagenic outcome of persistent lesions allows the authors to estimate lower bounds on the frequency of persistent lesions, show how that frequency varies with tissue, developmental time and exposure to exogenous mutagens, and remarkably allows an estimation of the persistence time per-lesion. As the authors have been appropriately conservative in calling multiallelic and "phylogeny violating" variants, and the observed phylogenetic trees represent a tiny fraction of the totality, all of these estimates likely reflect lower bounds. However, these limitations and the conservative nature of the estimates are well framed and explained in the manuscript.

In addition to the core findings of persistent lesions, clearly from exogenous and putatively endogenous mutagens, there are several additional but important findings. This includes the observation that APOBEC mutagenesis can generate a genome wide burst of mutations in a single cell cycle and that the previously noted strand co-ordination of clustered APOBEC induced mutations that was previously thought to arise from the modification of single stranded DNA, can alternately be explained as both strands being edited, but the asymmetry arising from lesion segregation. Another important finding is that multiallelic variation does not just manifest as alternate base substitutions, but can also be combinations of substitutions, insertions or deletions raising the possibility of different mutational outcomes from alternate lesion bypass strategies. Both of these findings are likely to motivate further work and better understanding of the underlying mutation and repair processes.

This is an exciting paper that will have considerable immediate impact on the fields of mutagenesis and cancer biology, with further implications for biochemistry, somatic and germline evolution. As well as being influential in its fields, the work is of general interest and written with very clear explanations making accessible to the wider audience. I think it's a good fit for Nature and I'm enthusiastically supportive of publication. I have a few minor comments that could be addressed but no substantial concerns.

#1. The title of the manuscript is short and punchy but I'm concerned that it can be read as "DNA lesions persist for longer in stem cells than other cells" which is not the conclusion of the work.

#2. Abstract "but whether some DNA damage can persist for longer durations remains unknown". Perhaps this be better as "but the extent to which some DNA damage....." since the persistence of lesions has previously been shown, but this work shows demonstrates the extent in human stem cells.

#3. Figure 1a-c. The shading of the newly replicated and template strands is not very distinct.

#4. Figures 1,4. Numbers in squares on branches are not documented. I assume these are probably branch lengths (new mutations on that branch). This should be annotated in a key or described in the legend.

#5. Figure 2. Many of the histograms in this figure show proportion as the y-axis value (panels a, b, g, h) but I don't find the interpretation very intuitive. Part of the problem is that displayed proportions don't always sum to 1.0 – for example in panel a

this appears to be because the 'unrelated' category is not shown. Minimally this should be documented in the figure or legend. I appreciate the authors are trying to compare observed with permuted in multiple categories across patients. A possible way to make this more intuitive is to show as stacked histograms (similar to panel f) but where simple MAVs are at the bottom, separated MAVs at the top and the currently missing unrelated are shown as lightly shaded in the middle. That would achieve direct side-by-side comparison of the observed and simulated proportions, and make clear what the proportions are and how they sum to 1.0.

#6. The authors repeatedly make reference to the repair of a lesion, e.g. "Since we know the nodes on the tree at which a persistent DNA lesion must have existed and, for PVVs and separated MAVs, the earliest node at which it was repaired, we can estimate the chronological age at which it occurred and a lower bound on the length of time it persisted unrepaired.", also in Figure 4 and Extended Figure 1. However, as the cell lineages are incomplete it is not possible to discriminate lesion repair from the loss of a lesion to a non-observed lineage. Most obviously a lesion that is retained in a stem cell for multiple cell divisions but then is inherited by a daughter cell destined for terminal differentiation or apoptosis, in this case the lesion can have shown multiallelic variation and then apparent repair in the observed lineages, but actually it was never repaired. For this reason it might be better to frame this as "repair or loss" rather than asserting repair.

#7. Figure 4 – "example of a MAV that must have been present in the zygote". While I accept that given the observed phylogeny shown in Figure 4a, it is possible the lesion was in the zygote, there are two further possibilities. First, it is conceivable (likely?) that all sequenced HSPC lineages do not coalesce to the zygote, but a later generation cell in the early embryo. Second, even if the root of the phylogeny is the zygote, then translesion resynthesis induced mutagenesis (TRIM, Anderson et al, bioRxiv 2022) involving a lesion introduced in the right hand daughter (as shown on the phylogeny) of the first cleavage division would generate the observed phylogeny.

martin.taylor@ed.ac.uk

Referee #2

(Remarks to the Author)

This is an interesting and elegant study examining the persistence of DNA lesions in human somatic cells. The authors have reanalysed their previously published data that used whole genome sequencing of multiple samples to reconstruct somatic cell phylogenies (mainly of haematological cells), finding evidence of sites in the genome where there are multiple mutations at the same site in different branches of the phylogenetic tree. Through various mathematical analyses they give good evidence that these multiply mutated sites arise because of (surprisingly) long-term persistence of (single stranded) DNA lesions. The mutations associated with the lesions have a distinct mutational signature. Overall I think the conclusions are quite well supported, and the paper is an interesting and novel read. Some comments and suggestions intended to help improve the study:

I found it hard to understand the need to use read overdispersion to find phylogeny violating variants (PVVs) - it seems a weaker test than just looking for the same variant multiple times in different clades on the tree? Can the authors add some (quantitative) justification here?

Can the authors provide some intuition as to why PVVs and MAVs should have a different mutational signature? A single process could generate both, and so PVVs could effectively just be a subset of all MAVs. (can the authors discount that idea?)

MAV persistence time: is there evidence in the data that the longevity of lesions changes through life (because, say, of changes in DNA repair efficiency)? (A simple first look would be to plot the correlation of inferred lesion persistence time with the number of mutations accrued in the phylogeny prior to the MAV arising)

Can the authors rule out copy number alterations (chromosome deletions) as causing some of the apparent action of APOBEC in a single cell division?

The methods section describing the simulation used to test for multiple mutations at the same location as a cause of MAVs/PVVs isn't clear. There is more maths methodology described for PVVs but this text comes after the MAV method have been presented so is unclear to read (and unclear if the MAVs method are the same as the PVVs).

I wasn't clear if the authors are correcting for the likelihood of different mutational types here (perhaps measured using mutational signature densities), which presumably could have a significant effect on the likelihood of seeing recurrent mutations (at highly mutable sites).

Can the authors also add an assessment how often the same mutation is seen in different individuals (in a particular tissue) as further discounting of the repeat mutation hypothesis?

For the ABC fits on the timing and duration of lesions, please can the authors confirm the accuracy of the method to recover the ground truth. They should perform inference with their framework on simulated datasets (where the ground truth is known). Where appropriate the posterior predictive fits of the model to the biological data should be illustrated.

Relatedly, I have a niggling concern that because the power to detect persistent lesions is greatest for early-ish times on the

phylogenetic tree (where the chance of sampling related descent lesions is maximised, as the authors explain) that conclusions about the rate and persistence of lesions throughout life are skewed. Could the authors perform simulations (probably agent based of stochastic branching process) where they simulate a growing phylogeny and empirical sampling from that phylogeny, and test their ability to measure parameters about the dynamics of persistent lesions (and specifically the correlation between inferred parameter accuracy and position on the phylogeny)? The understanding from these inferences may lead to tempering of conclusions here. My understanding is that this would be distinct to all the "permutation tests" the authors currently do (and the current ABC fitted simulations), which all take the observed phylogeny(s) as the starting point.

It would be great (of course) to know the mechanism by which some lesions persist and others do not. A think an analysis in reach is to look at the associations between lesion position in the genome and the typical epigenetic makeup of that position (chromatin state, replication timing etc) and see if there are correlations. Comparing the epigenetic state of early and late arising lesions (along the phylogenetic tree) should provide some kind of internal control to these analyses as the epigenetic state changes dramatically during development.

Referee #3

(Remarks to the Author)

Chapman et al. present their paper on the number, persistence, and timing of somatic lesions using public single-cell data from 89 individuals. The authors do a nice job setting up the problem, explaining the various lesion types and how they arise (i.e. figure 1) as well as providing solid context and motivation from the DEN exposure paper and its multiallelic outcomes. The paper uses somatic-mutation-generated trees to explore how long these lesions persist, and the various proportions of mutations (MAV types), and explore the minimum molecular lesion duration, a really interesting mutational timing that we know little about. The paper and its novelty grew on me as I worked through it a second time, and the authors did a nice job leveraging publicly available data to gain insights into an important aspect of somatic mutations. I have a few thoughts below that may (or not) help build the paper up and strengthen some of the arguments:

Really nice use of read-backed phasing and LOH (ASCAT) for validation of the PVVs

Much of this work rests on the trees and the validation of their structure. You go into it in the "Assessment of incorrect phylogeny structure as an artefactual cause of PVVs" section, but I feel like this needs even more validation or description. I know the data is public, and physical validation is out, but could you detail more about the branch-creating mutations? How are the coverage vs average mutations? Do they share any mutational signature (not COSMIC, just CpG or other simple class) or location? So much hinges on good trees it would be nice to be really confident

Along those lines, could you put more into characterizing the results of the trees – something along the lines of bootstrapping or general tree confidence? There's a lot resting on the structure (comment above) and the branch lengths, it would be nice to know how robust the trees are.

When you say, 'In all but one case, there was more than 1 somatic mutation that convincingly confirmed the consensus phylogeny – thus, for these branches, there was considerably more evidence for the original phylogeny than for the alternative phylogeny suggested by the single PVV' (section 'Assessment of incorrect phylogeny structure as an artefactual cause of PVVs'), what are the numbers here? Again more detail would be helpful.

Would it be possible to validate this in public cell culture data? It would be nice to see MAVs in a more artificial setting where the approaches used here could be validated. The best I could come up with is from the paper 'Quantification of somatic mutation flow across individual cell division events by lineage sequencing', but it would be great to see validation of some MAVs in a more constrained/synthetic system where some of the biological confoundings has been taken out. Maybe some of your groups' previous data sets could be used here too. Maybe the resolution isn't there in these data sets, but it would add a lot of confidence to the interpretations of this paper.

3e is a neat observation; again, it would be good to attempt to validate this class of indel mutations. Is the sample signature found in public data sets of blood cancers?

Version 1:

Reviewer comments:

Referee #1

(Remarks to the Author)

The authors have fully and satisfactorily addressed all the points I raised in first-round review. The updated ABC analysis motivated by referee #2 comments appears to have been important in accounting for the effects of lesion loss from the observed phylogenies, though the overall conclusions remain unchanged. Despite the substantial updates to the manuscript in response to reviewer comments, the main body of the manuscript remains concise and well presented, and I find the revised Figure 2 presentation much easier to interpret. I remain enthusiastically supportive of publication and I have no further points to raise.

Referee #2

(Remarks to the Author)

The rebuttal and revised manuscript are excellent and further improve what I thought was already a creative, fascinating and rigorous study.

The new simulation framework – and thorough assessment of the accuracy of model inferences – is particularly welcome and I commend the authors for their work here. I'm also glad this has caught and corrected what were previous slight misinterpretations of the data.

My previous final comment – about a possible role for the epigenome in facilitating lesion persistence - is relatively well addressed by the analysis of the correlation of lesion position and epigenetic features. It indicates that lesions persistence has some relationship to the methylation status of CpGs in the region. A minor suggestion for a small further analysis is to look at the position of late-arising lesions compared to lesions that arise early in development (as measured on the phylogeny), as presumably the methylation status of the genomic regions where lesions are found will tend to change through development. I emphasise this is a minor point and if it is inaccessible with current data (too unpowered), the manuscript stands without it.

Referee #3

(Remarks to the Author)

The authors do a nice job validating the tree reconstructions and filling in some missing pieces for me. I'm happy to see the paper accepted as-is.

Michelle Trekmann,
Senior Editor,
Nature.

1st August, 2024

Dear Michelle,

Re: “Prolonged persistence of mutagenic DNA lesions in somatic cells”

Thank you for the opportunity to provide a point-by-point outline response to the reviewer comments. We were pleased that overall the comments were positive. The major changes to the manuscript that we have made include:

- A new framework for the inference of lesion duration and frequency in the Approximate Bayesian Computation analysis, as suggested by reviewer 2;
- Increased robustness of phylogenetic tree reconstruction, comparing and incorporating trees from two further algorithms (IQ-tree and SCITE) to the original approach (MPBoot), as suggested by Reviewer 3; and
- More logical presentation of the analyses of alternative hypotheses for the observed data, as suggested by Reviewer 1.

Below, the reviewer comments are in blue, with our response in black and actions that we have undertaken for the revision in red and bold.

Referee #1 (Remarks to the Author):

The manuscript “Prolonged persistence of mutagenic DNA lesions in stem cells” by Campbell and colleagues builds on recent findings that show mutagenic DNA lesions can persist for multiple cell generations. They demonstrate that the persistence of lesions through multiple cell generations is a common occurrence through human development and tissue maintenance, and is consequently a significant and previously unappreciated source of mutations and somatic genetic variation. Using high-resolution phylogenetic trees to identify the mutagenic outcome of persistent lesions allows the authors to estimate lower bounds on the frequency of persistent lesions, show how that frequency varies with tissue, developmental time and exposure to exogenous mutagens, and remarkably allows an estimation of the persistence time per-lesion. As the authors have been appropriately conservative in calling multiallelic and “phylogeny violating” variants, and the observed phylogenetic trees represent a tiny fraction of the totality, all of these estimates likely reflect lower bounds. However, these limitations and the conservative nature of the estimates are well framed and explained in the manuscript.

In addition to the core findings of persistent lesions, clearly from exogenous and putatively endogenous mutagens, there are several additional but important findings. This includes the observation that APOBEC mutagenesis can generate a genome wide burst of mutations in a single cell cycle and that the previously noted strand co-ordination of clustered APOBEC induced mutations that was previously thought to arise from the modification of single stranded DNA, can

alternately be explained as both strands being edited, but the asymmetry arising from lesion segregation. Another important finding is that multiallelic variation does not just manifest as alternate base substitutions, but can also be combinations of substitutions, insertions or deletions raising the possibility of different mutational outcomes from alternate lesion bypass strategies. Both of these findings are likely to motivate further work and better understanding of the underlying mutation and repair processes.

This is an exciting paper that will have considerable immediate impact on the fields of mutagenesis and cancer biology, with further implications for biochemistry, somatic and germline evolution. As well as being influential in its fields, the work is of general interest and written with very clear explanations making accessible to the wider audience. I think it's a good fit for Nature and I'm enthusiastically supportive of publication. I have a few minor comments that could be addressed but no substantial concerns.

We thank the reviewer for these supportive comments.

#1. The title of the manuscript is short and punchy but I'm concerned that it can be read as "DNA lesions persist for longer in stem cells than other cells" which is not the conclusion of the work.

We appreciate the reviewer's point here **and have therefore changed the title to 'Prolonged persistence of mutagenic DNA lesions in somatic cells'**.

#2. Abstract "but whether some DNA damage can persist for longer durations remains unknown". Perhaps this be better as "but the extent to which some DNA damage...." since the persistence of lesions has previously been shown, but this work shows demonstrates the extent in human stem cells.

We agree that this wording is better **and have updated the abstract accordingly.**

#3. Figure 1a-c. The shading of the newly replicated and template strands is not very distinct.

We have altered the colour scheme to increase the contrast.

#4. Figures 1,4. Numbers in squares on branches are not documented. I assume these are probably branch lengths (new mutations on that branch). This should be annotated in a key or described in the legend.

The reviewer's assumption is correct. **We have updated the legend to clarify this point.**

#5. Figure 2. Many of the histograms in this figure show proportion as the y-axis value (panels a, b, g, h) but I don't find the interpretation very intuitive. Part of the problem is that displayed proportions don't always sum to 1.0 – for example in panel a this appears to be because the 'unrelated' category is not shown. Minimally this should be documented in the figure or legend. I appreciate the authors are trying to compare observed with permuted in multiple categories across patients. A possible way to make this more intuitive is to show as stacked histograms (similar to

panel f) but where simple MAVs are at the bottom, separated MAVs at the top and the currently missing unrelated are shown as lightly shaded in the middle. That would achieve direct side-by-side comparison of the observed and simulated proportions, and make clear what the proportions are and how they sum to 1.0.

This is an excellent suggestion which we have incorporated into the figure and updated the figure legend. The updated figure 2 is shown below (**Reviewer Figure 1**).

Figure 2

Reviewer Figure 1. Revisions of Figure 2. The panels that have changed are (a), (b), (g) and (h). We have separated the observed data and simulations, as suggested, and presented the proportions as stacked bar charts for each sample separately ((a) and (g)) or combined ((b) and (h)).

#6. The authors repeatedly make reference to the repair of a lesion, e.g. “Since we know the nodes on the tree at which a persistent DNA lesion must have existed and, for PVVs and separated MAVs, the earliest node at which it was repaired, we can estimate the chronological age at which it occurred and a lower bound on the length of time it persisted unrepaired.”, also in Figure 4 and Extended Figure 1. However, as the cell lineages are incomplete it is not possible to discriminate lesion repair from the loss of a lesion to a non-observed lineage. Most obviously a lesion that is retained in a stem cell for multiple cell divisions but then is inherited by a daughter cell destined for terminal differentiation or apoptosis, in this case the lesion can have shown multiallelic variation and then apparent repair in the observed lineages, but actually it was never repaired. For this reason it might be better to frame this as “repair or loss” rather than asserting repair.

This is an excellent point. **We have updated the language used around this in the manuscript (Results, ‘Timing and duration of persistent lesions’; p. 13).**

#7. Figure 4 – “example of a MAV that must have been present in the zygote”. While I accept that given the observed phylogeny shown in Figure 4a, it is possible the lesion was in the zygote, there are two further possibilities. First, it is conceivable (likely?) that all sequenced HSPC lineages do not coalesce to the zygote, but a later generation cell in the early embryo. Second, even if the root of the phylogeny is the zygote, then translesion resynthesis induced mutagenesis (TRIM, Anderson et al, bioRxiv 2022) involving a lesion introduced in the right hand daughter (as shown on the phylogeny) of the first cleavage division would generate the observed phylogeny.

Thank you for this insightful comment. Addressing the two possibilities suggested:

- (1) **The sequenced HSPC lineages may not coalesce to the zygote.** This is indeed possible. However, our observations from several HSPC phylogenies where we have targeted sequencing available suggests that in many cases the origin of the HSPC phylogeny is the origin of (at least) all embryonic lineages. We have shown, using bulk sequencing of one or more non-haematopoietic tissues, that the variant allele fractions of mutations on branches stemming from the root of the phylogeny sum to 0.5^{1,2}. If there were unobserved lineages contributing to these other tissues, then this would not be the case. In fact, in one case, using placental tissue, we demonstrated that the root of the HSPC tree was the origin of both embryonic and extraembryonic tissues². However, we accept that this may not always be the case, as there is stochasticity in early differentiation. Indeed, recent work suggests that marked asymmetry of the contribution of the first two blastomeres to embryonic and extraembryonic tissues may be the norm³.
- (2) **Translesion resynthesis induced mutagenesis produces the observed appearances.** We thank the reviewer for suggesting this possibility which we had not considered.

Considering both these possibilities, we have tempered our wording in the main text (‘Timing and duration of persistent lesions’; p. 6) and in the legend to Figure 4, citing the additional references.

Referee #2 (Remarks to the Author):

This is an interesting and elegant study examining the persistence of DNA lesions in human somatic cells. The authors have reanalysed their previously published data that used whole genome sequencing of multiple samples to reconstruct somatic cell phylogenies (mainly of haematological cells), finding evidence of sites in the genome where there are multiple mutations at the same site in different branches of the phylogenetic tree. Through various mathematical analyses they give good evidence that these multiply mutated sites arise because of (surprisingly) long-term persistence of (single stranded) DNA lesions. The mutations associated with the lesions have a distinct mutational signature. Overall I think the conclusions are quite well supported, and the paper is an interesting and novel read. Some comments and suggestions intended to help improve the study:

We thank the reviewer for these supportive comments.

I found it hard to understand the need to use read overdispersion to find phylogeny violating variants (PVVs) - it seems a weaker test than just looking for the same variant multiple times in different clades on the tree? Can the authors add some (quantitative) justification here?

We accept that it does not seem immediately intuitive for the overdispersion test to be the best way to find PVVs. The reviewer suggests 'just looking for the same variant multiple times in different clades on the tree'. While this sounds simple, it is not in fact straightforward, and we found the overdispersion test the most powerful method to detect such occurrences.

The branch assignment algorithm (*treemut*) assumes that each variant has been acquired only once in the phylogeny. The algorithm finds the single maximum likelihood branch for each mutation, even in cases where no single branch matches the distribution of positive samples particularly well. The statistical challenge is how to pick out the minority of mutations that do not neatly fit a single acquisition event, amongst up to 100,000 or more somatic mutations per phylogeny.

We experimented with two alternative approaches:

1) Identifying mutations that had a low p value calculated by the *treemut* algorithm.

Each branch assignment by *treemut* comes with an associated p-value: the chance of observing the read counts in the data, given a single acquisition event on the assigned branch. However, whatever p-value cut-offs we used, there was low specificity and sensitivity for the variants of interest. For example, some indels present in large clades had consistently biased mutant versus wild-type read counts leading to low p-values, even when a variant was consistently present only within the assigned clade.

2) Identifying mutations for which there were either (i) negative samples within the assigned clade, OR (ii) positive samples outside the assigned clade.

Similarly, this binary approach had poor sensitivity and specificity. It frequently yielded occurrences where there were a few variant reads in a single sample outside the assigned clade (likely sequencing errors or other artefacts), or negative samples within the positive clade where depth was relatively low and there were no reads reporting the variant allele by chance.

With optimised cut-off values for the over-dispersion parameter ρ , the beta-binomial test proved a powerful approach to detect occurrences where there were in fact two distinct positive and negative populations either within or outside the assigned clade. This approach is analogous to that used to detect true somatic mutations from artefacts in the mutation-calling algorithms⁴. The reason this approach works better than the other two methods we tried is that the observed data represents a mixture of two binomial distributions – a given sample will report the variant allele at either the base call error rate (if the sample is wild-type) or with probability 0.5 (for a sample with the mutation heterozygously on an autosome). Under a simple mutation assigned to the correct branch of the phylogeny, the two binomial distributions will assort perfectly such that the descendants in the mutant clade all report the variant allele at 0.5 and those outside the clade report it at the base-call error rate – there would be no overdispersion beyond this simple partition. For a PVV, however, there will always be a set of samples for which this partition of expected variant allele fraction is violated – either a wild-type subclone within the assigned mutant clade or a mutant subclone outside the assigned mutant clade. A test for overdispersion beyond that expected by chance would report on either scenario and formalises the statistical inference.

We have explained these points more fully in the manuscript (Methods; ‘Identification of phylogeny-violating variants’; pp. 26-27).

Can the authors provide some intuition as to why PVVs and MAVs should have a different mutational signature? A single process could generate both, and so PVVs could effectively just be a subset of all MAVs. (can the authors discount that idea?)

We agree that PVVs and MAVs are different manifestations of the same process: a persistent lesion giving rise to different bases through successive rounds of DNA replication or repair. Indeed, as the reviewer comments, it is entirely feasible for a single lesion to generate both PVVs and MAVs in varying proportions depending on (1) the lesion durations, (2) the phylogeny structure and (3) the probability of pairing with each of the 4 possible bases during translesion synthesis.

If this were the case, a couple of features would be anticipated:

- (1) If the same lesions underlie both MAVs and PVVs, they should have the same or very similar sequence context specificity;
- (2) Within comparable phylogeny structures, the relative proportions of PVVs and MAVs should be similar between individuals.

In most instances, this is not what we observe. For example, in the adult haematopoietic phylogenies, the PVVs almost exclusively affect C:G base pairs where there is a thymidine base 3' to the affected cytidine (**Figure 3c**). Conversely, within the same phylogenies, the MAVs have a less clear context predilection, with the main ‘cluster’ of MAVs being at T:A base pairs, particularly where there is a 5' adenine (**Figure 3b**). Even where there are MAVs affecting C:G base pairs, they do not display the same context specificity.

The exception to this is the chemotherapy phylogeny, PX001. Here, both PVVs and MAVs predominantly affect T:A base pairs, with similar sequence specificity. It is plausible, perhaps likely, that the same lesions generate both.

The high numbers of PVVs compared to MAVs in the normal blood phylogenies, and the distinctive sequence specificity, suggests a lesion giving rise almost exclusively to PVVs. This implies that translesion synthesis is highly selective for the pattern of base incorporation. In fact, simulations suggest >99.5% specificity for pyrimidine *versus* purine, but a lack of specificity for which pyrimidine is inserted opposite the lesion (**Reviewer Figure 2**).

We have highlighted these points in the Introduction (p. 3) and Results ('Numbers and signatures of MAVs and PVVs'; p. 10). Reviewer Figure 2 has been included as Extended Figure 9d.

Reviewer Figure 2 (included in manuscript as Extended Figure 9d). PVV:MAV ratio depending on pairing specificity during translesion synthesis. Simulated ratio of PVV:MAVs depending on the specificity for transition mutations during translesion synthesis (e.g. correctly incorporating a pyrimidine opposite a damaged guanine), rather than transversion mutations. TLS = translesion synthesis; PVV = phylogeny-violating variant; MAV = multi-allelic variant.

MAV persistence time: is there evidence in the data that the longevity of lesions changes through life (because, say, of changes in DNA repair efficiency)? (A simple first look would be to plot the correlation of inferred lesion persistence time with the number of mutations accrued in the phylogeny prior to the MAV arising)

The PVV data is most powerful to address this question, as all detected lesions have an associated minimal duration. There is no evidence from these data to suggest that the duration of lesions generating PVVs varies across the lifespan. Looking at the adult blood phylogenies (for which there is the most direct correlation between molecular time and age), there was no relationship between lesion duration and metrics relating to the age at which the lesion was present (linear regression, $R^2 = 0.005$, $p = 0.11$; **Reviewer Figure 3**). This suggests that repair efficiency for the causative lesions remains relatively constant through life. We have commented on this in the manuscript (Results, 'Timing and duration of persistent lesions'; p. 14) and included Reviewer Figure 3 as Extended Figure 10c.

Reviewer Figure 3 (included in manuscript as Extended Figure 10c). Minimum molecular lesion duration compared to the molecular time of the lesion.

Can the authors rule out copy number alterations (chromosome deletions) as causing some of the apparent action of APOBEC in a single cell division?

We have re-examined the copy number profiles and structural variant analysis for all the bronchial epithelial samples with evidence of significant APOBEC mutagenesis, and confirmed that none have any copy number alterations or structural variations. Even if there were deletions or uniparental disomy, this would not readily explain the Watson-Crick asymmetry observed. **We have now included this point in the manuscript (Results, ‘Strand asymmetry and lesion segregation’; p. 17).**

The methods section describing the simulation used to test for multiple mutations at the same location as a cause of MAVs/PVVs isn't clear. There is more maths methodology described for PVVs but this text comes after the MAV method have been presented so is unclear to read (and unclear if the MAVs method are the same as the PVVs).

As the detection of MAVs is more straightforward compared to PVVs, and less reliant on sample sequencing depth, the simulation approach for independent events is also simpler than for PVVs. The branch lengths, scaled to numbers of mutations, are estimates for the amount of time passed in that lineage. We therefore formally assessed the proportion of MAVs that would be expected to fall in ‘simple’, ‘separated’ or ‘unrelated’ orientations by modelling multi-allelic variant pairs occurring as random independent events within the phylogeny. To simulate this, we randomly selected pairs of phylogeny branches with probabilities proportional to their branch length, repeating this 50,000 times for each phylogeny. Each pair was categorised by the orientation of the two selected branches and compared with the proportions observed in the data. As for the observed data, any pair classified as

‘separated’ but with 2 or more intervening negative subclades were reclassified as ‘unrelated’. To assess the overall degree to which the set of MAVs may be contaminated by those occurring by chance, we calculated a weighted mean of the simulated proportions in each category, using the total number of MAVs detected in each phylogeny as weights.

We agree that the rationale and explanation of the methods section could be improved. **Therefore, we have expanded the relevant section of the Methods to include the detail described above (Methods, ‘Assessment of two independent mutations as an artefactual cause of MAVs’; p. 29).**

I wasn't clear if the authors are correcting for the likelihood of different mutational types here (perhaps measured using mutational signature densities), which presumably could have a significant effect on the likelihood of seeing recurrent mutations (at highly mutable sites).

We separately applied several different approaches to exclude the possibility that PVVs result from recurrent mutations at highly mutable sites. The different methods require different assumptions, but between them robustly refute the possibility that PVVs or MAVs result from independent mutation acquisition at hotspots. Given the difficulty of accurately modelling site-specific mutation probabilities, none of our methods explicitly make estimates regarding the overall likelihood of recurrent mutations.

(1) Examining the distribution of mutations within the phylogeny.

This method avoids making specific assumptions regarding the likelihood of recurrent mutations at different sites (which, as noted by the reviewer, would have to incorporate variables such as mutation signatures, chromatin accessibility, and local factors resulting in local hypermutation). Instead, the only assumption is that mutations at hotspots are equally likely to occur in any lineage, at any time. Thus, the phylogenetic relationship of two branches with the same mutation is essentially random. Conversely, two positive clades generated by a prolonged mutagenic lesion fall in a specific phylogenetic configuration (**Figure 1a-f**). As shown in **Figure 2g**, few mutations fall in this specific phylogenetic configuration by chance, whereas in the data, the majority do. This is particularly true for the large blood phylogenies (e.g. KX003, KX004, KX008), in which most PVVs are found.

(2) Examining the read-backed phasing of the mutant clades

Again, this method avoids making specific assumptions regarding the likelihood of recurrent mutations at different sites. Instead, we looked simply at whether the mutations at the same site in different positive subclades of a PVV were on the same or different alleles, making the assumption that independent acquisition of a mutation is as likely on opposite alleles as on the same allele. Although phasing was only possible for ~30% of variants, this showed that variants were on the same allele in almost all cases (**Figure 2d**), again making independent acquisition unlikely.

(3) Mutational signatures of PVVs

In this analysis, we compare the mutational signature of PVVs against that expected from a model in which the likelihood of mutation is defined by the mutation type and trinucleotide context. Thus, the likelihood of a mutation at any given site is proportional to the 96-profile

proportion of that mutation, divided by the density of that specific trinucleotide in the genome (fairly equal apart from a depletion of CpGs). By this simple model, the likelihood of the same site being mutated twice independently is thus proportional to the square of this value. To convert this likelihood back into an anticipated signature of such recurrent mutations, we can then multiply this likelihood by the density of each trinucleotide in the genome. The resultant signature is dominated by C>T transitions at CpGs (**Extended Data Figure 5d (i)**) which closely matches the signature of phylogeny-violating variants not in an orientation consistent with a prolonged mutagenic lesion (**Extended Data Figure 5e**). In support of this model is the degree to which the predicted signature matches that of mutations found across ≥ 2 individuals as discussed in the following point (see **Reviewer Figure 4**).

We have included these points in the manuscript (Results, ‘Identification of multi-allelic and phylogeny-violating variants’; pp. 6-8) and Reviewer Figure 4 as Extended Figure 5f.

Can the authors also add an assessment of how often the same mutation is seen in different individuals (in a particular tissue) as further discounting of the repeat mutation hypothesis?

We thank the reviewer for this excellent suggestion. Looking across 27 adult haematopoietic phylogenies, with 5,733,980 mutations among them, there were a total of 34,862 that were shared by two or more individuals. When doing pairwise comparisons, we found a very consistent rate of sharing of 2.5×10^{-9} of all mutation pairs when both individuals had mutational profiles dominated by the blood signature (calculated as *number of shared mutations / [total mutations in individual 1 * total mutations in individual 2]*). As expected, the mutational spectrum of these shared mutations was dominated by C>T transitions in a CpG context (**Reviewer Figure 4**), remarkably similar to the predicted spectrum of chance co-occurrence of mutations in blood cells (**Extended Fig. 5d (i)**). Importantly, this spectrum of identical mutations co-occurring in different individuals differs from the spectrum observed for PVVs occurring in the same individual, which is characterised by C>T variants in a CpT context (**Figure 3c**).

We have included this additional analysis in the manuscript (Methods, ‘Inference of expected PVV mutational signatures’; pp. 33-34) and Reviewer Figure 4 as Extended Figure 5f.

Reviewer Figure 4 (included in manuscript as Extended Figure 5f). Mutational signature of independent mutation acquisition events between phylogenies from different individuals.

For the ABC fits on the timing and duration of lesions, please can the authors confirm the accuracy of the method to recover the ground truth. They should perform inference with their framework on simulated datasets (where the ground truth is known). Where appropriate the posterior predictive fits of the model to the biological data should be illustrated.

Relatedly, I have a niggling concern that because the power to detect persistent lesions is greatest for early-ish times on the phylogenetic tree (where the chance of sampling related descent lesions is maximised, as the authors explain) that conclusions about the rate and persistence of lesions throughout life are skewed. Could the authors perform simulations (probably agent based of stochastic branching process) where they simulate a growing phylogeny and empirical sampling from that phylogeny, and test their ability to measure parameters about the dynamics of persistent lesions (and specifically the correlation between inferred parameter accuracy and position on the phylogeny)? The understanding from these inferences may lead to tempering of conclusions here. My understanding is that this would be distinct to all the "permutation tests" the authors currently do (and the current ABC fitted simulations), which all take the observed phylogeny(s) as the starting point.

We thank the reviewer for these points, and agree that the ABC framework needed additional refinement and validation. In fact, by working through this, we realised that the original ABC framework did not account for unobserved lineages, and we have therefore updated the framework to one that we believe is more accurate.

Assessing the original ABC framework

As the reviewer pointed out, the original framework was based on the observed phylogeny as the starting point. Within the simulations, lesions were randomly introduced into lineages, which then persisted through observed lineages in the phylogeny for a duration randomly selected from a gamma distribution.

To test this framework, we simulated 20 "elderly" complete haematopoietic stem cell populations using parameters drawn from the posterior distribution of the ABC from our study of normal haematopoiesis across the lifespan⁵ and an HSC population size of 100,000. This produced population structures that broadly reflected the oligoclonality of the 4 elderly phylogenies used in the ABC (**Reviewer Figure 5**). For each simulated population, and for a range of means, we then simulated the introduction of 5 million lesions, with durations randomly drawn from a gamma distribution, with a set mean [range: 0.5 - 5] and shape = 1. These were randomly introduced into the complete HSC phylogeny, again tracking their path through the phylogeny, and simulating whether a reference or alternate base was incorporated during translesion synthesis at any node that was crossed. Any lesion that created a detectable PVV in the complete phylogeny was recorded, others were discarded. The complete phylogenies were then down-sampled to a phylogeny of 500 samples. Depending on which samples were included in the down-sampled phylogeny, we assessed whether the PVV was still detectable. Notably only a small fraction of PVVs detectable in the complete phylogeny remained detectable in the down-sampled phylogeny (~1 in 1000 for lesion durations with a mean of 2 years). Using an average mutation rate of 17 mutations per year, and assuming a Poisson model of mutation acquisition, we then rescaled the phylogenies to molecular time, and recorded the minimum molecular lesion duration (MMLD) of each captured PVV.

Several interesting features became clear through these simulations:

- (1) The lesions captured in the final phylogeny were significantly longer than the average of all lesion durations. While this is perhaps unsurprising, it showed that captured lesions were, on average $\sim 2.5x$ longer than the average (**Reviewer Figure 6a**).
- (2) However, the duration of the captured lesions was only partially captured by the phylogeny. As discussed in the manuscript, the durations in the phylogeny are *minimum* molecular lesion durations, and indeed the MMLDs were, on average $\sim 0.3-0.4x$ the true lesion durations (**Reviewer Figure 6b**).
- (3) Overall, these two factors nearly cancel each other out, such that the average of the MMLDs of the captured lesions is close to the true average duration of the underlying distribution (**Reviewer Figure 6c**).

To test the original ABC framework, we then took the down-sampled simulated phylogenies and the MMLDs of the captured PVVs and went through the same lesion simulation procedure and ABC as we had originally done with the data. We then compared the inferred mean lesion durations with the ground truth lesion durations from the original simulation (**Reviewer Figure 7**). This highlighted the over-simplification in our original approach. At very low lesion durations, the framework tended to overestimate the lesion duration, although in general the true value was within the 95% posterior interval. However, for longer lesion durations, the ABC systematically underestimated the true lesion durations. We believe this to be because of the lack of consideration for the possibility of loss of a lesion to an unobserved lineage. Therefore, once the lesion duration increases, the simulations expect many extremely long lesions to be detected, whereas in reality, many lesions will be lost to an unobserved lineage. We therefore developed a new ABC framework based on complete phylogeny simulation.

Reviewer Figure 5 (included in manuscript as Extended Figure 11a). Oligoclonality of simulated populations. Stacked bar plot showing the contribution to the stem cell pool from clonal expansions (clonal fraction $>1\%$) in the four old individuals used for the ABC (red) and the 20 simulated populations used for benchmarking the ABC approach. Each bar represents a single clone. ABC, Approximate Bayesian Computation.

Reviewer Figure 6. Observations from simulations of prolonged lesions into complete HSC phylogenies. **a**, The y axis shows ratio of durations of lesions captured as PVVs to the mean of the underlying distribution of lesion durations. The x axis shows how this varies according to the mean of the underlying lesion duration. The black line shows the median ratio, and the grey shaded area shows the interquartile range. **b**, The y axis shows the ratio of the minimum lesion duration as captured in the phylogeny to the true lesion duration. The black line shows the median ratio, and the grey shaded area shows the interquartile range. **c**, The relationship between the mean minimum lesion duration (inferred from the molecular lesion duration) and the mean actual lesion duration.

Reviewer Figure 7. Performance of original ABC framework. Density plot showing the posterior distributions from the ABC on selected down-sampled phylogenies with known lesion durations. For each, the 'ground truth' lesion is shown by the red line and is labelled on the right. LD, lesion duration.

Improved ABC framework

Given the evident limitations of the original ABC framework, we developed a new framework based on *de novo* simulation of complete HSC phylogenies, as suggested by the reviewer.

As with the previous benchmarking analysis, the simulation framework started with 40 simulated “elderly” complete haematopoietic stem cell populations using parameters drawn from the posterior distribution of the ABC from our earlier study⁵ and an HSC population size of 100,000. These reflected the diversity of oligoclonality found in elderly phylogenies. Within each of these populations, lesions were randomly introduced with lesion duration drawn from a gamma distribution with shape = 1, and mean varying from 0.5 years to 5 years, $\mu = \{0.5, 0.6, 0.7 \dots 5\}$. For each population and lesion duration, enough lesions were introduced to ensure that at least 60,000 PVVs were theoretically detectable within the complete phylogenies. For longer lesion times, $\sim 2 \times 10^6$ lesions were sufficient, but for short lesion times (mean = 0.5 years), introduction of $> 3 \times 10^7$ lesions was sometimes necessary.

For each simulation run, a mean lesion duration between 0.5 and 5 was selected, and 4 of the aged HSC populations were randomly selected to represent the four elderly phylogenies in the data. Each selected population was then randomly down-sampled to the size of the actual phylogenies (328 tips for KX003, 922 tips for KX004, 315 tips for KX007 and 367 tips for KX008). We then iterated (in a shuffled order) through the potentially detectable lesions from the complete phylogeny to see if they remained detectable in the down-sampled phylogeny (again, this was typically ~ 1 in 1000). Once adequate lesions were detected to represent those found in the data (33 for KX003, 80 for KX004, 9 for KX007 and 22 for KX008), the simulation stopped. In some cases, despite assessing all 60,000 PVVs, the total number detectable remained somewhat fewer than the data (**Reviewer Figure 8**). Each down-sampled tree was then rescaled to molecular time and the MMLD of each captured PVV recorded. As with the data, any lesion with MMLD > 200 was removed from the set. Finally, a gamma generalised linear regression model (GLM) was used to estimate mean and dispersion parameters on the combined MMLD set. The dispersion and mean of the MMLD distribution clearly varied according to the mean duration of the underlying distribution (**Reviewer Figure 8a,b**), although the parameters plateaued somewhat once the mean lesion duration reached 3-4 years.

We next ran the ABC inference step using the *abc* function from the ‘abc’ package (<https://doi.org/10.32614/CRAN.package.abc>), using a tolerance of 0.05 and the ‘rejection’ method i.e. no regression step was included. This revealed a median posterior value of 2.1 years (1.7 – 2.8, 95% posterior credible interval) (**Reviewer Figure 8c**). This therefore implies that the lesions have mean durations of > 2 years before repair or loss to a lineage that undergoes cell death.

Reviewer Figure 8 (included in manuscript as Figure 5a and Extended Figure 11b,c). Inference of the true mean lesion duration using ABC. **a**, 2-dimensional density plot showing the mean MMLD of captured PVVs in simulation, compared to the true underlying mean lesion duration. **b**, as in **a**, but for the estimated gamma dispersion coefficient of the set of captured MMLDs in simulation. **c**, prior and posterior distributions of the true mean lesion duration of persistent lesions in HSCs based on the MMLDs in the data. The uniform prior distribution is shown with the pale blue fill, and the posterior distribution with the red fill. ABC, Approximate Bayesian Computation; MMLD, minimum molecular lesion duration; PVV, Phylogeny-violating variant.

Ability of novel ABC framework to recover the ‘ground truth’ lesion durations

As suggested by the reviewer, we next assessed the performance of the new framework in recovering the true lesion durations from simulated data. We therefore simulated 4 additional elderly phylogenies, and again introduced simulated lesions of varying duration drawn from a gamma distribution of varying mean, $\mu = \{0.5, 0.6, 0.7 \dots 5\}$ and shape = 1. As before, we then down-sampled the phylogenies to the size of those from the data and determined if each PVV from the full phylogeny remained detectable, and its MMLD. For each mean lesion duration, we sampled MMLDs from the complete set such that the total numbers were the same as the data. Finally, for each value of μ we performed the parameter inference using the *abc* function as described above. This showed that our improved framework performed well across the range of mean lesion durations (**Reviewer Figure 9**), with the ‘ground truth’ mean lesion duration falling within the 95% confidence intervals of the inferred values in 45 of 46 cases.

Reviewer Figure 9 (included in manuscript as Extended Figure 11d,e). Performance of the novel ABC framework in recovering ‘ground truth’ mean lesion durations. **a**, Density plot showing the posterior distribution of the inferred mean lesion duration of simulated data for selection durations across the simulated range. Red lines indicate the ‘ground truth’ mean lesion durations. **b**, error bar plot showing the 95% credibility intervals of the posterior distribution of the estimated mean lesion duration by applying the novel ABC framework to simulations. Red dotted line ($y = x$) illustrates the ‘ground truth’ values across the spectrum of simulations. LD, lesion duration.

Posterior predictive checks

We performed posterior predictive checks⁶, drawing values of the mean lesion duration parameter μ from the posterior distribution and running additional simulations. In this way, we performed 230 additional simulations. These further simulations confirmed that the observed data fell well within the expected posterior distribution estimated from the ABC (**Reviewer Figure 10**).

Reviewer Figure 10 (included in manuscript as Extended Figure 11f,g). Posterior predictive checks. a-b 2-D density plots showing the distribution of summary statistics (mean and estimated dispersion of the MMLDs of phylogeny-violating variants). **a** shows the data (red dot) compared to the summary statistics from the complete set of simulations, and **b** shows the data (red dot) compared to the summary statistics from simulations with parameters drawn from the posterior distribution i.e. the posterior predictive checks. MMLD, Minimum molecular lesion duration.

Inferring the average number of lesions per cell at any time

In the new ABC framework, one can record how many lesions need to be introduced in order to create sufficient PVVs to match the data. One also knows the total amount of ‘lineage time’ of the complete HSC population into which lesions are simulated, which is the sum of all the edge lengths (scaled to days). Using this data, and the distribution of lesion durations, one can readily infer the average number of lesions one would expect in any given lineage at any time. We therefore created a posterior distribution for this parameter, using the parameters from the accepted set of simulations from the *abc* step (**Reviewer Figure 11**). One limitation is that the sensitivity for PVVs varies by phylogeny structure, and this was not accounted for in the model: lesions were simply added to a phylogeny until sufficient PVVs were detected to match the data. Therefore, for a simulated ‘insensitive’ clonal structure the implied lesion density may be artificially high if it were trying to generate the same number of PVVs as a more sensitive clonal structure from the data. However, it at least gives a broad sense of the range in which the true value may sit. This is ~8.8 lesions per cell (1.9 – 16.0, 95% credibility interval).

Reviewer Figure 11 (included in the manuscript as Extended Figure 11h). Average number of lesions per cell. Density plot showing the average number of lesions in any lineage at any time, inferred from the accepted simulations from ABC.

Conclusions from the improved ABC framework

Overall, we feel that this new ABC framework provides a more accurate way of assessing the lesion duration distribution of the lesions responsible for the PVVs in blood. While the estimates have relatively broad credibility intervals, it nonetheless implies that the mean duration is ~2 years.

We have updated the text, figures, and methods with the updated framework and its results. Specifically, the new posterior estimates for duration and number of lesions per cell have been quoted in the main text (Results, ‘Frequency and properties of lesions causing PVVs in HSPCs’; pp. 16-17). The Methods section has a detailed summary of the new ABC framework (Methods, ‘Inference of mean lesion duration’; pp. 35-38). We have incorporated Reviewer Figures 5 and 8-11 into Figure 5a and Extended Figure 11.

It would be great (of course) to know the mechanism by which some lesions persist and others do not. I think an analysis in reach is to look at the associations between lesion position in the genome and the typical epigenetic makeup of that position (chromatin state, replication timing etc) and see if there are correlations. Comparing the epigenetic state of early and late arising lesions (along the phylogenetic tree) should provide some kind of internal control to these analyses as the epigenetic state changes dramatically during development.

We agree with the reviewer that this is a fascinating question. We performed an initial analysis of the genomic positions of the blood PVVs relative to various genomic features, defined in kilobase bins of the genome. This included the density of Alu repeats, centromeric regions, CpG islands, histone methylation marks, replication timing, GC density, recombination rates and other features (**Reviewer Table 1**, full metric descriptions available in the pan-cancer structural variation paper⁷). For each PVV and metric, we defined the quantiles for that metric, defined by a control set of mutation positions, here taken as the large set of 1.7×10^6 single nucleotide variants in HSPCs (from the transplant cohort). For each metric, across all PVVs, we binned the PVVs into deciles and applied a chi-squared test to determine if the PVVs deviated from a uniform distribution, applying multiple hypothesis correction (Benjamini-Hochberg method). The only significant metric was for the ‘DNAMethylSBS’ measure of the proportion of methylated CpGs within the 1kb bin (chi-squared test, $q = 0.001$). This is the average

fractional methylation value in the 1kb bin measured in myeloid cells from DNAMethylSBS experiments in the ‘Roadmap’ epigenomics study⁸. Curiously, although the distribution deviated from a uniform distribution, there was no consistent trend towards higher or lower average fractional methylation for the PVVs (**Reviewer Figure 12**).

To further investigate this possible relationship between PVVs and CpG methylation status, we analysed the set of blood PVVs in relation to various features of the methylation landscape, defined from methylation sequencing done specifically on colonies from donor CB001. Again, we divided the genome into 1kb bins, and defined (1) the CpG context of the region i.e. CpG island, shore, shelf or inter region, (2) the number of CpG sites and (3) the fractional methylation. We first looked at whether the PVVs were biased towards particular CpG contexts. Filtering down to just the 218 C>T at CpT PVVs from blood (the most confident set), there was a possible mild bias away from islands, shores and shelves to the ‘inter’ regions (95% of PVVs in the inter regions, compared to 90% of the overall genome; $p = 0.04$, chi-squared test). Looking at the CpG count and fractional methylation in combination also shed more light on the bimodal location of the PVVs on this 2D landscape, with a peak in low CpG count, low mean methylation regions, and a second peak in high CpG count, high mean methylation regions (**Reviewer Figure 13**). The underlying biological significance of this relationship remains unclear, and further work will be needed to better understand the mechanisms by which some lesions persist.

We have discussed these findings in the manuscript (Results, ‘Frequency and properties of lesions causing PVVs in HSPCs’; p. 17), and included Reviewer Figure 13 as Extended Figure 10e and Reviewer Table 1 as Table S3.

Feature	P value	Q value
DNAMethylSBS	3.06E-05	0.00113357
triplex_mirror_rep_dist_log10	0.06581838	0.81302677
recomb_rate_nearest_value	0.10362645	0.81302677
gc_content_value	0.10682858	0.81302677
telomere_dist_log10	0.1169569	0.81302677
L2_rep_dist_log10	0.17089747	0.81302677
ALU_rep_dist_log10	0.17578957	0.81302677
H3K4me2	0.17578957	0.81302677
cruciform_inverted_rep_dens_3e3	0.2502637	0.89062996
DNase	0.28101852	0.89062996
centromere_dist_log10	0.28459926	0.89062996
LTR_rep_dist_log10	0.29553863	0.89062996
H3K4me3	0.36734545	0.89062996
L1_rep_dist_log10	0.39357038	0.89062996
z_dna_motif_dist_log10	0.40709084	0.89062996
RNAseq	0.44440501	0.89062996

seq_complexity_value	0.50346028	0.89062996
MIR_rep_dist_log10	0.57068699	0.89062996
TAD_b_dist_log10	0.58653259	0.89062996
DNA_rep_dist_log10	0.60245761	0.89062996
H3K9ac	0.61843832	0.89062996
short_tandem_rep_dens_3e3	0.65046546	0.89062996
H3K79me2	0.65580031	0.89062996
H4K20me1	0.65580031	0.89062996
direct_rep_dist_log10	0.66645816	0.89062996
G4L1_dist_log10	0.68769629	0.89062996
H2A.Z	0.72442744	0.89062996
cpg_islands_dist_log10	0.72961097	0.89062996
LAD_dens_1e6	0.74503966	0.89062996
rep_timing_value_value	0.75013787	0.89062996
H3K4me1	0.76528128	0.89062996
H3K9me3	0.77027456	0.89062996
SIMPLE_REPEAT_rep_dist_log10	0.79477618	0.89111269
H3K36me3	0.88239868	0.94707842
gene_dens_1e6	0.92124323	0.94707842
H3K27ac	0.93908794	0.94707842
H3K27me3	0.94707842	0.94707842

Reviewer Table 1 (included in manuscript as Table S3). Features of the phylogeny-violating variants compared to a complete set of blood mutations. For each feature, the quantiles of the metric were calculated for all PVVs, with the quantiles defined from a large set of blood mutations. The quantiles were binned into deciles, and compared with the expected uniform distribution by applying a chi-square test to obtain p-values. q-values were obtained by applying Benjamini-Hochberg multiple hypothesis correction.

Reviewer Figure 12. Distribution of DNAMethylSBS quantiles of the phylogeny-violating variants. Density plot showing the distribution of the 'DNAMethylSBS' metric of the phylogeny-violating variants, assessed as quantiles from the metric measured from the positions of a large set of single nucleotide variants from blood. Under the

null model, these would show a uniform distribution. DNAMethylSBS, average fraction methylation as measured by sodium bisulphite sequencing experiments of myeloid cells.

Reviewer Figure 13 (included in manuscript as Extended Figure 10e). Distribution of the CpG count and mean methylation saturation of the genomic regions of C>T at CpT site PVVs. 2D density plot showing the distribution of PVVs within the overall methylation landscape of the genome. The CpG count and mean methylation are scaled as by their quantiles within the overall genome, and therefore the null expectation would be even density across the plot.

Referee #3 (Remarks to the Author):

Chapman et al. present their paper on the number, persistence, and timing of somatic lesions using public single-cell data from 89 individuals. The authors do a nice job setting up the problem, explaining the various lesion types and how they arise (i.e. figure 1) as well as providing solid context and motivation from the DEN exposure paper and its multiallelic outcomes. The paper uses somatic-mutation-generated trees to explore how long these lesions persist, and the various proportions of mutations (MAV types), and explore the minimum molecular lesion duration, a really interesting mutational timing that we know little about. The paper and its novelty grew on me as I worked through it a second time, and the authors did a nice job leveraging publicly available data to gain insights into an important aspect of somatic mutations. I have a few thoughts below that may (or not) help build the paper up and strengthen some of the arguments:

Really nice use of read-backed phasing and LOH (ASCAT) for validation of the PVVs

Much of this work rests on the trees and the validation of their structure. You go into it in the “Assessment of incorrect phylogeny structure as an artefactual cause of PVVs” section, but I feel like this needs even more validation or description. I know the data is public, and physical validation is out, but could you detail more about the branch-creating mutations?

The consensus phylogeny, as relates to each PVV, is defined by the clades within the ‘lesion node’ and the ‘lesion repair node’. If these are robust, then this suggests that the phylogeny is correct and that the PVV is the exception. We therefore assessed the robustness of these nodes using three different approaches: (i) bootstrap support for the nodes using the bootstrap support values from the ‘MPBoot algorithm; (ii) bootstrap support for the nodes using bootstrapping of the read count matrices; and (iii) the nodes being identical when the phylogenies are reconstructed using alternative algorithms.

MPBoot bootstrap support for the lesion node and lesion repair node

The ‘MPBoot’ algorithm used to construct the phylogenies gives bootstrap support values for each node⁹. Therefore, for 440 of 501 PVVs (those for which these bootstrap support values were available, all within the haematopoietic phylogenies), we examined the bootstrap support values for the lesion node / lesion repair node for each PVV. This showed that for 90% of PVVs both lesion node and lesion repair node had bootstrap support values $\geq 98\%$, and for 96% of PVVs, both had support values $\geq 80\%$ (**Reviewer Figure 14a**). Only 15/440 PVVs (3.4%), across 9 different nodes, had $< 80\%$ support for either node, these are discussed further below. This demonstrates the high confidence in relevant phylogeny structures.

Read count bootstrap support for the lesion node and lesion repair node

An alternative bootstrapping approach is to bootstrap the variant read counts across all samples and mutation sites, as previously described^{2,10}. In this approach, we use the partially filtered mutation set and bootstrap the sequencing read counts for each colony at each locus before subjecting this raw read count data to the same filtering and phylogeny-building algorithms as the original data, with 200-250 replicates per individual. This is computationally intensive and therefore was applied to only 9 blood phylogenies, which accounted for 202/501 PVVs. Again, this suggested that 93% of PVVs had 100% bootstrap support for both lesion node and lesion repair node, and only 7/202 assessed PVVs (3.5%) had less than 80% bootstrap support for either node, discussed further below (**Reviewer Figure**

14b,c). All but one of these had also been identified as low confidence by the MPboot bootstrap support measure.

Reviewer Figure 14 (included in manuscript as Supplementary Figure 1). Bootstrap support for the phylogeny nodes defining the phylogeny-violating variants. **a**, Jittered scatter plot showing the MPBoot bootstrap support values of the 'lesion node' and 'lesion repair node' of each of 440 PVVs. 90% of points are concentrated in the dense area in the top right where both nodes have >98% bootstrap support. **b**, as in **a**, but for the read count bootstrap support, and for a subset of 202 PVVs. **c**, Dot plot showing the MPBoot and readcount bootstrap support for all assessed PVV lesion/lesion repair nodes (the minimum support of the two) arranged by decreasing support values. Points are joined by a line to better illustrate the trend. PVV, phylogeny-violating variants.

Support for the lesion node and lesion repair node from alternative phylogeny algorithms

Another way to assess the robustness is to compare the phylogeny generated by 'MPBoot' to other phylogeny-reconstruction approaches to see if phylogenies are generated with identical 'lesion node' and 'lesion repair node' clades. Therefore, for three of the blood datasets accounting for 439/501 PVVs we compared the clades defined by the lesion nodes/lesion repair nodes in phylogenies generated by IQ-tree¹¹, and where feasible, SCITE¹². The output for the SCITE algorithm frequently generated phylogenies with polytomies at the site of PVVs, causing fairly frequent discrepancies in the nodes despite the presence of the underlying inconsistent genotypes. Therefore, for the remainder of the analysis we focussed on the IQ-tree phylogenies. For 93% (409/439) of PVVs, the lesion node and lesion repair node were identical. For the remainder, we re-ran the PVV detection algorithm using the trees generated by the IQ-tree algorithm, and confirmed that although the clades had subtle differences (sometimes due to removal of a low coverage sample), 18/30 of the remaining PVVs were still called. The remaining 12 mutations affected 8 different nodes, and again 8 of the 12 overlapped with those identified as low confidence by the MPBoot bootstrap confidence approach.

PVVs highlighted as possible low confidence

Altogether, 19/440 (4.3%) blood PVVs were identified as potentially low confidence by one or more of the above approaches (**Reviewer Table 2**). We manually inspected each of these in an attempt to understand which of these were incorrectly called, and the likely underlying reasons. Several patterns emerged:

Scenario 1. Due to the PVVs, there is a degree of equipoise as to the true underlying phylogeny; therefore an alternate configuration for the phylogeny may be produced from bootstrapping or alternative phylogeny reconstruction. However, there remains no single

phylogeny that fits all mutations, and one or more PVVs is always present. In this scenario, there may be question marks over which mutations follow the 'true' phylogeny, and which represent PVVs generated by persistent lesions. In several cases, the original MPBoot phylogeny was still best supported (**Reviewer Figure 15a,b**). However, in one example, highlighted by all three approaches, the original phylogeny was supported by 3 mutations, but the alternate phylogeny was supported by 4 mutations (**Reviewer Figure 15c**). This made the alternate phylogeny more likely. In addition, the mutational profile of the 3 mutations that became PVVs according to the alternate phylogeny were more in keeping with the typical PVV profile, as all were C>T at CpT sites. This suggested that the IQ-tree was the correct configuration. We therefore removed the original 4 PVVs and replaced them with the 3 from the IQ-tree analysis.

Scenario 2. An alternative phylogeny is generated which highlights a potential issue with the original phylogeny, and that there is in fact unlikely to be a prolonged DNA lesion. The reasons for the incorrect original phylogeny include:

- a. Likely independent acquisition of the same mutation leading to an incorrect original phylogeny (**Reviewer Figure 15g,h**). In one case (the chemotherapy phylogeny), this mutation is a driver mutation in *PPM1D*, showing an example of convergent evolution (**Reviewer Figure 15h**).
- b. Low sample coverage in one or more samples leading to incorrect phylogeny building (**Reviewer Figure 15d,e**).
- c. Inappropriate inclusion of a mixed colony in the MPBoot phylogeny which therefore cannot be correctly placed on the phylogeny (**Reviewer Figure 15f**).

Scenario 3. The reason for the low bootstrap support from MPBoot is unclear and the PVV appears correctly identified on manual inspection (**Reviewer Figure 16a-d**).

As a result of this detailed interrogation of 440/501 PVVs, a total of 10 PVVs were removed, and 3 added to the final mutation set. Overall, this demonstrates that, within the limits of the analysis, the vast majority of PVVs (98%) are robust. However, not all alternative mechanisms can be excluded for each individual PVV.

We have included these assessments of the quality of the phylogeny inference in the Methods, commenting on the potential sources of error we identified (Methods, 'Assessment of incorrect phylogeny structure as an artefactual cause of PVV'; pp. 34-37). We have included Reviewer Figures 14-16 as Supplementary Figures 1-3.

Sample_ID	PVV mutation reference	Flagged as low confidence by			Likely reason (Manual inspection)
		MPBoot Bootstrapping	Read count Bootstrapping	Alternative phylogeny	
KX003_5_01	3-102586135-T-A	TRUE	TRUE	TRUE	PVVs present, but alternative phylogeny more likely
KX003_5_01	4-32490708-T-A	TRUE	TRUE	TRUE	PVVs present, but alternative phylogeny more likely
KX003_5_01	8-128309550-C-A	TRUE	TRUE	TRUE	PVVs present, but alternative phylogeny more likely
KX003_5_01	8-29759354-G-A	TRUE	TRUE	TRUE	PVVs present, but alternative phylogeny more likely
KX004_5_01	1-23213901-G-A	TRUE	FALSE	FALSE	PVV appears genuine, low confidence likely due to the PVV itself
KX004_5_01	6-54014440-T-G	TRUE	FALSE	FALSE	PVV appears genuine, low confidence likely due to the PVV itself
KX004_5_01	8-138169531-G-A	TRUE	FALSE	FALSE	PVV appears genuine, low confidence likely due to the PVV itself
KX009_1_01	7-140167414-G-A	TRUE	TRUE	FALSE	PVV appears genuine, low confidence likely due to the PVV itself
PX001_2_01	3-57906539-C-T	TRUE	TRUE	TRUE	Incorrect original phylogeny likely due to independent mutation acquisition at PPM1D hotspot. No genuine PVV.
Pair24	6-20117614-C-T	TRUE	FALSE	TRUE	PVVs present. Alternative phylogeny less likely than original.
Pair31	2-171172346-C-T	TRUE	FALSE	FALSE	PVV appears genuine, unclear reason for low confidence
Pair38	X-108747536-C-A	TRUE	FALSE	TRUE	PVVs present. Alternative phylogeny less likely than original.
Pair38	19-51271716-T-TA	TRUE	FALSE	FALSE	PVV present, though alternate mechanisms plausible.
Pair41	3-182038851-AT-A	TRUE	FALSE	TRUE	Low coverage sample leading to incorrect original phylogeny. No genuine PVV.
PX001_2_01	1-32201632-C-T	FALSE	TRUE	TRUE	Incorrect original phylogeny likely due to independent mutation acquisition. No PVV.
Pair28	10-63986636-C-T	FALSE	FALSE	TRUE	Mixed colony included in original analysis. No genuine PVV.
Pair28	14-82666754-G-A	FALSE	FALSE	TRUE	Mixed colony included in original analysis. No genuine PVV.
AX001_4_01	6-104547705-C-A	FALSE	FALSE	TRUE	Low coverage sample leading to incorrect original phylogeny. No genuine PVV.

Reviewer Table 2. Potential low confidence PVVs. Phylogeny violating variants highlighted as potentially low confidence by one or more of (1) low MPBoot bootstrap support, (2) low read count bootstrap support or (3) not called in the phylogeny built using IQ-tree. In each case the likely reason for the discrepancy/ low-confidence is shown (as assessed by the authors).

Reviewer Figure 15 (included in manuscript as Supplementary Figure 2). **Phylogeny-violating variants for which either the lesion node, or lesion repair node are different in the IQ-tree phylogeny.** a-h, The clade defined by the PVV lesion node has been extracted from the full MPBoot phylogeny, shown on the left. On the right is the relevant clade extracted from the IQ-tree phylogeny, chosen as the latest clade containing all samples found within the MPBoot lesion node clade. These clades are truncated to better illustrate the features near the root. The y axis is molecular time, stated relative to the time of the lesion node (i.e. the root of the subtree shown). The heatmaps show the VAFs of mutations on a scale of white (absent) to orange (for the PVVs) or blue (for the branch-creating mutations, limited to ≤ 10 for visualisation). Mutation references are shown in the format Chromosome-Position-Reference base-Mutant base. PVV, Phylogeny-violating variant; BCM, Branch-creating mutations.

Reviewer Figure 16 (included in manuscript as Supplementary Figure 3). Phylogeny-violating variants for which either the lesion node, or lesion repair node has low bootstrap support. This figure includes variants where either the lesion or lesion repair node have bootstrap support values $<80\%$ by either the MPBoot or read count bootstrap assessments. However, it does not include those already shown in Rebuttal Fig. X. **a-d**, the clade defined by the PVV lesion node has been extracted from the full MPBoot phylogeny and truncated to better illustrate the features near the root. The y axis is molecular time, stated relative to the time of the lesion node. The heatmaps show the VAFs of mutations on a scale of white (absent) to orange (for the PVVs) or blue (for the branch-creating mutations, limited to ≤ 10 for visualisation). Mutation references are shown in the format Chromosome-Position-Reference base-Mutant base. PVV, Phylogeny-violating variant; BCM, Branch-creating mutations.

How are the coverage [of branch-creating mutations] vs average mutations? Do they share any mutational signature (not COSMIC, just CpG or other simple class) or location? So much hinges on good trees it would be nice to be really confident

The reviewer asks about the coverage of the branch-creating mutations compared to the other mutations. Indeed, plausibly, these mutations may represent low coverage/ low confidence mutations. However, we found that these $\sim 8,000$ mutations had an almost identical mean coverage to the mutation set as a whole (**Reviewer Table 3**).

The mutational spectrum of the 8,000 branch-creating mutations was the same as the full mutation set (**Reviewer Figure 17**). This suggests that these were not a distinct set of artefacts, which tend to have specific mutational profiles. Finally, we compared various genomic features of these branch-creating mutations to the overall mutation set. Although for 2/37 there was a suggestion of a weak association (distance to ALU repeat region, q value = 0.03; distance to z-DNA motif, q value = 0.06), there was no obvious relationship visually (**Reviewer Figure 18**). Overall, we believe the vast majority of the branch-creating mutations to be genuine and the trees robust.

We have discussed these points in the manuscript (Methods, ‘Assessment of incorrect phylogeny structure as an artefactual cause of PVVs’; p. 37).

Sample_ID	Branch-creating mutations	Full mutation set
AX001_4_01	11.4×	11.5×
KX002_2_01	13.2×	13.3×
KX003_5_01	14.6×	14.6×
KX004_5_01	14.1×	14.1×
KX007_2_01	15.2×	15.2×
KX008_2_01	14.6×	14.7×
KX009_1_01	14.9×	14.9×
KX010_1_01	16.2×	16.2×
PX002_2_01	13.5×	13.7×
SX001_5_01	13.7×	13.2×

Reviewer Table 3. Coverage of the branch-creating mutations. The mean coverage of all PVV branch-creating mutations is shown relative to the overall mean coverage of the sample.

Reviewer Figure 17. 96-mutation profile for the branch-creating mutations.

Reviewer Figure 18. Distribution of ALU repeat distance and z-DNA motif distance quantiles of the branch-creating mutations. Density plot showing the quantile distributions of the branch-creating mutations for 'distance to ALU repeat' and 'distance to z-DNA motif' metrics, assessed as quantiles from the metric measured from the positions of a large set of single nucleotide variants from blood. Under the null model, these would show a uniform distribution. ALU_re_dist_log10, distance to nearest ALU repeat measured on a log₁₀ scale; z_dna_motif_dist_log10, distance to nearest z-DNA motif, measured on a log₁₀ scale.

Along those lines, could you put more into characterizing the results of the trees – something along the lines of bootstrapping or general tree confidence? There's a lot resting on the structure (comment above) and the branch lengths, it would be nice to know how robust the trees are.

In general, the phylogenies built from the WGS of single-cell colonies are extremely high confidence compared to phylogenies from most other contexts. Compared to phylogenies from the field of evolutionary biology there are a number of features that contribute to this:

- (1) the whole 3.2Gb human genome represents an enormous alignment,
- (2) The 17 mutations per year of HSCs is a low mutation rate, such that independent acquisition of mutations (i.e. convergent evolution), and reversion of mutations are rare events
- (3) The high quality of the inference for germline SNPs gives a confident 'root' to the phylogeny
- (4) The short period of time for somatic evolution (i.e. the human lifespan) further contributes to the rarity of convergent evolution and reversion events.

In addition, compared to phylogenies built from true single-cell sequencing (as compared to the single-cell derived colony sequencing in our studies), the data is much cleaner, more evenly distributed across the genome and has no whole genome amplification artefacts – therefore, our data do not suffer from the issues of missing data and false negatives to nearly the same degree as for direct single-cell sequencing. Many of these features, and the discussion of the use of the maximum parsimony approach to phylogeny reconstruction are contained in our recent paper in Nature Protocols⁴.

Benchmarking analyses were performed within the original manuscripts of the blood phylogenies^{2,5,13} to assess phylogeny robustness. For 27 of the blood phylogenies (accounting 440/501 of the PVVs) we have repeated and summarised some of these analyses below.

Visualisation of the genotype matrices

One of the best ways to get a sense of the data is to visualise a heatmap of the raw genotype matrices that were fed into the phylogeny algorithm. Here one can see how many genotypes are missing, and how consistent the data might be with a phylogeny. Although these are too large to show here in their entirety, we have included the matrices with mutations shared by at least 5 samples, randomly down-sampled to a maximum of 1,000 mutations (**Reviewer Figure 19**).

Reviewer Figure 19. Raw genotype matrices for 27 blood phylogenies. These are clustered heatmaps of the raw genotype matrices. This data was fed into MPBoot for the subsequent phylogeny building. 'Positive' genotypes are shown in red, 'negative' in blue, and 'unknown' in grey. Genotypes are based on variant allele fraction and depth at each site. The matrices have been down-sampled to include only mutations shared by ≥ 5 samples and to a maximum of 1,000 mutations.

Reviewer Figure 19 (continued).

Reviewer Figure 19 (continued).

Comparison with alternative phylogeny reconstruction algorithms

We performed comparisons of the trees inferred using MPBoot compared to other phylogeny-inference algorithms including IQ-tree¹¹ (<http://www.iqtree.org/>) and SCITE¹². IQ-tree is a maximum likelihood algorithm, which we implemented using equal base and mutation probabilities (the Jukes-Cantor model). SCITE is designed for noisy single-cell data, and employs a Markov chain Monte Carlo

sampling scheme to infer a maximum-likelihood phylogeny. In general, these showed very high concordance as assessed by the Robinson-Foulds similarity or Quartet similarity (**Reviewer Table 4**). Where differences did occur these tended to affect the ordering of the early embryonic branches where there are fewer mutations supporting each branch point (**Reviewer Figure 20**). Notably, SCITE frequently had lower Robinson-Foulds similarities as it frequently creates polytomies if there is any uncertainty around the precise phylogeny e.g. at sites of PVVs. However, the Quartet similarity, which is less sensitive to these subtle differences, remained high in these cases. Importantly, differences rarely affected the lesion nodes/ lesion repair nodes that defined the phylogeny-violating variants (see answer to previous question).

SampleID	IQTree		SCITE	
	Robinson-Foulds similarity	Quartet similarity	Robinson-Foulds similarity	Quartet similarity
Pair11	0.9902	0.9982	0.9300	0.9913
Pair13	0.9777	0.9939	0.9344	0.9936
Pair21	0.9884	0.9539	0.9525	0.9537
Pair24	0.9756	0.9999	0.9567	0.9993
Pair25	0.9881	1.0000	0.9501	0.9990
Pair28	0.9956	0.9999	0.9733	0.9993
Pair31	0.9657	0.9909	0.9261	0.9790
Pair38	0.9702	0.9896	0.9628	0.9784
Pair40	0.9614	0.9984	0.8488	0.9858
Pair41	0.9806	0.9998	0.9612	0.9998
PD34493_FINAL	0.9943	0.9991	0.9886	0.9967
PD41276_v2_FINAL_rho05	0.9937	0.9974	1.0000	1.0000
PD41305_v3_FINAL	0.9872	0.9991	0.9936	1.0000
CB001_3_01	0.9968	1.0000	0.9937	1.0000
CB002_2_01	0.9929	1.0000	0.9512	0.9995
AX001_4_01	0.9211	0.9920	0.8647	0.9768
KX001_4_01	0.9967	1.0000	0.9398	0.9996
KX002_2_01	0.9839	1.0000	0.9527	0.9995
KX003_5_01	0.9847	0.9937	0.9537	0.9997
KX004_5_01	0.9976	Unable to run with >477 samples	Unable to build tree	Unable to build tree
KX007_2_01	0.9167	0.9934	0.9011	0.9927
KX008_2_01	0.9969	1.0000	0.9569	0.9977
KX009_1_01	0.9431	0.8289	0.9187	0.9976
KX010_1_01	1.0000	1.0000	0.9918	1.0000
PX001_2_01	0.8996	0.9304	Unable to build tree	Unable to build tree
PX002_2_01	0.9931	0.9996	0.9347	0.9973
SX001_5_01	0.9787	1.0000	0.9341	0.9994

Reviewer Table 4 (included in manuscript as Table S4). Similarity scores comparing the phylogeny inferred by MPBoot (used in the analysis), and alternative phylogeny-inference algorithms IQ-tree and SCITE. For each comparison, the Robinson-Foulds and Quartet similarities are given. Due to the extremely large phylogenies, and slower computation, SCITE did not complete analysis for PX001_2_01 or KX004_5_01 within the 28 days allowed by our longest compute farm queue.

Reviewer Figure 20. Comparison of phylogenies from MPBoot and IQ-tree (left side) or SCITE (right side). In each case the side-by-side comparison highlights all differences between the MPBoot tree and the comparator tree. Branches present in one tree and absent in the comparator are highlighted in red. All branch lengths have been set to a minimum length of 10 such that the structure of the embryonic tree is more readily visible.

Calculation of the disagreement score

The ‘disagreement score’ is a metric that is agnostic to the phylogeny structure, and is a measure of the extent to which the genotype matrix is consistent with the assumptions of a perfect phylogeny: that is, that for any pair of mutations, the sets of positive samples should be either (i) mutually exclusive, or (ii) one set is contained within the other. Of note, the PTVs break this assumption, but in general the disagreement scores are extremely low, generally $10^2 - 10^4$ -fold lower than random shuffles of the genotype matrix (Reviewer Figure 21).

Reviewer Figure 21. Disagreement scores. For 29 of the blood genotype matrices, the disagreement score of the data is shown in red, and the disagreement score of 20 random shuffles of the genotypes for each mutation is shown in blue.

Bootstrapping approaches

The MPBoot bootstrap support and read count bootstrap support values are described in the answer to the previous reviewer point. We can apply these measures across the whole phylogeny structure to get a sense of the confidence across the tree. Both approaches suggest high confidence across the majority of nodes (**Reviewer Figures 22-23**). As suggested by the previous approaches, some individuals have lower support for some of the embryonic branches (13% of embryonic branches had bootstrap support <80 by MPBoot), but almost all post-embryonic branches have high support, with 94% having support values of 100, and 98% having support values >90 by MPBoot (**Reviewer Figure 22**). Importantly, 97% of the blood PVVs relate to nodes in the post-embryonic tree which have the highest confidence.

We have added further elaboration on these points in the Methods ('Assessment of incorrect phylogeny structure as an artefactual cause of PVVs'; p. 35). Reviewer Table 4 has been included in the manuscript as Table S4.

Reviewer Figure 22. MPBoot bootstraps. a, Dot plot showing the MPBoot bootstrap support for all tree nodes arranged by decreasing support values for each phylogeny. Points are joined by a line to better illustrate the trend. **b,** As in **a**, but for only for embryonic nodes, defined as though occurring at <50 mutations of molecular time. **c,** As in **a**, but only for post-embryonic nodes, defined as those occurring at >50 mutations of molecular time.

Reviewer Figure 23. Read count bootstraps. a, Dot plot showing the read count bootstrap support for all tree nodes arranged by decreasing support values for each phylogeny. Points are joined by a line to better illustrate the trend. **b,** As in **a**, but for only for embryonic nodes, defined as though occurring at <50 mutations of molecular

time. **c.** As in **a.**, but only for post-embryonic nodes, defined as those occurring at >50 mutations of molecular time.

When you say, ‘In all but one case, there was more than 1 somatic mutation that convincingly confirmed the consensus phylogeny – thus, for these branches, there was considerably more evidence for the original phylogeny than for the alternative phylogeny suggested by the single PVV’ (section ‘Assessment of incorrect phylogeny structure as an artefactual cause of PVVs’), what are the numbers here? Again more detail would be helpful.

We agree that more detail would be helpful here. **Reviewer Figure 24** shows the original histogram of the number of mutations confirming the consensus phylogeny, which corresponds to the lesion duration. The mode for this number is 12. The PVV that had only 1 mutation confirming the consensus phylogeny was in fact highlighted as likely artefactual in the previous analysis and has been removed from the dataset.

We further looked into all PVVs with <5 mutations confirming the consensus phylogeny, with a total of 19 PVVs meeting this criterion. 11/19 have already been discussed in response to the previous questions and were highlighted as potentially low confidence by either the MPBoot bootstrap support, read count bootstrap support or alternative tree-building approaches. We visualised the remaining 8 mutations, all of which appeared genuine PVVs caused by prolonged DNA lesions rather than any other mechanism (**Reviewer Figure 25**).

We have discussed these points in the manuscript (Methods, ‘Assessment of incorrect phylogeny structure as an artefactual cause of PVVs’, pp. 34-35).

Reviewer Figure 24. Numbers of branch-creating mutations. Histogram showing the number of branch-creating mutations informing the consensus phylogeny for each phylogeny violating variant. Notably, this corresponds to the lesion duration.

PD41276 : 13-87847439-G-A

PD41276 : 2-189419015-C-T

PD41276 : X-116472701-C-T

KX004 : 11-57164205-C-T, 12-123174086-G-A

KX004 : 12-5632704-G-A

SX001: 2-77159008-T-A

Pair31 : 3-83477165-C-T

Pair31 : 8-113656080-C-T

Reviewer Figure 25. Visualisation of mutations with low numbers of mutations confirming the consensus phylogeny. Visualisation of all phylogeny-violating variants with <5 mutations supporting the consensus phylogeny. This does not include PVVs highlighted as low confidence by the bootstrapping or alternate phylogeny-building methods in the previous question. The phylogenies include only the clade defined by the lesion node, and have been truncated to better illustrate the relevant features near the root. The y axis is molecular time, stated relative to the time of the lesion node (i.e. the root of the subtree shown). The heatmaps show the VAFs of mutations on a scale of white (absent) to orange (for the PVVs) or blue (for the confirmatory mutations). Mutation references are shown in the format Chromosome-Position-Reference base-Mutant base.

Would it be possible to validate this in public cell culture data? It would be nice to see MAVs in a more artificial setting where the approaches used here could be validated. The best I could come up with is from the paper 'Quantification of somatic mutation flow across individual cell division events'

by lineage sequencing', but it would be great to see validation of some MAVs in a more constrained/synthetic system where some of the biological confoundings has been taken out. Maybe some of your groups' previous data sets could be used here too. Maybe the resolution isn't there in these data sets, but it would add a lot of confidence to the interpretations of this paper.

MAVs have indeed been validated in a model of mouse tumorigenesis. In the paper from Aitken et al¹⁴, mouse hepatocellular carcinomas induced by treatment with a single injection of diethylnitrosamine (DEN) showed frequent multi-allelic sites, in some cases showing sites with all 4 possible alleles present in the tumour (Figure 4 from Aitken et al¹⁴). While PVVs are currently difficult to validate directly in existing artificial datasets, given the need for high-resolution phylogenies, the same paper confirmed the concept that a single lesion may lead to alternating reference/ variant alleles at a given site by demonstrating combinatorial diversity of reference/ variant alleles on single reads at two nearby mutated loci (Figure 4d from Aitken et al¹⁴).

3e is a neat observation; again, it would be good to attempt to validate this class of indel mutations. Is the sample signature found in public data sets of blood cancers?

Deeper analysis of the mutation signatures is available in the original manuscript analysing this chemotherapy data, which is now available as a preprint¹⁵. **We now cite this wider analysis in our manuscript (Results, 'Numbers and signatures of MAVs and PVVs'; p. 11).**

References

1. Lee-Six, H. *et al.* Population dynamics of normal human blood inferred from somatic mutations. *Nature* **561**, 473–478 (2018).
2. Spencer Chapman, M. *et al.* Lineage tracing of human development through somatic mutations. *Nature* **595**, 85–90 (2021).
3. Junyent, S. *et al.* The first two blastomeres contribute unequally to the human embryo. *Cell* **187**, 2838–2854.e17 (2024).
4. Coorens, T. H. H. *et al.* Reconstructing phylogenetic trees from genome-wide somatic mutations in clonal samples. *Nat Protoc* **19**, 1866–1886 (2024).
5. Mitchell, E. *et al.* Clonal dynamics of haematopoiesis across the human lifespan. *Nature* **606**, 343–350 (2022).
6. Gelman, A. *et al.* *Bayesian Data Analysis*. (CRC Press, Boca Raton, FL, 2013).
7. Li, Y. *et al.* Patterns of somatic structural variation in human cancer genomes. *Nature* **578**, 112–121 (2020).
8. Consortium, R. E. *et al.* Integrative analysis of 111 reference human epigenomes. *Nature* **518**, 317–329 (2015).
9. Thi Hoang, D. *et al.* MPBoot: fast phylogenetic maximum parsimony tree inference and bootstrap approximation. *BMC Ecol Evol* **18**, 11 (2018).
10. Coorens, T. H. H. *et al.* Extensive phylogenies of human development inferred from somatic mutations. *Nature* **597**, 387–392 (2021).
11. Minh, B. Q. *et al.* IQ-TREE 2: New Models and Efficient Methods for Phylogenetic Inference in the Genomic Era. *Mol Biol Evol* **37**, 1530–1534 (2020).
12. Jahn, K., Kuipers, J. & Beerewinkel, N. Tree inference for single-cell data. *Genome Biol* **17**, 86 (2016).
13. Fabre, M. A. *et al.* The longitudinal dynamics and natural history of clonal haematopoiesis. *Nature* **606**, 335–342 (2022).
14. Aitken, S. J. *et al.* Pervasive lesion segregation shapes cancer genome evolution. *Nature* **583**, 265–270 (2020).
15. Mitchell, E. *et al.* The long-term effects of chemotherapy on normal blood cells. *bioRxiv* 2024.05.20.594942 (2024) doi:10.1101/2024.05.20.594942.

Referees' comments:

Referee #1 (Remarks to the Author):

The authors have fully and satisfactorily addressed all the points I raised in first-round review. The updated ABC analysis motivated by referee #2 comments appears to have been important in accounting for the effects of lesion loss from the observed phylogenies, though the overall conclusions remain unchanged. Despite the substantial updates to the manuscript in response to reviewer comments, the main body of the manuscript remains concise and well presented, and I find the revised Figure 2 presentation much easier to interpret. I remain enthusiastically supportive of publication and I have no further points to raise.

Referee #1 (Remarks on code availability): I have reviewed the content of the github repository but have not currently attempted to use the code. The code does provide a detailed README with guidance on installing and running. There do appear to be hard-coded file paths that would need to be updated for local install but this is noted in the README. Orchestration scripts assume the LSF scheduler, so some investment of time and effort would be required to get the code working at a new site, but this is entirely reasonable and overall I consider the code to be well presented and documented.

We thank the reviewer for these positive comments.

Referee #2 (Remarks to the Author):

The rebuttal and revised manuscript are excellent and further improve what I thought was already a creative, fascinating and rigorous study.

The new simulation framework – and thorough assessment of the accuracy of model inferences – is particularly welcome and I commend the authors for their work here. I'm also glad this has caught and corrected what were previous slight misinterpretations of the data.

My previous final comment – about a possible role for the epigenome in facilitating lesion persistence - is relatively well addressed by the analysis of the correlation of lesion position and epigenetic features. It indicates that lesions persistence has some relationship to the methylation status of CpGs in the region. A minor suggestion for a small further analysis is to look at the position of late-arising lesions compared to lesions that arise early in development (as measured on the phylogeny), as presumably the methylation status of the genomic regions where lesions are found will tend to change through development. I emphasise this is a minor point and if it is inaccessible with current data (too unpowered), the manuscript stands without it.

This is an interesting point – it is true that methylation status changes throughout development, as cells commit to different fates, and this would be expected to change the relationship between lesion occurrence and repair. Unfortunately, as the reviewer suggests, we are underpowered to address this convincingly, with only 29 persistent lesions timed to the *in utero* phase of life. **We have suggested this would be an interesting area for future study in the manuscript (Results; 'Frequency and properties of lesions causing PVVs in HSPCs').**

Referee #3 (Remarks to the Author):

The authors do a nice job validating the tree reconstructions and filling in some missing pieces for me. I'm happy to see the paper accepted as-is.

Referee #3 (Remarks on code availability):

Generally, publically available tools were used here (I'm generally strict about this stuff)